# Mitigating Variance Caused by Communication in Decentralized Multi-agent Deep Reinforcement Learning

## Abstract

Communication can facilitate agents to gain a better understanding of the environment and to coordinate their behaviors in multi-agent deep reinforcement learning (MADRL). However, in certain applications, communication is not available during execution due to factors such as security concerns or limited resources. This paper focuses on a decentralized MADRL setting where communication is used only during training, but not during execution, enabling the learning of coordinated behaviors while keeping decentralized execution. While beneficial, communication can introduce uncertainty, potentially increasing the variance in the learning process of decentralized agents. We conduct the first theoretical analysis to study the variance that is caused by communication in policy gradients using actor-critic methods. Motivated by our theoretical analysis, we propose modular techniques that are designed based on our analytical findings to reduce the variance in policy gradients with communication. We incorporate these techniques into two existing algorithms developed for decentralized MADRL with communication and evaluate them on multiple multi-agent tasks in the StarCraft Multi-Agent Challenge and Traffic Junction domains. The results demonstrate that decentralized MADRL communication methods extended with our proposed techniques not only achieve high-performing agents but also reduce variance in policy gradients during training.

## 1 Introduction

Numerous real-world scenarios involve multiple agents interacting within a shared environment, spanning domains like autonomous driving (Shalev-Shwartz et al., 2016), robotics (Kober et al., 2013), and game playing (Silver et al., 2017; Brown & Sandholm, 2019). Multi-agent Deep Reinforcement Learning (MADRL) has been widely used to develop cooperative behaviors of agents in partially observable environments (Gronauer & Diepold, 2022; Oroojlooy & Hajinezhad, 2023; Yang & Wang, 2020). MADRL agents can communicate various types of information, including observations, intentions, and experiences, to mitigate the limitations in agent observability and enhance the coordination of their behaviors (Zhu et al., 2024; Zaïem & Bennequin, 2019; Gronauer & Diepold, 2022). In recent years, there has been growing research interest in MADRL that focuses on communication via a range of values as encoded messages, rather than directly sharing agents' private and massive local information (Zhu et al., 2024). These research works are known as MADRL with learning communication (Comm-MADRL), which aims to establish adaptive and learnable communication protocols. Within the Comm-MADRL field, several settings have been utilized, focusing on whether agents are trained in a decentralized or centralized manner, and whether communication is possible during training or policy execution (Gronauer & Diepold, 2022; Zhu et al., 2024).

In practical applications such as UAVs, concerns about security or limited resources often necessitate that agents operate independently without communication, yet in a coordinated manner (Cavalcante et al., 2012; Skorobogatov et al., 2020). In such applications, decentralized and coordinated behaviors of independent agents without sharing information during execution become essential. To support such applications, communication can be introduced only during the training phase, ensuring and enhancing learning coordinated behaviors efficiently and effectively. Previous work has explored the use of actor-critic method and proposed to incorporate communication into critics but not actors such that critics (value functions) guide the training of actors (policies) (Iqbal & Sha, 2019; Liu et al., 2020). During execution, the critics can be discarded, and

only the actors are deployed. By allowing communication among independent critics during training, we can benefit from information sharing while still developing efficient and secure policies. In the rest of this paper, we build on these approaches and coin the term Decentralized Communicating Critics and Decentralized Actors (DCCDA) to refer to the settings where critics can communicate during training while independent actors cannot communicate neither during training nor during execution.

In DCCDA, instead of relying on predefined or full communication, agents learn to communicate low-dimensional messages (e.g., continuous or discrete values) among their respective critics, thereby reducing communication overhead and enabling adaptive communication. Under DCCDA, communication directly influences the critics and indirectly impacts the actors through the guidance provided by the communicating critics. The critics, actors, and communication modules are trained jointly toward convergence and enjoy computational efficiency from decentralized training. Despite the promising applications of DCCDA, communication can introduce challenges when incorporated into the critics. Specifically, communicated messages are often generated in a stochastic manner (Foerster et al., 2016; Jiang & Lu, 2018) such that, from the perspective of receiver agents, using messages as additional inputs to their critics introduces uncertainty during value estimation. As a result, the policy gradients of actors guided by communicating critics may exhibit high variance, leading to low sample efficiency and performance degradation. Previous research has focused on variance analysis in policy gradients without communication (Lyu et al., 2023; 2021), and thus not measuring variance caused by communication.

In this work, we conduct the first theoretical analysis of the variance in policy gradients within Comm-MADRL under the DCCDA setting. Variance analysis is a vigorous method that allows us to investigate the variability and dispersion of policy learning. Through our variance analysis, we prove that in both idealistic communication setting (where critics communicate sound & complete information) and non-idealistic communication setting (where sound & complete information is corrupted with noise), policy gradients under communicating critics (in the DCCDA setting) have equal or higher variance than that of the centralized critic. Our variance analysis motivates us to reduce the variance in policy gradients using communicating critics. A widely used approach for variance reduction is the *baseline* technique (Greensmith et al., 2004; Weaver & Tao, 2001; Wu et al., 2018; Kuba et al., 2021). Existing baseline techniques are designed to mitigate variance arising from states or actions, while do not account for the variance induced by communication among agents. Moreover, the application of existing baseline techniques to communication methods is not straightforward, which may not optimally reduce the variance caused by communication.

In this paper, we propose a message-dependent baseline technique that targets the variance caused by communicated messages. We further derive the optimal form of our proposed baseline and theoretically prove that it reduces the variance in policy gradients. Additionally, we observe that the introduction of our variance reduction technique can negatively affect the learning performance of agents. To mitigate this effect, we propose a novel use of KL divergence as a regularization technique for aligning the actors and critics. Specifically, our proposed KL regularization technique ensures that the experience generated by the actors is aligned with the decentralized communicating critics, thereby enhancing the critics' learning process. The two proposed techniques can be applied to any Comm-MADRL method under DCCDA. To show the effectiveness and efficiency of our proposed techniques, we extend two existing MADRL methods under the DCCDA setting and evaluate the two methods with and without our techniques on multiple tasks in two benchmark environments, StarCraft Multi-Agent Challenge (SMAC) (Samvelyan et al., 2019) and Traffic Junction (Singh et al., 2019). The results show that our proposed techniques can reduce the variance in policy gradients caused by communication under DCCDA and significantly improve learning performance.

## 2   Related Works

To position our focused DCCDA setting within the broader literature on MADRL, we illustrate various settings, with and without learning communication, across training and execution phases in Figure 1. Note that we specifically focus on actor-critic methods, which align with the DCCDA setting used in our work. We have summarized 5 settings in the MADRL literature: (Setting 1) Centralized Training and Decentralized Execution (CTDE) without learning communication, (Setting 2) CTDE with communicating actors, (Setting 3) Decentralized Training and Decentralized Execution (DTDE) without learning communication, (Setting

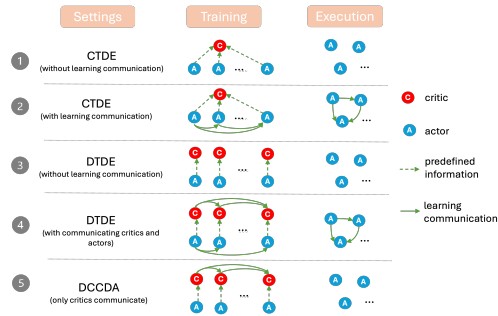

Figure 1: The training and execution phases for CTDE (without communication), CTDE (with communication), DTDE (without communication), and DCCDA using actor-critic methods.

4) DTDE with communicating critics and actors, and (Setting 5) DCCDA which features communicating critics.[1] Among them, Settings 1&2 use predefined full information during centralized training. Settings 2&4 require communicating actors during both training and execution, which may not satisfy practical requirements of security and limited resource as motivated in the introduction. Setting 3 is different from other settings as communication is not utilized in training or execution phases. Setting 5 is the DCCDA, which is fundamentally different from other settings, does not rely on predefined (full) information and agents keep their own critics and actors. DCCDA can be comparable to Setting 1 in certain cases, particularly when critics in DCCDA share full information deterministically, without any stochasticity in their messages. However, the key difference between Setting 1 and DCCDA is that Setting 1 utilizes a single centralized critic, whereas DCCDA employs a separate critic for each agent to improve learning efficiency. In DCCDA, agents learn how and to whom they communicate. By using low-dimensional messages, the input dimensionality of each critic in DCCDA can be significantly smaller than that of the centralized critic in Setting 1, which brings computational efficiency. Due to the similarity between DCDDA and Setting 1 in certain cases, we compare DCCDA with Setting 1 in our theoretical analysis to show that they have different variance properties. We also illustrate the differences in the schematic diagrams of DCCDA and CTDE in Appendix D.1.

**Learning Communication in MADRL**  Previous works mainly focus on learning efficient and effective communication to improve task-level performance under either the CTDE with communication setting (Foerster et al., 2016; Zhang et al., 2019; Das et al., 2019) (**Setting** 2) or the DTDE with communication setting (Iqbal & Sha, 2019; Liu et al., 2020) (**Settings** 4 **and** 5). In CTDE with communication, existing research works utilize either a shared Q-function (Foerster et al., 2016; Jiang & Lu, 2018; Peng et al., 2017), centralized but factorized Q-functions (Zhang et al., 2019; Wang et al., 2020b; Zhang et al., 2020; Yuan et al., 2022; Guan et al., 2023; Sun et al., 2024), or simply a joint value function/returns (Sukhbaatar et al., 2016; Das et al., 2019; Mao et al., 2020; Ding et al., 2020; Guo et al., 2023; Sun et al., 2024) to enable efficient training of communication architectures. Specifically, in actor-critic methods under CTDE with communication, agents ues communicating actors, where encoded messages are often viewed as additional inputs for policies, such as CommNet (Sukhbaatar et al., 2016), BiCNet (Peng et al., 2017), ATOC (Jiang & Lu, 2018), TarMAC (Das et al., 2019), I2C (Ding et al., 2020), GACML (Mao et al., 2020), and T2MAC (Sun et al., 2024). These methods require explicit message transmission among agents during both policy updates in training and policy execution.

Compared to CTDE with communication, communication in the DTDE setting is under-explored. The existing works mainly rely on actor-critic methods (Iqbal & Sha, 2019; Liu et al., 2020; Niu et al., 2021; Chen et al., 2024) (**Settings** 4 **and** 5). When communication is allowed between individual critics (i.e., **the DCCDA setting**), learning communication relies on MAAC (Iqbal & Sha, 2019) and its variant GAAC (Liu et al., 2020). In MAAC, agents individually determine their actions under their Q-functions, while receiving and aggregating information from other agents during learning Q-functions. Based on MAAC, GAAC proposes to incorporate graph neural networks in the critic (Q) network to aggregate important information from neighboring agents. MAAC and GAAC can realize fully decentralized execution by discarding their

---

[1]For simplicity, we use CTDE to refer to CTDE without learning communication, and DTDE to refer to DTDE without learning communication.

communicating critics. In the specific case where agents communicate full information deterministically rather than learning to communicate in a stochastic manner, MADDPG (Lowe et al., 2017) is used to provide each agent with an individual but global Q-function as a critic to guide the learning of decentralized actors. When communication is allowed among actors in DTDE with communication under Setting 4, it can be combined with Setting 5 to enable communication among both critics and actors. In such a combined setting, MAGIC (Niu et al., 2021) is proposed to learn how to schedule encoded messages. In MAGIC, each agent's critic and actor share the same neural network components, including the communication architecture, which also allows the exchange of encoded messages between critics. Additionally, under Setting 4, RGMComm (Chen et al., 2024) introduces a setup where each agent incorporates encoded messages into its policies. Each agent also employs an individual Q-function as a critic, considering the actions and observations of all other agents, implicitly assuming full observability during training. In contrast, DCCDA methods (e.g., MAAC and GAAC) under Setting 5 do not involve message sharing among actors during execution, allowing decentralized execution without communication while still benefiting from communication during training.

Other communication methods using value-based approaches (which do not involve actors) rely on value decomposition techniques, which allow for message sharing among centralized but factorized value functions, such as VBC (Zhang et al., 2019), NDQ (Zhang et al., 2020), MAIC (Yuan et al., 2022), MASIA (Guan et al., 2023), CACOM (Li & Zhang, 2024), and MAGI (Ding et al., 2024). In these methods, policies are derived from decomposed Q-values during execution, and communication is considered when determining the values and execution of actions. In addition to learning communication in MADRL, we also notice research works considering predefined communication with DTDE, where agents share experience buffers to enhance training (Christianos et al., 2020; Gerstgrasser et al., 2023). However, in these studies, communication is not the subject of learning, and directly sharing experience buffers may raise security concerns for individual agents. Moreover, a bunch of research works consider communication between decentralized agents and a central server to achieve low regret using no-regret algorithms, to provide provable regret bounds under cooperative agents (Dubey & Pentland (2021)), asynchronous communication (Min et al. (2023)), and randomized exploration (Hsu et al. (2024)). In contrast, our work considers communication among decentralized agents without assuming the presence of a central server. As none of the above studies address the issue of high variance when communication is learned and integrated into policy gradients, our theoretical analysis, along with the proposed techniques, can offer important insights and solutions to mitigate this challenge effectively.

**Variance Reduction in MADRL** Variance reduction is an essential topic in MADRL (Tucker et al., 2018; Kuba et al., 2021). Previous works have built a theoretical analysis of the variance in policy gradients without considering communication. Lyu et al. (2021; 2023) theoretically contrast policy gradients under CTDE and DTDE settings and claim that the uncertainty of other agents' observations and actions appeared in centralized Q-functions can increase the variance in policy gradients. One of the most successfully applied and extensively studied methods to reduce variance is known as the *baseline* technique (Wu et al., 2018; Foerster et al., 2018; Kuba et al., 2021). Concretely, Wu et al. (2018) utilizes an action-dependent baseline to eliminate the influence of the other agents' policies. Foerster et al. (2018) introduces a counterfactual baseline that marginalizes out a single agent's action, while keeping the other agents' actions fixed. More recently, Kuba et al. (2021) mathematically analyze the variance of policy gradients under CTDE and quantify how agents contribute to the total variance. They propose a baseline technique to achieve minimal variance when estimating policy gradients under CTDE. In summary, existing baseline techniques consider the source of variance from the uncertainty in other agents' observations or actions. In contrast, our baseline technique considers the source of variance from the uncertainty in messages, which are generated in a stochastic manner. To the best of our knowledge, this is the first work to study variance in policy gradients considering learning communication in decentralized MADRL and propose a baseline technique to decrease such variance.

## 3 Preliminaries

### 3.1 Multi-Agent Reinforcement Learning

We consider cooperative multi-agent tasks where a team of agents interacts within the same environment to achieve some common goals. The tasks are generally modeled as decentralized partially observable

Markov decision processes (Dec-POMDPs) (Oliehoek & Amato, 2016). A Dec-POMDP is defined by a tuple $\langle \mathcal{I}, \mathcal{S}, \rho^0, \{\mathcal{A}_i\}, P, \{\mathcal{O}_i\}, O, \mathcal{R}, \gamma \rangle$, where $\mathcal{I}$ is a set of (finite) agents indexed as $\{1, ..., N\}$, $\mathcal{S}$ is a set of environment states, $\rho^0$ is the initial state distribution, $\mathcal{A}_i$ is a set of actions available to agent $i$, and $\mathcal{O}_i$ is a set of observations of agent $i$. We denote a joint action space as $\boldsymbol{\mathcal{A}} = \times_{i \in \mathcal{I}} \mathcal{A}_i$ and a joint observation space as $\boldsymbol{\mathcal{O}} = \times_{i \in \mathcal{I}} \mathcal{O}_i$. Therefore, transition function $P : \mathcal{S} \times \boldsymbol{\mathcal{A}} \to \Delta(\mathcal{S})$ specifies the transition probability $p(s'|s, \boldsymbol{a})$ from state $s \in \mathcal{S}$ to new state $s' \in \mathcal{S}$ given joint action $\boldsymbol{a} = \langle a_1, ..., a_N \rangle$ and $\boldsymbol{a} \in \boldsymbol{\mathcal{A}}$. With the environment transitioning to new state $s'$, given joint action $\boldsymbol{a}$, the probability of a joint observation $\boldsymbol{o} = \langle o_1, ..., o_N \rangle$ ($\boldsymbol{o} \in \boldsymbol{\mathcal{O}}$) is determined according to the observation function $O : \mathcal{S} \times \boldsymbol{\mathcal{A}} \to \Delta(\boldsymbol{\mathcal{O}})$. Each agent then receives a shared reward according to the reward function $\mathcal{R} : \mathcal{S} \times \boldsymbol{\mathcal{A}} \to \mathbb{R}$. The rewards $r_t = \mathcal{R}(s_t, \boldsymbol{a}_t)$ are discounted by the discount factor $\gamma$ over time step $t$. The joint policy $\boldsymbol{\pi}$ of agents induces an on-policy joint Q-function: $Q^{\boldsymbol{\pi}}(s, \boldsymbol{a}) = \mathbb{E}_{s_t \sim P, \boldsymbol{a}_t \sim \boldsymbol{\pi}}[\sum_{t=0}^{T} \gamma^t r_t | s_0 = s, \boldsymbol{a}_0 = \boldsymbol{a}]$, which is the expected discounted return by applying the joint action $\boldsymbol{a}$ and following the joint policy afterward till the time horizon $T$. Note that the on-policy Q-function is learned using a replay buffer, which is refreshed at every update. Whenever the state $s$ is not observable, we use the joint history $\boldsymbol{h} = \{h_1, ..., h_N\}$ instead, where $h_i = (o_0^i, a_0^i, ..., o_t^i)$ is the individual observation-action history of agent $i$ up to time step $t$. Therefore, we obtain the history-based joint Q-function $Q^{\boldsymbol{\pi}}(\boldsymbol{h}, \boldsymbol{a})$ (Lyu et al., 2021). During implementation, histories are often processed using LSTM neural networks (Omidshafiei et al., 2017), which stack past observations and actions into fixed-size memory cells. For notational readability, we omit the time step $t$.

## 3.2 Policy Gradients under Different Settings

**Policy gradients under CTDE** In various policy gradient methods under CTDE, e.g., MAPPO (Yu et al., 2022) and COMA (Foerster et al., 2018), a centralized and joint critic (e.g., a joint Q-function) is used to guide the learning of decentralized actors (policies). Following the setting of Lyu et al. (2023), the CTDE policy gradient of agent $i$ is defined as follows:

$$g_{CTDE}^i \doteq \mathbb{E}_{\boldsymbol{h}, \boldsymbol{a}}[Q^{\boldsymbol{\pi}}(\boldsymbol{h}, \boldsymbol{a})\nabla_{\theta_i} \log \pi_i(a_i|h_i, \theta_i)]$$

where $Q^{\boldsymbol{\pi}}(\boldsymbol{h}, \boldsymbol{a})$ is the on-policy joint values and $\theta_i$ is the parameters of policy $\pi_i$. We further follow the work of Lyu et al. to use $\hat{g}_{CTDE}^i$ to denote the (single-sample) estimate of $g_{CTDE}^i$, i.e., $\hat{g}_{CTDE}^i = Q^{\boldsymbol{\pi}}(\boldsymbol{h}, \boldsymbol{a})\nabla_{\theta_i} \log \pi_i(a_i|h_i, \theta_i)$. Agent $i$ then utilizes $g_{CTDE}^i$ to update parameter $\theta_i$. In CTDE, Q-function $Q^{\boldsymbol{\pi}}(\boldsymbol{h}, \boldsymbol{a})$ gathers information from all agents during training phases, while each actor $\pi_i(a_i|h_i, \theta_i)$ do not communicate and can be used for decentralized execution.

**Policy gradients under DTDE** In various policy gradient methods under DTDE, e.g., IPPO (Yu et al., 2022), an individual and local critic is used to guide the learning of decentralized policies. Similarly, by following the setting of Lyu et al. (2023), the DTDE policy gradient of agent $i$ is defined as follows:

$$g_{DTDE}^i \doteq \mathbb{E}_{\boldsymbol{h}, \boldsymbol{a}}[Q_i^{\pi}(h_i, a_i)\nabla_{\theta_i} \log \pi_i(a_i|h_i, \theta_i)]$$

where $Q_i^{\pi}(h_i, a_i)$ is the on-policy values of agent $i$. Similarly, $\hat{g}_{DTDE}^i$ is used to denote the (single-sample) estimate of $g_{DTDE}^i$, i.e., $\hat{g}_{DTDE}^i = Q_i^{\pi}(h_i, a_i)\nabla_{\theta_i} \log \pi_i(a_i|h_i, \theta_i)$. Each agent $i$ then performs gradient updates to update its actor, parameterized by $\theta_i$. In DTDE, both critics and actors do not communicate and can be trained and executed in a fully decentralized manner.

**Policy gradients under DCCDA** Similar to policy gradients in CTDE and DTDE, we formulate the policy gradients under DCCDA based on the literature (Iqbal & Sha, 2019; Liu et al., 2020). Essentially, we define messages $m_i$ as being generated from a probabilistic message function based on each agent's history ($h_i$) and actions from the actor ($a_i$): $m_i \sim f^{msg}(\cdot|h_i, a_i)$, where function $f^{msg}$ is typically implemented using neural networks with learnable parameters $\theta^{msg}$. Different DCCDA methods define message $m_i$ in various ways. For example, GAAC (Liu et al., 2020) employs a two-layer attention mechanism to produce a vector of continuous values as messages. In our experiments, we consider different strategies for using continuous and discrete values as messages in two respective DCCDA methods. Then, with broadcast communication, we denote the received messages of agent $i$ from all the other agents (denoted as $-i$) as $m_{-i} = \{m_1, ..., m_{i-1}, m_{i+1}, ..., m_N\}$. The DCCDA policy gradient of receiver agent $i$ given by on-policy

values is defined as follows:

$$g^i_{DCCDA} \doteq \mathbb{E}_{\boldsymbol{h},\boldsymbol{a},\boldsymbol{m}}[Q^\pi_i(h_i, a_i, m_{-i})\nabla_{\theta_i} \log \pi_i(a_i|h_i, \theta_i)]$$

where $Q^\pi_i(h_i, a_i, m_{-i})$ is the on-policy Q-values of agent $i$. Similarly, $\hat{g}^i_{DCCDA}$ is used to denote the (single-sample) estimate of $g^i_{DCCDA}$, i.e., $\hat{g}^i_{DCCDA} = Q^\pi_i(h_i, a_i, m_{-i})\nabla_{\theta_i} \log \pi_i(a_i|h_i, \theta_i)$. In DCCDA, each agent has its own critic that takes messages from other agents as an additional input. During training, communication directly affects the critics and indirectly influences the actors via the policy gradients. Regarding the learning of communication with respect to $f^{msg}$, we adopt two strategies following the literature in Comm-MADRL (Foerster et al., 2016): one uses backpropagation to propagate gradients from the critics to the communication modules through the communication channel, while the other leverages environmental rewards to update $f^{msg}$. The critics, actors, and the message functions are learned jointly towards convergence.

## 4 Methods

During training in DCCDA, agents communicate a range of values (or a vector of values) as messages rather than the entire local information, which avoids sharing private information and reduces communication overhead. With communication, messages are then integrated into receiver agents' critics, guiding the gradient updates of their actors (policies). During execution in DCCDA, agents can discard communicating critics and use actors to make decisions independently and locally. We are interested in how messages affect the policy updates of receiver agents in the training period. We specifically focus on how policy gradients diverge, in terms of the variance measurement. Inspired by previous variance analysis in MADRL with CTDE and DTDE settings (Lyu et al., 2021; Kuba et al., 2021; Lyu et al., 2023), we conduct variance analysis in DCCDA policy gradients, focusing on the variance induced by communication. In our variance analysis, we consider both idealistic communication and non-idealistic communication settings. In both scenarios, we demonstrate that the variance of DCCDA policy gradients can be equal to or higher than that of CTDE policy gradients. Motivated by the variance analysis, we propose techniques for practical learning of agents, to reduce the potential variance introduced by communication and to improve value learning.

### 4.1 Variance Analysis

**Idealistic Communication Setting.** We first consider an idealistic communication setting by assuming the existence of a perfect message decoder. Under such idealistic scenarios, the decentralized communicating critics $Q^\pi_i(h_i, a_i, m_{-i})$ and the centralized critics $Q^\pi(\boldsymbol{h}, \boldsymbol{a})$ can be related as communication induces complete and sound information from all agents. However, the probabilistic nature of messages (as commonly used by MADRL with communication methods) can lead to variance in policy gradient samples. Hence, we come to the following theorem:

**Theorem 1.** *The DCCDA sample gradient has a variance greater or equal than that of the CTDE sample gradient in idealistic communication setting: $Var(\hat{g}^i_{DCCDA}) \geq Var(\hat{g}^i_{CTDE})$.*

**Proof Sketch** (full proof in Appendix A.1). We leverage the Bellman equation to find the equivalence between $Q^\pi_i(h_i, a_i, m_{-i})$, used as critics in $\hat{g}^i_{DCCDA}$, and $Q^\pi(\boldsymbol{h}, \boldsymbol{a})$, used as critics in $\hat{g}^i_{CTDE}$. Essentially, as $Q^\pi(\boldsymbol{h}, \boldsymbol{a})$ and the expected value of $Q^\pi_i(h_i, a_i, m_{-i})$ over messages converge to the unique fixed point, we get: $Q^\pi(\boldsymbol{h}, \boldsymbol{a}) = \mathbb{E}_{m_{-i}|\boldsymbol{h},\boldsymbol{a}}[Q^\pi_i(h_i, a_i, m_{-i})]$. Based on this, we find that $\hat{g}^i_{DCCDA}$ and $\hat{g}^i_{CTDE}$ are equal in expectation such that the difference between $Var(\hat{g}^i_{DCCDA})$ and $Var(\hat{g}^i_{CTDE})$ ends with an expectation of the square of gradients minus the square of the expectation of gradients. According to Jensen's inequality, we conclude that $Var(\hat{g}^i_{DCCDA})$ is equal to or higher than $Var(\hat{g}^i_{CTDE})$.

**Non-idealistic Communication Setting.** We now consider the variance analysis under a non-idealistic communication setting, where messages received by agent $i$ are corrupted by a noise term $\epsilon_i$. The noise term can come from the imperfection of decoders, e.g., due to the use of neural networks. To simplify the analysis, we lift noise in received messages to Q-values, where $m_{-i} = <h_{-i}, a_{-i}, \epsilon_i>$ for receiver agent $i$, leading to $Q^\pi_i(h_i, a_i, m_{-i}) = Q^\pi_i(h_i, a_i, <h_{-i}, a_{-i}, \epsilon_i>) = Q^\pi_i(\boldsymbol{h}, \boldsymbol{a}, \epsilon_i)$. The individual but joint Q-function with additive noise, $Q^\pi_i(\boldsymbol{h}, \boldsymbol{a}, \epsilon_i)$, is used as the critics of decentralized actors, forming a noise version of

DCCDA policy gradients $\hat{g}^i_{DCCDA-noise}$. Inspired by Wang et al. (2020a), the noise term can affect the rewards received by each agent, such as flipping the sign in case the reward is binary. We then prove that removing the effect of the noise term (thereby become unbiased) can still increase variance, resulting in the following theorem:

**Theorem 2.** *The noisy version of DCCDA sample gradient has a variance greater or equal than that of the CTDE sample gradient in non-idealistic communication setting: $Var(\hat{g}^i_{DCCDA-noise}) \geq Var(\hat{g}^i_{CTDE})$.*

**Proof Sketch** (full proof in Appendix B.1). We first relate the noise term with the probability of changes in rewards. Inspired by Wang et al. (2020a), a surrogate reward function can be defined to remove the effect of noise in rewards. Upon the surrogate reward function, we define a surrogate Q-function $\hat{Q}^\pi_i(\boldsymbol{h}, \boldsymbol{a}, \epsilon_i)$. By summing up noisy terms $\epsilon_i$, the expected value of $\hat{Q}^\pi_i(\boldsymbol{h}, \boldsymbol{a}, \epsilon_i)$ is shown to be equal to the centralized Q-function $Q^\pi(\boldsymbol{h}, \boldsymbol{a})$ (defined on noise-free rewards in Dec-POMDP), i.e., $Q^\pi(\boldsymbol{h}, \boldsymbol{a}) = \mathbb{E}_{\epsilon_i}[\hat{Q}^\pi_i(\boldsymbol{h}, \boldsymbol{a}, \epsilon_i)]$ by induction proof. The equality greatly simplifies the variance analysis between the noise version of DCCDA policy gradients $\hat{g}^i_{DCCDA-noise}$ (using $\hat{Q}^\pi_i(\boldsymbol{h}, \boldsymbol{a}, \epsilon_i)$ as critics) and the CTDE policy gradients $\hat{g}^i_{CTDE}$ (using $Q^\pi(\boldsymbol{h}, \boldsymbol{a})$ as critics). By comparing the variance in gradients, we have $Var(\hat{g}^i_{DCCDA-noise}) \geq Var(\hat{g}^i_{CTDE})$.

## 4.2 The Message-dependent Baseline Technique and Regularized Policies

Inspired by our variance analysis, communication in DCCDA policy gradients can introduce variance in both the idealistic and non-idealistic communication settings. To mitigate the variance introduced by communication, we adopt a baseline technique inspired by the literature on variance reduction in MADRL (Kuba et al., 2021; Wu et al., 2018). However, existing baseline methods are designed to address variance from states and actions, without considering histories or communicated messages. To address this gap, we propose a novel message-dependent baseline and derive its optimal formulation to minimize the variance induced by communication. As variance reduction may affect the learning performance of MADRL algorithms, we also investigate how to enhance the learning of decentralized communicating critics and decentralized actors in DCCDA. Specifically, the communicating critics implicitly suggest that experience is generated by policies with communication. However, in DCCDA, decentralized policies ($\pi_i(a_i|h_i, \theta_i)$) do not use communication, generating experiences that can mislead the training of communicating critics. Thus, using decentralized policies for the learning of communicating critics can be problematic. To resolve the issue, we further propose a KL divergence term to regularize policies for enhancing the learning of critics. The message-dependent baseline technique and the KL divergence term jointly form our modular techniques, which will be integrated into existing communication methods under the DCCDA setting.

We introduce a novel message-dependent baseline $b_i(h_i, m_{-i})$ to achieve the minimal variance with the presence of communication. For the learning of critics, we use Q-function $Q_i(h_i, a_i, m_{-i})$ to describe the samples of the cumulative discounted return of agent $i$ with communication, where message $m_{-i}$ can be either noisy or noise-free. We assume that $Q_i(h_i, a_i, m_{-i})$ can converge to the true on-policy values $Q^\pi_i(h_i, a_i, m_{-i})$. Based on the definitions, we write out DCCDA policy gradients with the message-dependent baseline (denoted as DCCDA-OB) as follows:

$$g^i_{DCCDA-OB} = \mathbb{E}_{\boldsymbol{h}, \boldsymbol{a}, \boldsymbol{m}}[(Q_i(h_i, a_i, m_{-i}) - b_i(h_i, m_{-i}))\nabla_{\theta_i} \log \pi_i(a_i|h_i, \theta_i)] \tag{1}$$

where actions are sampled from decentralized policies $\pi_i(\cdot|h_i)$ in practice. We then use $\hat{g}^i_{DCCDA-OB}$ to denote the (single-sample) estimate of $g^i_{DCCDA-OB}$, i.e., $\hat{g}^i_{DCCDA-OB} = (Q_i(h_i, a_i, m_{-i}) - b_i(h_i, m_{-i}))\nabla_{\theta_i} \log \pi_i(a_i|h_i, \theta_i)$. We seek the optimal message-dependent baseline $b^*_i(h_i, m_{-i})$ to minimize the variance $Var(\hat{g}^i_{DCCDA-OB})$ of the policy gradient estimate $\hat{g}^i_{DCCDA-OB}$. Therefore, we come to the following theorem:

**Theorem 3.** *The optimal message-dependent baseline for DCCDA-OB gradient estimator is:*

$$b^*_i(h_i, m_{-i}) = \frac{\mathbb{E}_{a_i}[Q_i(h_i, a_i, m_{-i})S]}{\mathbb{E}_{a_i}[S]} \tag{2}$$

*where $S = \nabla_{\theta_i} \log \pi_i(a_i|h_i, \theta_i)^T \nabla_{\theta_i} \log \pi_i(a_i|h_i, \theta_i)$.*

**Proof Sketch** (For the full proof see Appendix C.1). The key idea is to determine an optimal baseline to minimize the variance of $Var(g_{DCCDA-OB}^i)$ by analyzing the derivatives of the variance w.r.t. the baseline. In Equation 2, $S$ is the inner product of the gradient $\nabla_{\theta_i} \log \pi_i(a_i|h_i, \theta_i)$, indicating the magnitude of the gradient vector.

The resulting formula $b_i^*(h_i, m_{-i})$ aligns with previous works on baseline techniques considering states and actions (Kuba et al., 2021; Wu et al., 2018), while we incorporate partially observable information (histories) and communication (messages) into the Q-function. Based on Theorem 3, we have:

**Corollary 1.** *The variance of DCCDA policy gradients is reduced with the optimal message-dependent baseline:* $Var(\hat{g}_{DCCDA-OB}^i) \leq Var(\hat{g}_{DCCDA}^i)$.

**Proof Sketch** (full proof in Appendix C.2). The key idea is to integrate the optimal baseline $b_i^*(h_i, m_{-i})$ into the variance $Var(\hat{g}_{DCCDA-OB}^i)$, which ends with $Var(\hat{g}_{DCCDA}^i)$ minus a non-negative term. Therefore, the variance with the baseline is less than or equal to the variance without the baseline.

Corollary 1 is formulated with respect to idealistic communication setting, i.e., $Var(\hat{g}_{DCCDA-OB}^i) \leq Var(\hat{g}_{DCCDA}^i)$, but it holds also for non-idealistic communication setting, i.e., $Var(\hat{g}_{DCCDA-OB}^i) \leq Var(\hat{g}_{DCCDA-noise}^i)$, by replacing messages with the noisy version $m_{-i} = <h_{-i}, a_{-i}, \epsilon_i>$ and following the same derivations.

The optimal message-dependent baseline we propose relies on communicating critics to generate baseline values. Improving the learning of communicating critics can potentially lead to more accurate baseline estimates. To facilitate this, we propose to enhance the consistency between the critic $Q_i(h_i, a_i, m_{-i})$, which leverages communication during training, and the actor $\pi_i(\cdot|h_i, \theta_i)$, which excludes communication during execution. Specifically, we align the execution policy $\pi_i(\cdot|h_i, \theta_i)$ with the policy induced by the critic's Q-values. Since only the policy $\pi_i(\cdot|h_i, \theta_i)$ is available in implementations, we estimate the policy induced by the critic using the Boltzmann softmax distribution of local Q-values (Cesa-Bianchi et al., 2017). This results in the following KL divergence term to regularize $\pi_i(\cdot|h_i, \theta_i)$ for receiver agent $i$:

$$\mathcal{L}_{KL}(\theta_i) = -D_{KL}(\pi_i(\cdot|h_i, \theta_i)||\text{SoftMax}(\frac{1}{\alpha}Q_i(h_i, \cdot, m_{-i})) \tag{3}$$

where $\alpha$ is a temperature parameter and the KL divergence term $\mathcal{L}_{KL}(\theta_i)$ minimizes the KL divergence between the execution policy $\pi_i(\cdot|h_i, \theta_i)$ and the policy suggested by the critics (which we desire to achieve but not modeled during execution). A higher temperature results in a more uniform policy distribution regarding the Q-values, while a lower temperature results in a more greedy policy distribution regarding the Q-values. The KL divergence term penalizes policy $\pi_i(\cdot|h_i, \theta_i)$ when it assigns a high probability to actions that the estimated policy with communication assigns a low probability to, helping to align decentralized policies with the desired behavior.

The optimal message-dependent baseline (OB) and the KL divergence term (KL) jointly constitute our proposed techniques regarding the variance reduction in policy gradients and the learning of critics. The final gradient for agent $i$ is:

$$g_{DCCDA-OB-KL}^i = g_{DCCDA-OB}^i + \beta\nabla_{\theta_i}\mathcal{L}_{KL}(\theta_i) \tag{4}$$

where $\beta$ is the scaling factor. $g_{DCCDA-OB-KL}^i$ is used to update the policy parameter $\theta_i$ for each agent. In practice, the baseline (Equation 2) and the KL term (Equation 3) are computed using samples of experience (i.e., mini-batches from the experience buffer). Concretely, we store the observation-action-message-reward tuples together with Q-values and policy distribution for each agent in the buffer. Then, we compute the inner product of the policy gradient ($S$) based on an analytical form of the softmax policy (Kuba et al., 2021) and sampled Q-values. Moreover, we estimate the KL divergence through Q-functions and policies with sampled observation-action-message tuples. The algorithmic procedures and the implementations of the proposed baseline and regularization techniques are in Appendix D.3. We also discuss how the learning of agents affects the estimations of KL divergence and the baseline values in Appendix D.3.

**Schematic Diagram of Integrating OB and KL.** To illustrate how our proposed OB and KL techniques integrate with DCCDA methods, we present a schematic diagram that includes actors, critics, communication

modules, a communication channel, and training objectives in Figure 4 in Appendix D.1. In the diagram, the actor, communication (comm), and critic modules are neural networks (e.g., MLPs or RNNs), determined by DCCDA methods. Each agent $i$ decides its action $a_i$ based on its individual history $h_i$. Then, each agent communicates messages $m_i$ sampled from $f^{msg}(m_i|h_i, a_i)$ (see details in Section 3.2). In the experiments, we implement $f^{msg}$ as a multilayer neural network with a recurrent structure, which outputs a range of values to be used as messages. Based on received messages $m_i^{rec}$, agents estimate their Q-values $Q_i(h_i, a_i, m_{-i})$. Different DCCDA methods may adopt different communication strategies for transmitting messages between sender and receiver agents. In the case of broadcast communication, messages are exchanged through a shared communication channel and aggregated into a message vector that excludes the sender's own message, denoted as $m_{-i}$, for each receiver agent. In Figure 4, red arrows highlight the individual training process for each agent's actor, which incorporates our proposed OB and KL techniques (see Equation 4).

**Model-agnostic techniques.** Our proposed techniques focus on the learning of critics and actors. The design of the communication process, including determining message content and how to exchange communicated messages, can be implemented and covered by any Comm-MADRL method under DCCDA. This reflects the adaptability and flexibility of our techniques, making our techniques model-agnostic to existing Comm-MADRL methods under DCCDA. In this paper, we show two cases of how to extend and adapt communication methods using our proposed techniques, resulting in two extended algorithms (the details of the algorithms can be found in Appendix D.2). In the next section, we demonstrate that the two methods using our techniques significantly enhance learning performance and achieve a stable learning process.

## 5  Experiments

We evaluate our proposed techniques in two well-established and challenging multi-agent environments, SMAC (Samvelyan et al., 2019) and Traffic Junction (Das et al., 2019) in MADRL.[2] Both environments consist of a varying number of cooperative agents with shared rewards and show difficulties in coordinating agents' behaviors. We compare with the following methods:

- **CTDE methods**: COMA (Foerster et al., 2018), MAPPO (Yu et al., 2022), and MAT (Wen et al., 2022) that serve as strong baselines in MADRL. These methods are based on actor-critics to align with our setting DCCDA. Notably, MAPPO and MAT have achieved SOTA performance across several MARL benchmarks (Yu et al., 2022; Wen et al., 2022). We compare CTDE methods with DCCDA methods that incorporate our proposed techniques, demonstrating that variance can be reduced without compromising learning performance, even when compared to critics with access to complete and full information.

- **DCCDA methods**: GAAC (Liu et al., 2020) and IPPO (Yu et al., 2022) extended with communication (IPPO-Comm). As mentioned in related work, GAAC is the state-of-the-art method under the DCCDA setting focusing on actor-critic methods. Essentially, GAAC employs a two-layer attention mechanism in the critics to enable learnable communication graphs between agents. Due to the scarcity of communication methods under DCCDA, we adapt the state-of-the-art actor-critic method under the DTDE setting, IPPO, with a communication architecture, named IPPO-Comm. To demonstrate the adaptability and feasibility of our proposed techniques, we adopt different communication and training strategies in IPPO-Comm compared to GAAC. Specifically, IPPO-Comm does not allow gradient backpropagation; instead, it uses Q-values to guide the training of communication modules. In contrast, GAAC enables gradient backpropagation from the critics to the communication modules. Additionally, agents in IPPO-Comm communicate discrete values (integers), whereas agents in GAAC communicate a vector of continuous values. These design choices highlight the plug-in nature of our proposed techniques. Details regarding the communication structure and the use of messages in different environments can be found in Appendix D.2.

- **DCCDA methods with OB-KL**: GAAC-OB-KL and IPPO-Comm-OB-KL. We extend DCCDA methods, GAAC and IPPO-Comm, with our proposed techniques (OB-KL), forming GAAC-OB-KL and IPPO-Comm-OB-KL. Notably, GAAC-OB-KL and IPPO-Comm-OB-KL not only demonstrate the

---

[2]The source code will be made publicly accessible upon the acceptance.

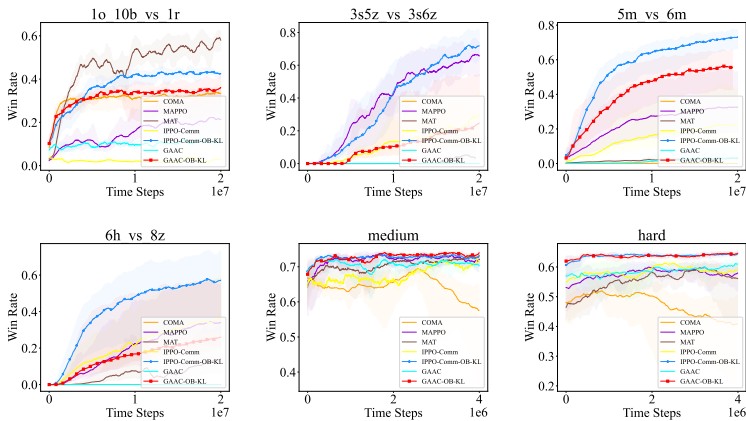

Figure 2: Averaged win rate of all methods.

effectiveness of our techniques but also highlight the possibility of integrating these techniques with various communication constraints and information structures (e.g., using communication graphs).

We illustrate the essential components of all methods and how they differ from each other in MADRL setting, critics and policy regularization techniques in Appendix D.4. Notably, IPPO-Comm-OB-KL and GAAC-OB-KL inherently differ other methods due to the proposed message-dependent baseline and the regularization concerning communication. COMA, MAPPO, and MAT use baseline techniques based on state-value or action-value functions that do not account for encoded messages. The communication method GAAC does not employ a baseline technique. IPPO-Comm, on the other hand, follows the same training strategies as IPPO, which includes a baseline based on state-value functions. For evaluation, all results are reported as the median win rate with a 95% confidence interval. The statistical reports are provided in Appendix D.5. Details regarding the choice of hyperparameters, such as learning rates, are presented in Appendix D.6. Importantly, to enable fair comparison, we use the same training parameters for all methods. The optimization strategies for all methods are either empirical or fine-tuned to ensure fair comparisons. Additionally, our introduced parameters, the temperature $\alpha$ and the scaling factor $\beta$, are fine-tuned using a grid search, with the parameter search results presented in Appendix D.7.

## 5.1 Evaluation Results

**Starcraft Multi-Agent Challenge (SMAC).** SMAC is a real-time strategy game serving as a benchmark in the MADRL community. In SMAC, $N$ units controlled by the learned algorithm try to kill all the enemies, demanding proper cooperation strategies and micro-control of movement and attack. We choose several maps where communication plays an important role in broadening agent's view of the environment or coordinating their strategies to effectively attack the enemies: 1o_10b_vs_1r, 3s5z_vs_3s6z, 5m_vs_6m, and 6h_vs_8z. These maps feature a diverse range of agents, roles, and terrains, each presenting different communication requirements. For example, map 1o_10b_vs_1r involves agents with distinct roles, where an overseer that detects the enemy needs to communicate 10 baneling agents who act to kill the enemy. The variety and richness of the tested maps (tasks) can provide a comprehensive evaluation of different aspects of MADRL. In experiments, all methods are evaluated within 20M time steps. Each episode consists of a maximum of 100 time steps. For each seed, we compute the win rate over 32 evaluation episodes after each 25 training episodes.

The results are presented in Figure 2. As we can see, IPPO-Comm-OB-KL achieves a significantly higher win rate than all other methods on three out of four maps. In map 1o_10b_vs_1r, where agents with different roles need to coordinate, the CTDE method MAT outperforms other methods. This is likely because MAT utilizes a transformer to effectively process agents' observations and generate optimal action sequences, while IPPO-based methods such as IPPO, MAPPO, IPPO-Comm, and IPPO-Comm-OB-KL do not leverage such

a structure. Meanwhile, in map 3s5z_vs_3s6z, the CTDE method MAPPO outperforms other methods up to approximately 15 million steps but is ultimately surpassed by IPPO-Comm-OB-KL. In the remaining maps, IPPO-Comm-OB-KL surpasses all other methods. These results indicate that IPPO-Comm-OB-KL is able to leverage the computational efficiency of decentralized critics to gain an advantage over the CTDE method MAPPO. Furthermore, the performance of DCCDA methods incorporating our proposed techniques depends on the underlying algorithm. Among IPPO-based methods, including both IPPO and MAPPO, IPPO-Comm-OB-KL consistently achieves the highest win rate. Overall, IPPO-Comm-OB-KL and GAAC-OB-KL achieve more stable and effective learning compared to the versions without our proposed techniques, namely IPPO-Comm and GAAC.

**Traffic Junction.** Traffic Junction is a popular benchmark used to test communication ability, where many cars move along two-way roads with one or more road junctions following predefined routes. We test on the medium and hard maps. We evaluate the success rate within 4M time steps. Each episode consists of a maximum of 40 time steps in the medium map and 80 time steps in the hard map. For each 25 training episodes, we compute the win rate over 32 evaluation episodes. We regard an episode as successful if no collision happens during this episode. The results are shown in Figure 2. GAAC-OB-KL achieves a higher win rate compared to all the other methods. IPPO-Comm-OB-KL has a similar win rate as MAT in the medium map while surpassing other methods in the hard map.

## 5.2 The Analysis of Variance in Policy Gradients

We analyze whether the proposed techniques can reduce the variance in gradient updates of policies. To show the magnitude of the gradients under different methods, we use the norm of the policy gradient vector, following the work of Kuba et al. (2021). Then, we show the standard deviation of gradient norms across 8 seeds in Table 1. The lower values the better. As we can see, CTDE methods (COMA, MAPPO, and MAT) have a high variance of gradient norms in most maps. The variance of DCCDA methods without our proposed techniques (IPPO-Comm and GAAC) depends on the underlying optimization strategies. In our implementation, we utilize the same optimization strategy for IPPO-Comm and GAAC as MAPPO to enhance on-policy learning performance, while the variance of IPPO-Comm and GAAC is higher than MAPPO in most maps. Nevertheless, IPPO-Comm-OB-KL and GAAC-OB-KL decrease the variance of gradient norms in all maps compared to IPPO-Comm and GAAC. Compared to CTDE methods, IPPO-Comm-OB-KL achieves a much lower variance in gradient norms across all maps, and GAAC-OB-KL shows lower variance in 5 out of 6 maps. Specifically, in map 3s5z_vs_3s6z, GAAC-OB-KL has a slightly higher variance than MAPPO, which may stem from the higher variance of GAAC (compared to MAPPO). The comparison shows that the amount of variance reduction depends on the underlying communication methods. In most scenarios, IPPO-Comm-OB-KL and GAAC-OB-KL achieve significantly lower policy gradient variance compared to various CTDE methods. This gives IPPO-Comm-OB-KL and GAAC-OB-KL an advantage over CTDE methods in stabilizing the learning process and further enhancing learning performance. We also plot the changes of variance in gradient norms across training steps in Appendix D.7, where IPPO-Comm-OB-KL and GAAC-OB-KL consistently demonstrate low variance in gradients over time steps. The analysis of gradient norms shows that our proposed techniques can lead to not only performance improvements but also less variance in policy gradients.

## 5.3 Ablation Studies

We further conduct experiments to investigate how the proposed techniques affect learning performance. We ablate the KL divergence term and the baseline (OB) from the full version of IPPO-Comm-OB-KL and GAAC-OB-KL and report the results based on the same settings in SMAC and Traffic Junction. As shown in Figure 3, IPPO-Comm-OB-KL surpasses IPPO-Comm-OB (without the KL term) and IPPO-Comm-KL (without the baseline) in all maps, which confirms the advantage of jointly applying the baseline and policy regularization to IPPO-Comm. On the other hand, GAAC-OB-KL outperforms GAAC-OB and GAAC-KL in all maps except for map 6h_vs_8z, where GAAC-OB has a higher win rate than GAAC-OB-KL. In this map, agents may not be able to correctly estimate the policy with communication based on the learned

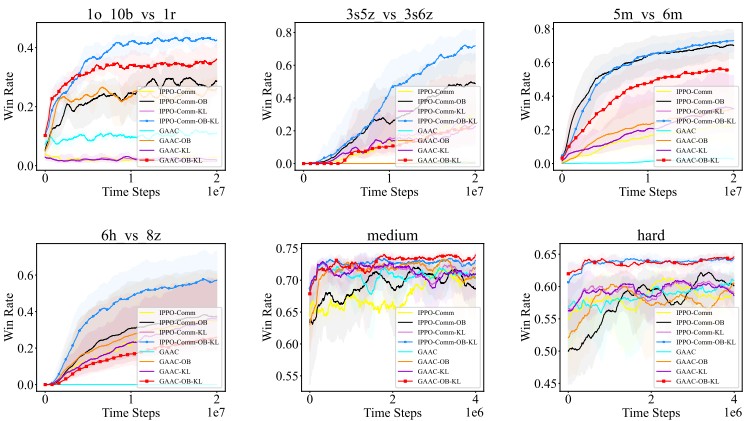

Figure 3: Averaged win rate when ablating OB and KL.

Table 1: The standard deviation of gradient norms, provided on a scale of 0.01. We mark the best (lowest) value in each column.

|  | 1O10BVS1R | 3S5ZVS3S6Z | 5MVS6M | 6HVS8Z | MEDIUM | HARD |
|---|---|---|---|---|---|---|
| COMA | 6.84 | 3.5 | 0.92 | 3.11 | 257.66 | >1000 |
| MAPPO | 0.55 | 0.54 | 2.4 | 1.81 | 0.79 | 0.3 |
| MAT | 4.47 | 5.24 | 4.08 | 20.14 | 10.12 | 14.18 |
| IPPO-COMM | 0.57 | 0.82 | 2.14 | 1.57 | 0.5 | 0.43 |
| IPPO-COMM-OB-KL | **0.39** | **0.24** | **0.46** | 0.9 | **0.01** | 0.01 |
| GAAC | 1.76 | 1.09 | 0.64 | 6.28 | 1.11 | 0.67 |
| GAAC-OB-KL | 0.49 | 0.81 | 0.6 | **0.4** | 0.04 | **<0.01** |

Q-values, leading to significant drops using the KL divergence term. In all the other maps, removing any one of OB and KL results in a decrease in the performance of IPPO-Comm-OB-KL and GAAC-OB-KL.

# 6 Conclusions

In this paper, we investigate the variance of policy gradients caused by communication in decentralized multi-agent deep reinforcement learning. Specifically, we focus on the Decentralized Communicating Critics and Decentralized Actors (DCCDA) setting, where communication is allowed only among critics, while actors do not communicate during training and execution. By variance analysis, we prove that DCCDA policy gradients have a higher or equal variance than the policy gradients under CTDE without communication. We further propose a message-dependent baseline technique for variance reduction in policy gradients, and a regularization technique to improve the learning of critics with experience generated by decentralized policies. We theoretically prove that the optimal message-dependent baseline can reduce the variance in DCCDA policy gradients. The experiments on several tasks show that our proposed techniques achieve not only a higher learning performance but also reduced variance in policy gradients. In the future, we would like to investigate the theoretical variance analysis and the techniques under various communication scenarios. Moreover, we will consider how quantization or bandwidth limitations on continuous message values, as well as the use of adaptive baselines (Schulman et al. (2016)), affect the variance of policy gradients.

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

## Appendix

We list important symbols and definitions used in the following deviations,

- $\boldsymbol{h}_t = \{h_t^1, ..., h_t^N\}$: the joint histories of $N$ agents at time step $t$, which include all previous observations, actions, and the current observation till time step $t$. We use $\boldsymbol{H}$ denote all possible values of $\boldsymbol{h}_t$, i.e., $\boldsymbol{h}_t \in \boldsymbol{H}$. We use $h_t^{-i}$ to denote the other agents' histories and therefore: $\boldsymbol{h}_t = \{h_t^i, h_t^{-i}\}$.

- $\boldsymbol{a}_t = \{a_t^1, ..., a_t^N\}$: the joint actions of $N$ agents at time step $t$, generated by policies. We use $a_t^{-i}$ to denote the other agents' actions and therefore: $\boldsymbol{a}_t = \{a_t^i, a_t^{-i}\}$.

- $\boldsymbol{o}_{t+1} = \{o_{t+1}^1, ..., o_{t+1}^N\}$: the joint new observations of $N$ agents (at time step $t+1$). We use $o_t^{-i}$ to denote the other agents' new observations and therefore: $\boldsymbol{o}_t = \{o_t^i, o_t^{-i}\}$.

- $\boldsymbol{h}_{t+1} = \{\boldsymbol{h}_t, \boldsymbol{a}_t, \boldsymbol{o}_{t+1}\}$ is the next joint history of all agents. We further use $\boldsymbol{hao}$ to denote $\{\boldsymbol{h}_t, \boldsymbol{a}_t, \boldsymbol{o}_{t+1}\}$.

- $\boldsymbol{m}_t = \{m_t^1..., m_t^N\}$: the joint (sending) messages of $N$ agents at time step $t$. During the implementation, each message is generated from a probabilistic message function $f^{msg}$ conditioned on the sender agent's history and action with parameters: $m_j \sim f^{msg}(m_j|h_j, a_j)$, where $j \in \{1, ..., i-1, i+1, ..., N\}$. With broadcasting communication, the received message of each agent $i$ is denoted by $m_{-i} = \{m_t^1..., m_t^{i-1}, m_t^{i+1}, m_t^N\}$, i.e., the set of messages received from all other agents.

- $\boldsymbol{\pi}(\boldsymbol{a}_t|\boldsymbol{h}_t)$: the centralized and joint policies of agents at time step $t$, considering the joint histories $\boldsymbol{h}_t$ of all agents.

- $\pi_i(a_t^i|h_t^i)$: the decentralized policy without communication of each agent $i$ at time step $t$, considering only the individual history $h_t^i$.

- $\pi_i(a_t^i|h_t^i, m_t^{-i})$: the decentralized policy with communication of each agent $i$ at time step $t$, considering the individual history $h_t^i$ and received messages $m_t^{-i}$.

- $\epsilon_i$ is the noise associated to agent $i$, which corrupts received messages from other agents. We define $\epsilon_i \in \mathcal{E}$, where $\mathcal{E}$ is the set of all possible noise values, i.e., integers. Each value of noise represents a specific type of error. The noise term for each agent is sampled from a distribution, i.e., $\epsilon_i \sim p(\epsilon_i)$, where $p(\epsilon_i)$ denotes the probability distribution over $\epsilon_i \in \mathcal{E}$. We assume that a noise term will be sampled at every time step and denote as $\epsilon_t^i$ with probability $p(\epsilon_t^i)$.

Note that we omit the time step of histories, actions, observations, messages, and policies in the following deviations for notation simplification. The notation follows conventions in RL and MARL (Haarnoja et al., 2018; Lyu et al., 2023).

# A   Proofs of the Theoretical Results in Idealistic Communication Setting

We first consider an idealistic communication setting which can simplify complex distributions in the following theoretical analysis. Specifically, we have the following assumption:

**Assumption 1.** In idealistic communication setting, each received message correctly represents the sender's local information, i.e., histories and actions.

The ideal communication assumption implies the existence of a perfect message decoder from the receiver's perspective, capable of reconstructing the sender agents' information from the received messages. Therefore, communication induces complete and sound information from all agents, which can be used to relate communicating critics and centralized critics.

Under the idealistic communication assumption, we come to the following lemma:

**Lemma 1.** *The on-policy centralized critic equals to the expectation of on-policy decentralized communicating critics in idealistic communication setting:* $Q^{\pi}(\boldsymbol{h}, \boldsymbol{a}) = \mathbb{E}_{m_{-i}|\boldsymbol{h}, \boldsymbol{a}}[Q_i^{\pi}(h_i, a_i, m_{-i})]$

*Proof of Lemma 1.* To prove the lemma, we first write the centralized critics and decentralized communicating critics based on the Bellman equations, and then investigate their relation. Followed by Lyu et al. (2023), the centralized Q-function (critic) under the Bellman equality is defined as follows[3]:

$$Q^{\boldsymbol{\pi}}(\boldsymbol{h}, \boldsymbol{a}) = \mathcal{R}(\boldsymbol{h}, \boldsymbol{a}) + \gamma \mathbb{E}_{o|\boldsymbol{h}, \boldsymbol{a}}[\sum_{\boldsymbol{a}'} \boldsymbol{\pi}(\boldsymbol{a}'|\boldsymbol{h}\boldsymbol{a}\boldsymbol{o})Q^{\boldsymbol{\pi}}(\boldsymbol{h}\boldsymbol{a}\boldsymbol{o}, \boldsymbol{a}')] \tag{5}$$

where $\mathcal{R}(\boldsymbol{h}, \boldsymbol{a}) \doteq \mathbb{E}_{s|\boldsymbol{h}, \boldsymbol{a}}[\mathcal{R}(s, \boldsymbol{a})]$ is the joint history reward function and $s \sim p(s|\boldsymbol{h}, \boldsymbol{a})$ is how agents infer the state distribution based on current information. $Q^{\boldsymbol{\pi}}(\boldsymbol{h}, \boldsymbol{a})$ is the expected long-term performance of the team of agents when each agent (with policy $\pi_i$) has observed the individual history $h_i$ and has opted to perform a first action $a_i$.

Similarly, we define a decentralized communicating Q-function (critic) following the Bellman equality as follows:

$$Q_i^{\pi}(h_i, a_i, m_{-i}) = \mathcal{R}_i(h_i, a_i, m_{-i}) + \gamma \mathbb{E}_{o_i, m'_{-i}|h_i, a_i, m_{-i}}[\sum_{a'_i} \pi_i(a'_i|hao_i, m'_{-i})Q_i^{\pi}(hao_i, a'_i, m'_{-i})] \tag{6}$$

where $\mathcal{R}_i(h_i, a_i, m_{-i}) \doteq \mathbb{E}_{s|h_i, a_i, m_{-i}}[\mathcal{R}(s, (a_i, a_{-i}))]$ for $a_{-i}$ from the same joint action. $\mathcal{R}_i(h_i, a_i, m_{-i})$ is the individual reward function with communication when observing individual history $h_i$, action $a_i$, and received messages $m_{-i}$ when other agents opt for action $a_{-i}$. In the equation, $m'_{-i}$ is the next received messages. The individual reward function is defined to consider all possible states under history information. Given the same joint action considered at the current time step, we derive the relation between the joint history reward function and the individual reward function with communication as follows:

---

[3]We use Q-functions instead of critics when involving the Bellman equations.

$$\mathcal{R}(\boldsymbol{h}, \boldsymbol{a}) = \mathbb{E}_{s|\boldsymbol{h}, \boldsymbol{a}}[\mathcal{R}(s, \boldsymbol{a})] \tag{7a}$$

$$= \sum_{s} p(s|\boldsymbol{h}, \boldsymbol{a})\mathcal{R}(s, \boldsymbol{a}) \tag{7b}$$

$$= \sum_{s, m_{-i}} p(s, m_{-i}|\boldsymbol{h}, \boldsymbol{a})\mathcal{R}(s, \boldsymbol{a}) \tag{7c}$$

$$= \sum_{m_{-i}} p(m_{-i}|\boldsymbol{h}, \boldsymbol{a}) \sum_{s} p(s|m_{-i}, \boldsymbol{h}, \boldsymbol{a})\mathcal{R}(s, \boldsymbol{a}) \tag{7d}$$

$$\stackrel{imp.}{=} \sum_{m_{-i}} p(m_{-i}|\boldsymbol{h}, \boldsymbol{a}) \sum_{s} p(s|m_{-i}, h_i, a_i)\mathcal{R}(s, \boldsymbol{a}) \tag{7e}$$

$$\stackrel{def.}{=} \sum_{m_{-i}} p(m_{-i}|\boldsymbol{h}, \boldsymbol{a})\mathcal{R}_i(h_i, a_i, m_{-i}) \tag{7f}$$

$$= \mathbb{E}_{m_{-i}|\boldsymbol{h}, \boldsymbol{a}}[\mathcal{R}_i(h_i, a_i, m_{-i})] \tag{7g}$$

The line 7c is due to the marginalized expectation w.r.t. messages. In line 7d, as received messages can correctly represent sender agents' history and actions (Assumption 1), $s$ is conditionally independent of other agents' information $h_{-i}$ and $a_{-i}$ given received messages $m_{-i}$, and therefore: $p(s|m_{-i}, \boldsymbol{h}, \boldsymbol{a}) = p(s|m_{-i}, h_i, a_i)$. The simplification indicates that each receiver agent $i$ can infer the distribution of the current states ($s$) based on its current information, i.e., individual history ($h_i$), action ($a_i$), and received message ($m_{-i}$). In line 7f, we use the definition of individual reward function with communication in Equation 6. As a result, Equation 7 shows that the joint history reward function is equivalent to the individual reward function by summing up all possible messages when agents opt for the same joint action.

We then analyze how messages affect future rewards in the Bellman equations, considering the stochastic effect of messages in future time steps. In the following equation, we derive the relation between Q-functions for the next time step. We substitute the Q-function $\mathbb{E}_{m'_{-i}|\boldsymbol{hao}, \boldsymbol{a'}}[Q_i^\pi(hao_i, a'_i, m'_{-i})]$ (at the next time step) into the second term of Equation 5 and obtain:

$$\gamma \mathbb{E}_{\boldsymbol{o}|\boldsymbol{h},\boldsymbol{a}}[\sum_{\boldsymbol{a'}} \boldsymbol{\pi}(\boldsymbol{a'}|\boldsymbol{hao})\mathbb{E}_{m'_{-i}|\boldsymbol{hao},\boldsymbol{a'}}[Q_i^\pi(hao_i, a'_i, m'_{-i})]] \tag{8a}$$

$$= \gamma \mathbb{E}_{\boldsymbol{o}|\boldsymbol{h},\boldsymbol{a}}[\sum_{\boldsymbol{a'}} \boldsymbol{\pi}(\boldsymbol{a'}|\boldsymbol{hao}) \sum_{m'_{-i}} p(m'_{-i}|\boldsymbol{hao},\boldsymbol{a'})Q_i^\pi(hao_i, a'_i, m'_{-i})] \tag{8b}$$

$$= \gamma \sum_{\boldsymbol{o}} p(\boldsymbol{o}|\boldsymbol{h},\boldsymbol{a}) \sum_{\boldsymbol{a'}} \boldsymbol{\pi}(\boldsymbol{a'}|\boldsymbol{hao}) \sum_{m'_{-i}} p(m'_{-i}|\boldsymbol{hao},\boldsymbol{a'})Q_i^\pi(hao_i, a'_i, m'_{-i}) \tag{8c}$$

$$= \gamma \sum_{\boldsymbol{o},\boldsymbol{a'},m'_{-i}} p(\boldsymbol{o}|\boldsymbol{h},\boldsymbol{a})\boldsymbol{\pi}(\boldsymbol{a'}|\boldsymbol{hao})p(m'_{-i}|\boldsymbol{hao},\boldsymbol{a'})Q_i^\pi(hao_i, a'_i, m'_{-i}) \tag{8d}$$

$$= \gamma \sum_{\boldsymbol{o},\boldsymbol{a'},m'_{-i}} \frac{p(\boldsymbol{h},\boldsymbol{o},\boldsymbol{a})}{p(\boldsymbol{h},\boldsymbol{a})}\frac{p(\boldsymbol{a'},\boldsymbol{h},\boldsymbol{a},\boldsymbol{o}))}{p(\boldsymbol{h},\boldsymbol{a},\boldsymbol{o})}\frac{p(m'_{-i},\boldsymbol{h},\boldsymbol{a},\boldsymbol{o},\boldsymbol{a'})}{p(\boldsymbol{h},\boldsymbol{a},\boldsymbol{o},\boldsymbol{a'})}Q_i^\pi(hao_i, a'_i, m'_{-i}) \tag{8e}$$

$$= \gamma \sum_{\boldsymbol{o},\boldsymbol{a'},m'_{-i}} \frac{p(m'_{-i},\boldsymbol{h},\boldsymbol{a},\boldsymbol{o},\boldsymbol{a'})}{p(\boldsymbol{h},\boldsymbol{a})}Q_i^\pi(hao_i, a'_i, m'_{-i}) \tag{8f}$$

$$= \gamma \sum_{\boldsymbol{o},\boldsymbol{a'},m'_{-i}} p(\boldsymbol{o},\boldsymbol{a'},m'_{-i}|\boldsymbol{h},\boldsymbol{a})Q_i^\pi(hao_i, a'_i, m'_{-i}) \tag{8g}$$

$$= \gamma \sum_{o_i,a'_i,m'_{-i}} p(o_i,a'_i,m'_{-i}|\boldsymbol{h},\boldsymbol{a})Q_i^\pi(hao_i, a'_i, m'_{-i}) \tag{8h}$$

$$= \gamma \sum_{m_{-i},o_i,a'_i,m'_{-i}} p(m_{-i},o_i,a'_i,m'_{-i}|\boldsymbol{h},\boldsymbol{a})Q_i^\pi(hao_i, a'_i, m'_{-i}) \tag{8i}$$

$$= \gamma \sum_{m_{-i},o_i,a'_i,m'_{-i}} p(m_{-i}|\boldsymbol{h},\boldsymbol{a})p(o_i,a'_i,m'_{-i}|\boldsymbol{h},\boldsymbol{a},m_{-i})Q_i^\pi(hao_i, a'_i, m'_{-i}) \tag{8j}$$

$$= \gamma \sum_{m_{-i},o_i,a'_i,m'_{-i}} p(m_{-i}|\boldsymbol{h},\boldsymbol{a})p(o_i,m'_{-i}|\boldsymbol{h},\boldsymbol{a},m_{-i})p(a'_i|\boldsymbol{h},\boldsymbol{a},m_{-i},o_i,m'_{-i})Q_i^\pi(hao_i, a'_i, m'_{-i}) \tag{8k}$$

$$\stackrel{imp.}{=} \gamma \sum_{m_{-i},o_i,a'_i,m'_{-i}} p(m_{-i}|\boldsymbol{h},\boldsymbol{a})p(o_i,m'_{-i}|h_i,a_i,m_{-i})\pi_i(a'_i|h_i,a_i,o_i,m'_{-i})Q_i^\pi(hao_i, a'_i, m'_{-i}) \tag{8l}$$

$$= \gamma \sum_{m_{-i}} p(m_{-i}|\boldsymbol{h},\boldsymbol{a}) \sum_{o_i,m'_{-i}} p(o_i,m'_{-i}|h_i,a_i,m_{-i}) \sum_{a'_i} \pi_i(a'_i|h_i,a_i,o_i,m'_{-i})Q_i^\pi(hao_i, a'_i, m'_{-i}) \tag{8m}$$

$$= \gamma \mathbb{E}_{m_{-i}|\boldsymbol{h},\boldsymbol{a}}[\mathbb{E}_{o_i,m'_{-i}|h_i,a_i,m_{-i}}[\sum_{a'_i} \pi_i(a'_i|hao_i,m'_{-i})Q_i^\pi(hao_i, a'_i, m'_{-i})]] \tag{8n}$$

In line 8g, we sum up the other agents' next observations ($o_{-i}$) and next actions ($a'_{-i}$) since they do not determine agent $i$' Q-values (due to marginalization). In line 8i, adding a new variable $m_{-i}$ does not change the expected quantity due to marginalized distribution, where all possible messages are considered in the enumeration. From line 8k to line 8l, we use Assumption 1 to simplify the distribution $p(o_i, m'_{-i}|\boldsymbol{h}, \boldsymbol{a}, m_{-i})$ to $p(o_i, m'_{-i}|m_{-i}, h_i, a_i)$: the currently received message ($m_{-i}$) has included other agents' current information (i.e., $h_{-i}$, $a_{-i}$), and agent $i$ can infer the next observations and the next received messages based on its current information. We further simplify the distribution $p(a'_i|\boldsymbol{h}, \boldsymbol{a}, m_{-i}, o_i, m'_{-i})$ to $p(a'_i|h_i, a_i, m_{-i}, o_i, m'_{-i})$, because the next received message $m'_{-i}$ has already included sender agents' (previous) histories $h_{-i}$ and $a_{-i}$ (due to Assumption 1). Since messages are generated to include all historical information, we assume that each agent's policy depends only on current messages and can ignore history information (e.g., previous messages $m_{-i}$) due to Markovian property. This simplifies the distribution $p(a'_i|h_i, a_i, o_i, m_{-i}, m'_{-i})$ to $p(a'_i|h_i, a_i, o_i, m'_{-i})$. The distribution $p(a'_i|h_i, a_i, o_i, m'_{-i})$ indicates that receiver agent $i$ can decide its next action ($a'_i$) based on its next history ($h_i$, $a_i$, $o_i$) and the next received message ($m'_{-i}$). We further denote $p(a'_i|h_i, a_i, o_i, m'_{-i})$ as $\pi_i(a'_i|h_i, a_i, o_i, m'_{-i})$ for notation consistency.

By bringing Equations 7 and 8 together, we achieve the following equality:

$$\mathcal{R}(\boldsymbol{h}, \boldsymbol{a}) + \gamma \mathbb{E}_{\boldsymbol{o}|\boldsymbol{h},\boldsymbol{a}}[\sum_{\boldsymbol{a'}} \boldsymbol{\pi}(\boldsymbol{a'}|\boldsymbol{hao}) \mathbb{E}_{m'_{-i}|\boldsymbol{hao},\boldsymbol{a'}}[Q_i^\pi(hao_i, a'_i, m'_{-i})]]$$

$$\overset{eq.7\&8}{=} \mathbb{E}_{m_{-i}|\boldsymbol{h},\boldsymbol{a}}[\mathcal{R}_i(h_i, a_i, m_{-i})] + \gamma \mathbb{E}_{m_{-i}|\boldsymbol{h},\boldsymbol{a}}[\mathbb{E}_{o_i,m'_{-i}|h_i,a_i,m_{-i}}[\sum_{a'} \pi_i(a'|hao_i, m'_{-i})Q_i^\pi(hao_i, a', m'_{-i})]] \tag{9}$$

$$\overset{def.}{=} \mathbb{E}_{m_{-i}|\boldsymbol{h},\boldsymbol{a}}[\mathcal{R}_i(h_i, a_i, m_{-i}) + \gamma \mathbb{E}_{o_i,m'_{-i}|h_i,a_i,m_{-i}}[\sum_{a'} \pi_i(a'|hao_i, m'_{-i})Q_i^\pi(hao_i, a', m'_{-i})]]$$

$$= \mathbb{E}_{m_{-i}|\boldsymbol{h},\boldsymbol{a}}[Q_i^\pi(h_i, a_i, m_{-i})]$$

where the final equality follows Equation 6. Note that $\mathbb{E}_{m'_{-i}|\boldsymbol{hao},\boldsymbol{a'}}[Q_i^\pi(hao_i, a'_i, m'_{-i})]$ and $\mathbb{E}_{m_{-i}|\boldsymbol{h},\boldsymbol{a}}[Q_i^\pi(h_i, a_i, m_{-i})]$ are the same Q-function evaluated at different time steps. Based on Equation 9, we have,

$$\mathbb{E}_{m_{-i}|\boldsymbol{h},\boldsymbol{a}}[Q_i^\pi(h_i, a_i, m_{-i})] = \mathcal{R}(\boldsymbol{h}, \boldsymbol{a}) + \gamma \mathbb{E}_{\boldsymbol{o}|\boldsymbol{h},\boldsymbol{a}}[\sum_{\boldsymbol{a'}} \boldsymbol{\pi}(\boldsymbol{a'}|\boldsymbol{hao}) \mathbb{E}_{m'_{-i}|\boldsymbol{hao},\boldsymbol{a'}}[Q_i^\pi(hao_i, a'_i, m'_{-i})]] \tag{10}$$

Essentially, after substituting the decentralized communicating Q-function ($\mathbb{E}_{m'_{-i}|\boldsymbol{hao},\boldsymbol{a'}}[Q_i^\pi(hao_i, a'_i, m'_{-i})]$) into the left side of the centralized Q-function under the Bellman equation (Equation 5), we still get back the decentralized communicating Q-function ($\mathbb{E}_{m_{-i}|\boldsymbol{h},\boldsymbol{a}}[Q_i^\pi(h_i, a_i, m_{-i})]$). Followed by Lyu et al. (2023), the centralized Q-function $Q^\pi(\boldsymbol{h}, \boldsymbol{a})$ is the unique fixed point of the Bellman equation defined in Equation 5 (due to the contraction mapping of the Bellman operator). Since $\mathbb{E}_{m_{-i}|\boldsymbol{h},\boldsymbol{a}}[Q_i^\pi(h_i, a_i, m_{-i})]$ appears to be also the solution of the same Bellman equation (Equation 5), we have $Q^\pi(\boldsymbol{h}, \boldsymbol{a}) = \mathbb{E}_{m_{-i}|\boldsymbol{h},\boldsymbol{a}}[Q_i^\pi(h_i, a_i, m_{-i})]$, which completes the proof.

## A.1 Proof of Theorem 1

We have the following theorem:

**Theorem 1.** *The DCCDA sample gradient has a variance greater or equal than that of the CTDE sample gradient in idealistic communication setting: $Var(\hat{g}_{DCCDA}^i) \geq Var(\hat{g}_{CTDE}^i)$.*

*Proof of Theorem 1.* We first check the relation between the two expected gradients $g_{DCCDA}^i$ and $g_{CTDE}^i$, which can greatly simplify the later variance analysis. Based on Lemma 1, as $Q^\pi(\boldsymbol{h}, \boldsymbol{a}) = \mathbb{E}_{m_{-i}|\boldsymbol{h},\boldsymbol{a}}[Q_i^\pi(h_i, a_i, m_{-i})]$, we have,

$$g_{DCCDA}^i = \mathbb{E}_{\boldsymbol{h},\boldsymbol{a},\boldsymbol{m}}[Q_i^\pi(h_i, a_i, m_{-i})\nabla_{\theta_i} \log \pi_i(a_i|h_i, \theta_i)] \tag{11a}$$

$$= \mathbb{E}_{\boldsymbol{h},\boldsymbol{a}}[\mathbb{E}_{\boldsymbol{m}|\boldsymbol{h},\boldsymbol{a}}[Q_i^\pi(h_i, a_i, m_{-i})\nabla_{\theta_i} \log \pi_i(a_i|h_i, \theta_i)] \tag{11b}$$

$$= \mathbb{E}_{\boldsymbol{h},\boldsymbol{a}}[\mathbb{E}_{m_{-i}|\boldsymbol{h},\boldsymbol{a}}[Q_i^\pi(h_i, a_i, m_{-i})\nabla_{\theta_i} \log \pi_i(a_i|h_i, \theta_i)]] \tag{11c}$$

$$= \mathbb{E}_{\boldsymbol{h},\boldsymbol{a}}[\mathbb{E}_{m_{-i}|\boldsymbol{h},\boldsymbol{a}}[Q_i^\pi(h_i, a_i, m_{-i})]\nabla_{\theta_i} \log \pi_i(a_i|h_i, \theta_i)] \tag{11d}$$

$$\overset{lem.1}{=} \mathbb{E}_{\boldsymbol{h},\boldsymbol{a}}[Q^\pi(\boldsymbol{h}, \boldsymbol{a})\nabla_{\theta_i} \log \pi_i(a_i|h_i, \theta_i)] \tag{11e}$$

$$= g_{CTDE}^i \tag{11f}$$

where line 11c is due to the quantity inside the expectation does not depend on agent $i$'s message $m_i$.

Based on Equation 11, we come to the following proof of Theorem 1. Note that $\hat{g}_{CTDE}^i = Q^\pi(\boldsymbol{h}, \boldsymbol{a})\nabla_{\theta_i} \log \pi_i(a_i|h_i, \theta_i)$ and $\hat{g}_{DCCDA}^i = Q_i^\pi(h_i, a_i, m_{-i})\nabla_{\theta_i} \log \pi_i(a_i|h_i, \theta_i)$. Regarding the variance analysis of a vector of gradients, it necessarily involves the covariance matrix, where the diagonal elements denote the variance of each gradient component. The covariance matrix is constructed using the outer product of the gradient vector. To simplify the analysis, we focus on the total variance of the policy gradients, which measures the overall variability across all dimensions. The total variance is given by the trace of the covariance matrix, that is the sum of its diagonal elements, using the squared norm of the gradient vector. We denote $S = \nabla_{\theta_i} \log \pi_i(a_i|h_i, \theta_i)^T \nabla_{\theta_i} \log \pi_i(a_i|h_i, \theta_i)$, which is the inner product (squared norm) of the vector $\nabla_{\theta_i} \log \pi_i(a_i|h_i, \theta_i)$. Then, we come to the following variance analysis:

$$Var(\hat{g}^i_{DCCDA}) - Var(\hat{g}^i_{CTDE}) \tag{12a}$$

$$\overset{def.}{=} \left( \mathbb{E}_{\boldsymbol{h},\boldsymbol{a},\boldsymbol{m}}[(\hat{g}^i_{DCCDA})^2] - \left( \mathbb{E}_{\boldsymbol{h},\boldsymbol{a},\boldsymbol{m}}[\hat{g}^i_{DCCDA}] \right)^2 \right) - \left( \mathbb{E}_{\boldsymbol{h},\boldsymbol{a}}[(\hat{g}^i_{CTDE})^2] - \left( \mathbb{E}_{\boldsymbol{h},\boldsymbol{a}}[\hat{g}^i_{CTDE}] \right)^2 \right) \tag{12b}$$

$$= \left( \mathbb{E}_{\boldsymbol{h},\boldsymbol{a},\boldsymbol{m}}[(\hat{g}^i_{DCCDA})^2] - \mathbb{E}_{\boldsymbol{h},\boldsymbol{a}}[(\hat{g}^i_{CTDE})^2] \right) - \left( \left( \mathbb{E}_{\boldsymbol{h},\boldsymbol{a},\boldsymbol{m}}[\hat{g}^i_{DCCDA}] \right)^2 - \left( \mathbb{E}_{\boldsymbol{h},\boldsymbol{a}}[\hat{g}^i_{CTDE}] \right)^2 \right) \tag{12c}$$

$$\overset{eq.11}{=} \left( \mathbb{E}_{\boldsymbol{h},\boldsymbol{a},\boldsymbol{m}}[(\hat{g}^i_{DCCDA})^2] - \mathbb{E}_{\boldsymbol{h},\boldsymbol{a}}[(\hat{g}^i_{CTDE})^2] \right) - \underbrace{\left( (g^i_{DCCDA})^2 - (g^i_{CTDE})^2 \right)}_{=0} \tag{12d}$$

$$= \mathbb{E}_{\boldsymbol{h},\boldsymbol{a},\boldsymbol{m}}[(\hat{g}^i_{DCCDA})^2] - \mathbb{E}_{\boldsymbol{h},\boldsymbol{a}}[(\hat{g}^i_{CTDE})^2] \tag{12e}$$

$$\overset{def.}{=} \mathbb{E}_{\boldsymbol{h},\boldsymbol{a},\boldsymbol{m}}[(Q^\pi_i(h_i,a_i,m_{-i})\nabla_{\theta_i}\log\pi_i(a_i|h_i,\theta_i))^2] - \mathbb{E}_{\boldsymbol{h},\boldsymbol{a}}[(Q^\pi(\boldsymbol{h},\boldsymbol{a})\nabla_{\theta_i}\log\pi_i(a_i|h_i,\theta_i))^2] \tag{12f}$$

$$\overset{def.}{=} \mathbb{E}_{\boldsymbol{h},\boldsymbol{a},\boldsymbol{m}}[Q^\pi_i(h_i,a_i,m_{-i})^2 S] - \mathbb{E}_{\boldsymbol{h},\boldsymbol{a}}[Q^\pi(\boldsymbol{h},\boldsymbol{a})^2 S] \tag{12g}$$

$$= \mathbb{E}_{\boldsymbol{h},\boldsymbol{a}}[\mathbb{E}_{\boldsymbol{m}|\boldsymbol{h},\boldsymbol{a}}[Q^\pi_i(h_i,a_i,m_{-i})^2 S]] - \mathbb{E}_{\boldsymbol{h},\boldsymbol{a}}[Q^\pi(\boldsymbol{h},\boldsymbol{a})^2 S] \tag{12h}$$

$$= \mathbb{E}_{\boldsymbol{h},\boldsymbol{a}}[\mathbb{E}_{m_{-i}|\boldsymbol{h},\boldsymbol{a}}[Q^\pi_i(h_i,a_i,m_{-i})^2 S]] - \mathbb{E}_{\boldsymbol{h},\boldsymbol{a}}[Q^\pi(\boldsymbol{h},\boldsymbol{a})^2 S] \tag{12i}$$

$$= \mathbb{E}_{\boldsymbol{h},\boldsymbol{a}}[\left( \mathbb{E}_{m_{-i}|\boldsymbol{h},\boldsymbol{a}}[Q^\pi_i(h_i,a_i,m_{-i})^2] - Q^\pi(\boldsymbol{h},\boldsymbol{a})^2 \right) S] \tag{12j}$$

$$= \mathbb{E}_{\boldsymbol{h},\boldsymbol{a},\boldsymbol{m}}[\left( Q^\pi_i(h_i,a_i,m_{-i})^2 - Q^\pi(\boldsymbol{h},\boldsymbol{a})^2 \right) S] \tag{12k}$$

$$\overset{lem.1}{=} \mathbb{E}_{\boldsymbol{h},\boldsymbol{a}}[\underbrace{\left( \mathbb{E}_{m_{-i}|\boldsymbol{h},\boldsymbol{a}}[Q^\pi_i(h_i,a_i,m_{-i})^2] - (\mathbb{E}_{m_{-i}|\boldsymbol{h},\boldsymbol{a}}[Q^\pi_i(h_i,a_i,m_{-i})])^2 \right)}_{u} S] \tag{12l}$$

$$\geq 0 \tag{12m}$$

where line 12b follows the definition of variance. Line 12b follows the definitions that $g^i_{DCCDA} = \mathbb{E}[\hat{g}^i_{DCCDA}]$ and $g^i_{CTDE} = \mathbb{E}[\hat{g}^i_{CTDE}]$. Moreover, due to Equation 11, we have $g^i_{DCCDA} = g^i_{CTDE}$ and then $(g^i_{DCCDA})^2 - (g^i_{CTDE})^2 = 0$ in line 12d. In line 12f, we replace $g^i_{DCCDA}$ and $g^i_{CTDE}$ with the defined formula and it shows a square of the gradient $\nabla_{\theta_i}\log\pi_i(a_i|h_i,\theta_i)$. We then use $S$ to simplify the notation. Line 12i is because the quantity inside the expectation does not depend on agent $i$'s message $m_i$. Line 12l is according to Lemma 1. The final inequality in line 12m follows because $u \geq 0$ by Jensen's inequality: $\mathbb{E}[X^2] \geq (\mathbb{E}[X])^2$, and $S$ is the inner product of a vector itself and therefore non-negative. Therefore, the DCCDA sample gradient has a variance greater or equal than that of the CTDE sample gradient, i.e., $Var(\hat{g}^i_{DCCDA}) \geq Var(\hat{g}^i_{CTDE})$, which completes the proof.

## B   Proofs of the Theoretical Results in Non-idealistic Communication Setting

We further consider a non-idealistic communication setting where messages are corrupted with a noisy term. The noisy term could come from the imperfection of decoders, e.g., due to the use of neural networks. Therefore, from receiver agent $i$'s perspective, received message $m^i_j$ from sender agent $j$ is dedicated to incorporate a noisy term $\epsilon_i \in \mathcal{E}$ when sender agent $j$'s sends a message: $m^i_j = <m_j, \epsilon_i>$, where $m_j \sim f^{msg}(m_j|h_j,a_j)$ is a message broadcast by sender agent $j$ and $\epsilon_i$ is the noise associated to agent $i$. In order to simplify the analysis, we assume there exists a decoder that could decode $h_j$ and $a_j$ from message $m_j$ and therefore we could omit $f_j$, i.e., $m^i_j = <h_j, a_j, \epsilon_i>$, while messages can still be imperfect and remain noisy. Then, the message used by receiver agent $i$ is denoted as $m_{-i} = <m^i_1, ..., m^i_{i-1}, m^i_{i+1}, ..., m^i_N> = <h_{-i}, a_{-i}, \epsilon_i>$

Based on the above non-idealistic communication setting, the communicating critic of receiver agent $i$ is denoted as $Q^\pi_i(h_i, a_i, m_{-i})$, and we have: $Q^\pi_i(h_i, a_i, m_{-i}) = Q^\pi_i(h_i, a_i, <h_{-i}, a_{-i}, \epsilon_i>) = Q^\pi_i(\boldsymbol{h}, \boldsymbol{a}, \epsilon_i)$, where noise in communication is lifted to Q-values, leading to individual but joint critics with additive noise.

We further investigate how noise affects the value estimation in critics and also the relation between individual but joint critics with additive noise $Q^\pi_i(\boldsymbol{h}, \boldsymbol{a}, \epsilon_i)$ and individual but joint critics without noise $Q^\pi_i(\boldsymbol{h}, \boldsymbol{a})$, which is essential for our variance analysis. Due to the noise in communication, rewards become noisy as well. Given history $\boldsymbol{h} \in \boldsymbol{H}$, action $\boldsymbol{a} \in \mathcal{A}$, and noise term $\epsilon_i \in \mathcal{E}$, we define individual but joint critics with noise

at time step $t$ as:

$$Q_{i,t}^{\pi}(\boldsymbol{h}, \boldsymbol{a}, \epsilon_i)$$

$$= \mathbb{E}_{s_{t+k}, \boldsymbol{a}_{t+k}, \epsilon_{t+k}^i}[\sum_{k=0}^{T} \gamma^k \mathcal{R}_i(s_{t+k}, \boldsymbol{a}_{t+k}, \epsilon_{t+k}^i)|\boldsymbol{h}_t = \boldsymbol{h}, \boldsymbol{a}_t = \boldsymbol{a}, \epsilon_t^i = \epsilon_i]$$

$$= \sum_{s_t, s_{t+1} \in \mathcal{S}} p(s_t, s_{t+1}|\boldsymbol{h}, \boldsymbol{a}) \Big( \mathcal{R}_i(s_t, \boldsymbol{a}, \epsilon_i) + \gamma \sum_{\boldsymbol{a}_{t+1} \in \mathcal{A}, \boldsymbol{h}_{t+1} \in \boldsymbol{H}} p(\boldsymbol{h}_{t+1}|s_{t+1}) \boldsymbol{\pi}(\boldsymbol{a}_{t+1}|\boldsymbol{h}_{t+1}) \sum_{\epsilon_{t+1}^i} p(\epsilon_{t+1}^i) Q_{i,t+1}^{\pi}(\boldsymbol{h}_{t+1}, \boldsymbol{a}_{t+1}, \epsilon_{t+1}^i) \Big)$$

$$(13)$$

where $p(s_t, s_{t+1}|\boldsymbol{h}_t, \boldsymbol{a}_t) = p(s_t|\boldsymbol{h}_t, \boldsymbol{a}_t)p(s_{t+1}|\boldsymbol{h}_t, \boldsymbol{a}_t, s_t)$ represents the probabilities transiting from state $s_t$ to state $s_{t+1}$ under joint action $\boldsymbol{a}$ when observing history $\boldsymbol{h}_t$, $p(\boldsymbol{h}_{t+1}|s_{t+1})$ represents the probabilities of history $\boldsymbol{h}_{t+1}$ under future state $s_{t+1}$, $\boldsymbol{\pi}(\boldsymbol{a}_{t+1}|\boldsymbol{h}_{t+1})$ represents the history-based policy, and $\mathcal{R}_i(s_t, \boldsymbol{a}_t, \epsilon_t^i)$ is the noisy reward of agent $i$ at time step $t$. In Equation 13, rewards considering noisy messages are accumulated when observing history $\boldsymbol{h}$, the first joint action to be $\boldsymbol{a}$, and the first noise to be $\epsilon_i$. Without noise, we define individual but joint critics at time step $t$ as:

$$Q_{i,t}^{\pi}(\boldsymbol{h}, \boldsymbol{a})$$

$$= \mathbb{E}_{s_{t+k}, \boldsymbol{a}_{t+k}}[\sum_{k=0}^{T} \gamma^k \mathcal{R}_i(s_{t+k}, \boldsymbol{a}_{t+k}, s_{t+k+1})|\boldsymbol{h}_t = \boldsymbol{h}, \boldsymbol{a}_t = \boldsymbol{a}]$$

$$= \sum_{s_t, s_{t+1} \in \mathcal{S}} p(s_t, s_{t+1}|\boldsymbol{h}, \boldsymbol{a}) \Big( \mathcal{R}_i(s_t, \boldsymbol{a}) + \gamma \sum_{\boldsymbol{a}_{t+1} \in \mathcal{A}, \boldsymbol{h}_{t+1} \in \boldsymbol{H}} p(\boldsymbol{h}_{t+1}|s_{t+1}) \boldsymbol{\pi}(\boldsymbol{a}_{t+1}|\boldsymbol{h}_{t+1}) Q_{i,t+1}^{\pi}(\boldsymbol{h}_{t+1}, \boldsymbol{a}_{t+1}) \Big)$$

$$(14)$$

where $\mathcal{R}_i(s_t, \boldsymbol{a}_t)$ is the true reward of agent $i$ at time step $t$. Without noise, rewards in $Q_{i,t}^{\pi}(\boldsymbol{h}, \boldsymbol{a})$ are accumulated when observing history $\boldsymbol{h}$ and the first joint action to be $\boldsymbol{a}$. Due to shared rewards in the Dec-POMDP setting, we have: $Q_{i,t}^{\pi}(\boldsymbol{h}, \boldsymbol{a}) = Q_t^{\pi}(\boldsymbol{h}, \boldsymbol{a})$ by convergence, where $Q_t^{\pi}(\boldsymbol{h}, \boldsymbol{a})$ is the centralized Q-function at time step $t$.

Based on the individual but joint critics with and without noise, we can see that the noise affects the estimation of Q-values when accumulating rewards. In single-agent reinforcement learning, Wang et al. (2020a) shows that noisy rewards can be compensated by utilizing a surrogate reward function, leading to unbiased value estimation. However, removing the effect of noise that we defined in value estimation may lead to increased variance when summing up noisy rewards. Inspired by the surrogate rewards proposed by Wang et al. (2020a), we prove that in multi-agent reinforcement learning scenarios, using a surrogate reward function can also remove the effect of noise in value estimation (thereby become unbiased), while at the cost of increased variance. In the following proof, we consider a binary case where rewards indicate either success ($\mathbf{r}_+$) or failure ($\mathbf{r}_-$), while this can be generalized to rewards beyond binary (see Wang et al. (2020a)), which employs a confusion matrix where each entry represents the probability of a true reward being observed as a different value. In this non-binary reward setting, conducting a variance analysis requires assuming the domain knowledge of how noise alters each reward. Note that due to the shared rewards in the Dec-POMDP setting, agents have the same values of success or failure, i.e., $\mathbf{r}_+ = \mathbf{r}_+^1 = \mathbf{r}_+^2 = ... = \mathbf{r}_+^N$ and $\mathbf{r}_- = \mathbf{r}_-^1 = \mathbf{r}_-^2 = ... = \mathbf{r}_-^N$, where $\mathbf{r}_+^i$ and $\mathbf{r}_-^i$ are rewards obtained by agent $i$ for success and failure, respectively. In the binary reward case, we further assume the noise can be categorized into two types: those that alter a given reward and those that do not. For simplicity in the theoretical analysis, we also assume the noise is discrete, allowing the use of summation over all possible noisy values. Then, noisy rewards can be characterized by the noise rate parameter $e$, where we have:

$$e := p(\mathcal{R}_i(s_t, \boldsymbol{a}_t, \epsilon_t^i) \neq \mathcal{R}_i(s_t, \boldsymbol{a}_t))$$

where $e$ defines the probability that noisy rewards are different from noise-free rewards under noise $\epsilon_t^i$ and $\mathcal{R}_i(s_t, \boldsymbol{a}_t) \in \mathbf{R} := \{\mathbf{r}_+, \mathbf{r}_-\}$. $e$ implies that the probability of $\mathcal{R}_i(s_t, \boldsymbol{a}_t, \epsilon_t^i) \neq \mathcal{R}_i(s_t, \boldsymbol{a}_t)$ can be affected by the noise term $\epsilon_t^i$. The noise term captures how communication under noisy conditions alters agents' chances of success or failure. With shared rewards, agents have the same values of $e$.

We determine the noisy reward $\mathcal{R}_i(s_t, \boldsymbol{a}_t, \epsilon_t^i)$ based on the noisy term as follows:

Suppose the noise-free (true) reward indicates success (i.e., $\mathcal{R}_i(s_t, \boldsymbol{a}_t) = \mathbf{r}_+$) and there exists a value $\delta_\epsilon \in \mathcal{E}$, if the noise term $\epsilon_t^i \geq \delta_\epsilon$, then the noisy reward equals to the true reward and thus $\mathcal{R}_i(s_t, \boldsymbol{a}_t, \epsilon_t^i) = \mathbf{r}_+$, otherwise if the noise term $\epsilon_t^i < \delta_\epsilon$, the reward is considered to be erroneous and thus $\mathcal{R}_i(s_t, \boldsymbol{a}_t, \epsilon_t^i) = \mathbf{r}_-$.

Let the set of noise terms $\boldsymbol{\epsilon}_+ = \{\epsilon_t^i \in \mathcal{E} | \epsilon_t^i \geq \delta_\epsilon\}$ represents the cases that noisy rewards equal to true rewards and $\boldsymbol{\epsilon}_- = \{\epsilon_t^i \in \mathcal{E} | \epsilon_t^i < \delta_\epsilon\}$ represents the cases that noisy rewards differ from true rewards. We have $\boldsymbol{\epsilon}_+ \cup \boldsymbol{\epsilon}_- = \mathcal{E}$ and $\boldsymbol{\epsilon}_+ \cap \boldsymbol{\epsilon}_- = \emptyset$. Then we denote $E = \{\boldsymbol{\epsilon}_+, \boldsymbol{\epsilon}_-\}$. We determine $e$ as:

$$e = p(\mathcal{R}_i(s_t, \boldsymbol{a}_t, \epsilon_t^i) \neq \mathcal{R}_i(s_t, \boldsymbol{a}_t)) = p(\boldsymbol{\epsilon}_-)$$

Then we have,

$$1 - e = p(\mathcal{R}_i(s_t, \boldsymbol{a}_t, \epsilon_t^i) = \mathcal{R}_i(s_t, \boldsymbol{a}_t)) = p(\boldsymbol{\epsilon}_+)$$

where $\boldsymbol{\epsilon}_+ \in E$ and $\boldsymbol{\epsilon}_- \in E$.

In the following two lemmas, we show that the effect of noisy rewards can be removed upon the inspiration of Wang et al. (2020a), leading to the equality between individual but joint Q-function (without noise) and a defined surrogate Q-function (with noise). The equality between the two Q-functions significantly simplifies the variance analysis in CTDE policy gradients and the noisy version of DCCDA policy gradients. However, it still presents the issue of increased variance, as demonstrated later in Theorem 2.

We list the definition of important symbols used in the proof of Lemma 2 for better reading:

- $r^i \in \{\mathbf{r}_+, \mathbf{r}_-\}$ is the true reward of agent $i$ where noise is not presented.

- $r_\epsilon^i \in \{\mathbf{r}_+, \mathbf{r}_-\}$ is the noisy reward of agent $i$ where noise is presented.

Wang et al. (2020a) propose a surrogate reward function (Equation 1 in the paper of Wang et al.) to help remove the effect of noise in value estimation. Inspired by Wang et al., we define a surrogate reward function $\hat{\mathcal{R}}_i(s_t, \boldsymbol{a}_t, r^i, \epsilon_t^i)$ of agent $i$ as a function of state $s_t$, actions $\boldsymbol{a}_t$, the true reward $r^i$ and the noise term $\epsilon_t^i$:

$$\hat{\mathcal{R}}_i(s_t, \boldsymbol{a}_t, r^i, \epsilon_t^i) := \begin{cases} \dfrac{(1-e)\mathbf{r}_+ - e\,\mathbf{r}_-}{1 - 2e}, & if (r^i = \mathbf{r}_+ \,\&\, \epsilon_t^i \in \boldsymbol{\epsilon}_+)\, or\, (r^i = \mathbf{r}_- \,\&\, \epsilon_t^i \in \boldsymbol{\epsilon}_-), \\[4mm] \dfrac{(1-e)\mathbf{r}_- - e\,\mathbf{r}_+}{1 - 2e}, & if (r^i = \mathbf{r}_+ \,\&\, \epsilon_t^i \in \boldsymbol{\epsilon}_-)\, or\, (r^i = \mathbf{r}_- \,\&\, \epsilon_t^i \in \boldsymbol{\epsilon}_+). \end{cases} \tag{15}$$

where the true rewards and noise term determine noisy rewards, i.e., $r_\epsilon^i = \mathbf{r}_+$ is equivalent to $(r^i = \mathbf{r}_+ \,\&\, \epsilon_t^i \in \boldsymbol{\epsilon}_+)$ or $(r^i = \mathbf{r}_- \,\&\, \epsilon_t^i \in \boldsymbol{\epsilon}_-)$, and $r_\epsilon^i = \mathbf{r}_-$ is equivalent to $(r^i = \mathbf{r}_+ \,\&\, \epsilon_t^i \in \boldsymbol{\epsilon}_-)$ or $(r^i = \mathbf{r}_- \,\&\, \epsilon_t^i \in \boldsymbol{\epsilon}_+)$.

Based on our defined surrogate reward function $\hat{\mathcal{R}}_i(s_t, \boldsymbol{a}_t, r^i, \epsilon_t^i)$, we come to the following lemma:

**Lemma 2.** *We set true reward $r^i = \mathcal{R}_i(s_t, \boldsymbol{a}_t)$, where $r^i \in \{\mathbf{r}_+, \mathbf{r}_-\}$. For any value of $r^i$, with noise term $\epsilon_t^i$, we have: $\mathbb{E}_{\epsilon_t^i}[\hat{\mathcal{R}}_i(s_t, \boldsymbol{a}_t, r^i, \epsilon_t^i)] = \mathcal{R}_i(s_t, \boldsymbol{a}_t)$ in non-idealistic communication setting.*

*Proof of Lemma 2.* In the following proof, we compute the expectation of surrogate rewards under associated probabilities when the true reward is specified. We prove that under any value of the true reward $r^i = \mathcal{R}_i(s_t, \boldsymbol{a}_t)$, the expectation of surrogate reward $\hat{\mathcal{R}}_i(s_t, \boldsymbol{a}_t, r^i, \epsilon_t^i)$ equals to the true reward $\mathcal{R}_i(s_t, \boldsymbol{a}_t)$.

When true reward $r^i = \mathbf{r}_+$, based on the definition of the surrogate reward in Equation 15, the expected values of surrogate rewards under noise will be:

$$\mathbb{E}_{\epsilon_t^i}[\hat{\mathcal{R}}_i(s_t, \boldsymbol{a}_t, r^i = \mathbf{r}_+, \epsilon_t^i)] = (1-e) \cdot \frac{(1-e)\mathbf{r}_+ - e\,\mathbf{r}_-}{1 - 2e} + e \cdot \frac{(1-e)\mathbf{r}_- - e\,\mathbf{r}_+}{1 - 2e} \tag{16a}$$

$$= \frac{(1-e)(1-e)\mathbf{r}_+ - e(1-e)\mathbf{r}_- + e(1-e)\mathbf{r}_- - e^2\mathbf{r}_+}{1 - 2e} \tag{16b}$$

$$= \frac{(1-2e)\mathbf{r}_+}{1 - 2e} \tag{16c}$$

$$= \mathbf{r}_+ \tag{16d}$$

$$= r^i = \mathcal{R}_i(s_t, \boldsymbol{a}_t) \tag{16e}$$

where the value in line 16a comes from the surrogate rewards multiplied with the probabilities $p(\boldsymbol{\epsilon}_+)$ and $p(\boldsymbol{\epsilon}_-)$. When true reward $r^i = \mathbf{r}_-$, it also verifies that $\mathbb{E}_{\epsilon_t^i}[\hat{\mathcal{R}}_i(s_t, \boldsymbol{a}_t, r^i = \mathbf{r}_-, \epsilon_t^i)] = \mathbf{r}_-$. So we have $\mathbb{E}_{\epsilon_t^i}[\hat{\mathcal{R}}_i(s_t, \boldsymbol{a}_t, r^i, \epsilon_t^i)] = \mathcal{R}_i(s_t, \boldsymbol{a}_t)$ for any value of $r^i$, which completes the proof.

We list the definition of important symbols used in the proof of Lemma 3 and the following theorem for better reading:

- $\mathbf{R} = \{\mathbf{r}_+, \mathbf{r}_-\} = \{\mathbf{r}_n^i\}_{n\in\{+,-\}}$ is the set of true rewards of agent $i$ where $\mathbf{r}_+^i$ represents success and $\mathbf{r}_-^i$ represents failure. Due to shared rewards, we have $\mathbf{r}_+^i = \mathbf{r}_+$ and $\mathbf{r}_-^i = \mathbf{r}_-$. Note that we express $r_t^i \in \mathbf{R}$ for reward variable at time step $t$.

- $\hat{\mathbf{R}}_n = \{\hat{\mathbf{r}}_{n,l}^i\}_{l\in\{+,-\}}$ is the set of surrogate reward of agent $i$ when true reward is $\mathbf{r}_n^i$. In the set $\hat{\mathbf{R}}_n$, $\hat{\mathbf{r}}_{n,+}^i$ is the surrogate reward if noisy term $\epsilon_t^i \in \boldsymbol{\epsilon}_+$ and given true reward $\mathbf{r}_n^i$. And, $\hat{\mathbf{r}}_{n,-}^i$ is the surrogate reward if noisy term $\epsilon_t^i \in \boldsymbol{\epsilon}_-$ and given true reward $r_n^i$.

Similar to Equation 13, we define a surrogate Q-function $\hat{Q}_i^\pi(\boldsymbol{h}, \boldsymbol{a}, \epsilon_i)$ at time step $t$ as:

$$
\begin{aligned}
\hat{Q}_{i,t}^\pi(\boldsymbol{h}, \boldsymbol{a}, \epsilon_i) &= \mathbb{E}_{s_{t+k}, \boldsymbol{a}_{t+k}, r_{t+k}^i, \epsilon_{t+k}^i}\Big[\sum_{k=0}^T \gamma^k \hat{\mathcal{R}}_i(s_{t+k}, \boldsymbol{a}_{t+k}, r_{t+k}^i, \epsilon_{t+k}^i)|\boldsymbol{h}_t = \boldsymbol{h}, \boldsymbol{a}_t = \boldsymbol{a}, \epsilon_t^i = \epsilon_i\Big] \\
&= \sum_{s_t, s_{t+1}\in\mathcal{S}} \sum_{r^i\in\mathbf{R}} p(s_t, s_{t+1}, r^i|\boldsymbol{h}, \boldsymbol{a})\Big(\hat{\mathcal{R}}_i(s_t, \boldsymbol{a}, r^i, \epsilon_i)+ \\
&\quad \gamma \sum_{\boldsymbol{a}_{t+1}\in\mathcal{A}, \boldsymbol{h}_{t+1}\in H} p(\boldsymbol{h}_{t+1}|s_{t+1})\boldsymbol{\pi}(\boldsymbol{a}_{t+1}|\boldsymbol{h}_{t+1}) \sum_{\epsilon_{t+1}^i} p(\epsilon_{t+1}^i)\hat{Q}_{i,t+1}^\pi(\boldsymbol{h}_{t+1}, \boldsymbol{a}_{t+1}, \epsilon_{t+1}^i)\Big)
\end{aligned}
\tag{17}
$$

where surrogate rewards are accumulated when observing history $\boldsymbol{h}$, the first joint action to be $\boldsymbol{a}$, and the first noise to be $\epsilon_i$.

According to Lemma 2, the surrogate rewards and true rewards are related at every time step. Based on this, we would like to investigate how the accumulation of rewards in Q-functions relates to each other. Since individual but joint Q-function equals to centralized Q-function with shared rewards, we have the following lemma:

**Lemma 3.** *Given $\mathcal{R}_i(s_t, \boldsymbol{a}_t) = \mathbb{E}_{\epsilon_t^i}[\hat{\mathcal{R}}_i(s_t, \boldsymbol{a}_t, r^i, \epsilon_t^i)]$, the centralized Q-function equals to the expectation of surrogate Q-function in non-idealistic communication setting: $Q^\pi(\boldsymbol{h}, \boldsymbol{a}) = \mathbb{E}_{\epsilon_i}[\hat{Q}_i^\pi(\boldsymbol{h}, \boldsymbol{a}, \epsilon_i)]$.*

*Proof of Lemma 3.* In the following proof, we first relate true rewards and surrogate rewards under transition probabilities. Then we relate individual but joint Q-function and surrogate Q-function upon the summation over true rewards and surrogate rewards per step, respectively. Finally we achieve the equality between the centralized Q-function and the surrogate Q-function.

According to Lemma 2, for any true reward $r^i = \mathbf{r}_n^i$, we have:

$$
\mathbb{E}_{\epsilon_t^i}[\hat{\mathcal{R}}_i(s_t, \boldsymbol{a}_t, r^i = \mathbf{r}_n^i, \epsilon_t^i)] = p(\boldsymbol{\epsilon}_+)\cdot\hat{\mathbf{r}}_{n,+}^i + p(\boldsymbol{\epsilon}_-)\cdot\hat{\mathbf{r}}_{n,-}^i = \sum_{l\in\{+,-\}} p(\boldsymbol{\epsilon}_l)\hat{\mathbf{r}}_{n,l}^i = \mathbf{r}_n^i
\tag{18}
$$

We further use $p(s_t, s_{t+1}, \mathbf{r}_n^i|\boldsymbol{h}_t, \boldsymbol{a}_t)$ to represent the probabilities of a certain true reward $\mathbf{r}_n^i$ when transiting from state $s_t$ to the next state $s_{t+1}$ under joint actions $\boldsymbol{a}_t$ given current history $\boldsymbol{h}_t$. The true rewards and surrogate rewards for every time step $t$ satisfy:

$$
\sum_{s_t, s_{t+1}\in\mathcal{S}} p(s_t, s_{t+1}|\boldsymbol{h}_t, \boldsymbol{a}_t)\,\mathcal{R}_i(s_t, \boldsymbol{a}_t) = \sum_{s_t, s_{t+1}\in\mathcal{S}} \sum_{n\in\{+,-\}} p(s_t, s_{t+1}, \mathbf{r}_n^i|\boldsymbol{h}_t, \boldsymbol{a}_t)\mathbf{r}_n^i
\tag{19a}
$$

$$
\stackrel{eq.18}{=} \sum_{s_t, s_{t+1}\in\mathcal{S}} \sum_{n\in\{+,-\}} p(s_t, s_{t+1}, \mathbf{r}_n^i|\boldsymbol{h}_t, \boldsymbol{a}_t) \sum_{l\in\{+,-\}} p(\boldsymbol{\epsilon}_l)\hat{\mathbf{r}}_{n,l}^i
\tag{19b}
$$

$$
= \sum_{s_t, s_{t+1}\in\mathcal{S}} \sum_{r^i\in\mathbf{R}} p(s_t, s_{t+1}, r^i|\boldsymbol{h}_t, \boldsymbol{a}_t) \sum_{\epsilon_t^i\in\mathcal{E}} p(\epsilon_t^i)\hat{\mathcal{R}}_i(s_t, \boldsymbol{a}_t, r^i, \epsilon_t^i)
\tag{19c}
$$

where $p(s_t, s_{t+1}|\boldsymbol{h}_t, \boldsymbol{a}_t)$ is the transition probabilities giving current history $\boldsymbol{h}_t$ and action $\boldsymbol{a}_t$. Line 19a sums over all possible values of true rewards multiplied by their probabilities. Line 19b is due to Equation 18. In line 19c, we achieve the summation over surrogate rewards by replacing the index with reward variable: a) the summation over index $n$ is transformed into the summation over $r^i \in \mathbf{R}$; b) the summation over the set $E = \{\boldsymbol{\epsilon}_l\}_{l \in \{+,-\}}$ is transformed into the summation over all possible $\epsilon_t^i$ as $\boldsymbol{\epsilon}_+ \cup \boldsymbol{\epsilon}_- = \mathcal{E}$.

Equation 19 implies that the true rewards and surrogate rewards under transitions per time step can be equal in expectation, which can lead to the equality between individual but joint Q-function (accumulating true rewards) and surrogate Q-function (accumulating surrogate rewards). In the following derivations, we first prove that the equality hold at the last time step. Then we can achieve that the equality holds for every time step by induction.

**Base Step:** At terminal time step $T$, we have:

$$Q_{i,T}^{\pi}(\boldsymbol{h}_T, \boldsymbol{a}_T) = \sum_{s_T \in \mathcal{S}} p(s_T, s_{terminal}|\boldsymbol{h}_T, \boldsymbol{a}_T) \mathcal{R}_i(s_T, \boldsymbol{a}_T) \tag{20a}$$

$$\overset{eq.19}{=} \sum_{s_T \in \mathcal{S}} \sum_{r^i \in \mathbf{R}} p(s_T, s_{terminal}, r^i|\boldsymbol{h}_T, \boldsymbol{a}_T) \sum_{\epsilon_T^i \in \mathcal{E}} p(\epsilon_T^i) \hat{\mathcal{R}}_i(s_T, \boldsymbol{a}_T, r^i, \epsilon_T^i) \tag{20b}$$

$$= \sum_{\epsilon_T^i \in \mathcal{E}} p(\epsilon_T^i) \Big( \sum_{s_T \in \mathcal{S}} \sum_{r^i \in \mathbf{R}} p(s_T, s_{terminal}, r^i|\boldsymbol{h}_T, \boldsymbol{a}_T) \hat{\mathcal{R}}_i(s_T, \boldsymbol{a}_T, r^i, \epsilon_T^i) \Big) \tag{20c}$$

$$\overset{eq.17}{=} \sum_{\epsilon_T^i \in \mathcal{E}} p(\epsilon_T^i) \hat{Q}_{i,T}^{\pi}(\boldsymbol{h}_T, \boldsymbol{a}_T, \epsilon_T^i) \tag{20d}$$

where $s_{terminal}$ is the terminal state. Line 20a is due to the definition of Q-values in the last transitions and we do not use the discounted term. Line 20b is due to Equation 19 by replacing true rewards with the expectation of surrogate rewards. Line 20c holds since the summation over noisy term $\epsilon_T^i$ at line 20b is factored out over the summation. The equation between lines 20c and 20d holds due to Equation 17. Note we do not use the discounted term in 17 because $T$ is the terminal step.

**Induction step:** We now assume that $Q_{i,t+1}^{\pi}(\boldsymbol{h}_{t+1}, \boldsymbol{a}_{t+1}) = \sum_{\epsilon_{t+1}^i \in \mathcal{E}} p(\epsilon_{t+1}^i) \hat{Q}_{i,t+1}^{\pi}(\boldsymbol{h}_{t+1}, \boldsymbol{a}_{t+1}, \epsilon_{t+1}^i)$ holds for time step $t + 1 < T$, and prove that this holds also for time step $t$, i.e., we prove that $Q_{i,t}^{\pi}(\boldsymbol{h}_t, \boldsymbol{a}_t) = \sum_{\epsilon_t^i \in \mathcal{E}} p(\epsilon_t^i) \hat{Q}_{i,t}^{\pi}(\boldsymbol{h}_t, \boldsymbol{a}_t, \epsilon_t^i)$.

$$Q_{i,t}^{\pi}(\boldsymbol{h}_t, \boldsymbol{a}_t) \tag{21a}$$

$$= \sum_{s_t, s_{t+1} \in \mathcal{S}} p(s_t, s_{t+1}|\boldsymbol{h}_t, \boldsymbol{a}_t) \Big( \mathcal{R}_i(s_t, \boldsymbol{a}_t) + \gamma \sum_{\boldsymbol{a}_{t+1} \in \mathcal{A}, \boldsymbol{h}_{t+1} \in \boldsymbol{H}} p(\boldsymbol{h}_{t+1}|s_{t+1}) \boldsymbol{\pi}(\boldsymbol{a}_{t+1}|\boldsymbol{h}_{t+1}) \underbrace{Q_{i,t+1}^{\pi}(\boldsymbol{h}_{t+1}, \boldsymbol{a}_{t+1})}_{\overset{ind.step}{=} \sum_{\epsilon_{t+1}^i \in \mathcal{E}} p(\epsilon_{t+1}^i) \hat{Q}_{i,t+1}^{\pi}(\boldsymbol{h}_{t+1}, \boldsymbol{a}_{t+1}, \epsilon_{t+1}^i)}$$

$$\tag{21b}$$

$$\overset{eq.19}{=} \sum_{s_t, s_{t+1} \in \mathcal{S}} \sum_{r^i \in \mathbf{R}} p(s_t, s_{t+1}, r^i|\boldsymbol{h}_t, \boldsymbol{a}_t) \sum_{\epsilon_t^i \in \mathcal{E}} p(\epsilon_t^i) \Big( \hat{\mathcal{R}}_i(s_t, \boldsymbol{a}_t, r^i, \epsilon_t^i) + \tag{21c}$$

$$\gamma \sum_{\boldsymbol{a}_{t+1} \in \mathcal{A}, \boldsymbol{h}_{t+1} \in \boldsymbol{H}} p(\boldsymbol{h}_{t+1}|s_{t+1}) \boldsymbol{\pi}(\boldsymbol{a}_{t+1}|\boldsymbol{h}_{t+1}) \sum_{\epsilon_{t+1}^i \in \mathcal{E}} p(\epsilon_{t+1}^i) \hat{Q}_{i,t+1}^{\pi}(\boldsymbol{h}_{t+1}, \boldsymbol{a}_{t+1}, \epsilon_{t+1}^i) \Big) \tag{21d}$$

$$= \sum_{\epsilon_t^i \in \mathcal{E}} p(\epsilon_t^i) \Big( \sum_{s_t, s_{t+1} \in \mathcal{S}} \sum_{r^i \in \mathbf{R}} p(s_t, s_{t+1}, r^i|\boldsymbol{h}_t, \boldsymbol{a}_t) \Big( \hat{\mathcal{R}}_i(s_t, \boldsymbol{a}_t, r^i, \epsilon_t^i) + \tag{21e}$$

$$\gamma \sum_{\boldsymbol{a}_{t+1} \in \mathcal{A}, \boldsymbol{h}_{t+1} \in \boldsymbol{H}} p(\boldsymbol{h}_{t+1}|s_{t+1}) \boldsymbol{\pi}(\boldsymbol{a}_{t+1}|\boldsymbol{h}_{t+1}) \sum_{\epsilon_{t+1}^i \in \mathcal{E}} p(\epsilon_{t+1}^i) \hat{Q}_{i,t+1}^{\pi}(\boldsymbol{h}_{t+1}, \boldsymbol{a}_{t+1}, \epsilon_{t+1}^i) \Big) \Big) \tag{21f}$$

$$\overset{eq.17}{=} \sum_{\epsilon_t^i \in \mathcal{E}} p(\epsilon_t^i) \hat{Q}_{i,t}^{\pi}(\boldsymbol{h}_t, \boldsymbol{a}_t, \epsilon_t^i) \tag{21g}$$

where derivation 21b is due to the definition of $Q_{i,t}^\pi(\boldsymbol{h}_t, \boldsymbol{a}_t)$. Moreover, in derivation 21b, we use the assumption $Q_{i,t+1}^\pi(\boldsymbol{h}_{t+1}, \boldsymbol{a}_{t+1}) = \sum_{\epsilon_{t+1}^i \in \mathcal{E}} p(\epsilon_{t+1}^i) \hat{Q}_{i,t+1}^\pi(\boldsymbol{h}_{t+1}, \boldsymbol{a}_{t+1}, \epsilon_{t+1}^i)$, leading to the discounted term in 21d. We also use Equation 19 to replace true rewards with the expectation of surrogate rewards in derivation 21c. Derivations 21e-21f holds since the summation over noisy term $\epsilon_t^i$ at derivation 21c is factored out over the summation. The equation between derivations 21e-21f and 21g holds due to Equation 17 (the surrogate Q-function). Then, we achieve the equality $Q_{i,t}^\pi(\boldsymbol{h}_t, \boldsymbol{a}_t) = \sum_{\epsilon_t^i \in \mathcal{E}} p(\epsilon_t^i) \hat{Q}_{i,t}^\pi(\boldsymbol{h}_t, \boldsymbol{a}_t, \epsilon_t^i)$. By induction, we conclude that $Q_i^\pi(\boldsymbol{h}, \boldsymbol{a}) = \sum_{\epsilon_i \in \mathcal{E}} p(\epsilon_i) \hat{Q}_i^\pi(\boldsymbol{h}, \boldsymbol{a}, \epsilon_i)$ hold for any time step (and we drop the time step). Due to shared rewards and the definition of expectation we have $Q_i^\pi(\boldsymbol{h}, \boldsymbol{a}) = Q^\pi(\boldsymbol{h}, \boldsymbol{a})$ and $\sum_{\epsilon_i \in \mathcal{E}} p(\epsilon^i) \hat{Q}_i^\pi(\boldsymbol{h}, \boldsymbol{a}, \epsilon^i) = \mathbb{E}_{\epsilon_i}[\hat{Q}_i^\pi(\boldsymbol{h}, \boldsymbol{a}, \epsilon_i)]$. Therefore, we complete the proof that $Q^\pi(\boldsymbol{h}, \boldsymbol{a}) = \mathbb{E}_{\epsilon_i}[\hat{Q}_i^\pi(\boldsymbol{h}, \boldsymbol{a}, \epsilon_i)]$.

### B.1  Proof of Theorem 2

We use Q-functions $\hat{Q}_i^\pi(\boldsymbol{h}, \boldsymbol{a}, \epsilon_i)$ and $Q^\pi(\boldsymbol{h}, \boldsymbol{a})$ as critics for decentralized actors $\pi_i(a_i|h_i, \theta_i)$. Then, we have the following policy gradients:

$$g_{DCCDA-noise}^i = \mathbb{E}_{\boldsymbol{h}, \boldsymbol{a}, \epsilon_i}[\hat{Q}_i^\pi(\boldsymbol{h}, \boldsymbol{a}, \epsilon_i) \nabla_{\theta_i} \log \pi_i(a_i|h_i, \theta_i)]$$

$$g_{CTDE}^i = \mathbb{E}_{\boldsymbol{h}, \boldsymbol{a}}[Q^\pi(\boldsymbol{h}, \boldsymbol{a}) \nabla_{\theta_i} \log \pi_i(a_i|h_i, \theta_i)]$$

where $g_{DCCDA-noise}^i$ is the noise version of DCCDA and the noise in communication is lifted to a surrogate Q-function.

Based on $g_{DCCDA-noise}^i$ and $g_{CTDE}^i$, we have the following theorem, which follows similar procedures as in the proof of idealistic communication setting:

**Theorem 2.** *The noisy version of DCCDA sample gradient has a variance greater or equal than that of the CTDE sample gradient in non-idealistic communication setting:* $Var(\hat{g}_{DCCDA-noise}^i) \geq Var(\hat{g}_{CTDE}^i)$.

*Proof of Theorem 2.* In the following proof, we first check the relation between the two expected gradients $g_{DCCDA-noise}^i$ and $g_{CTDE}^i$, which can greatly simplify the later variance analysis. Then we compare the variance of gradients $g_{DCCDA-noise}^i$ and $g_{CTDE}^i$.

Based on Lemma 3, as $Q^\pi(\boldsymbol{h}, \boldsymbol{a}) = \mathbb{E}_{\epsilon_i}[Q_i^\pi(\boldsymbol{h}, \boldsymbol{a}, \epsilon_i)]$:

$$g_{DCCDA-noise}^i = \mathbb{E}_{\boldsymbol{h}, \boldsymbol{a}, \epsilon_i}[Q_i^\pi(\boldsymbol{h}, \boldsymbol{a}, \epsilon_i) \nabla_{\theta_i} \log \pi_i(a_i|h_i, \theta_i)] \tag{22a}$$

$$= \mathbb{E}_{\boldsymbol{h}, \boldsymbol{a}}[\mathbb{E}_{\epsilon_i}[Q_i^\pi(\boldsymbol{h}, \boldsymbol{a}, \epsilon_i) \nabla_{\theta_i} \log \pi_i(a_i|h_i, \theta_i)] \tag{22b}$$

$$\overset{lem.3}{=} \mathbb{E}_{\boldsymbol{h}, \boldsymbol{a}}[Q^\pi(\boldsymbol{h}, \boldsymbol{a}) \nabla_{\theta_i} \log \pi_i(a_i|h_i, \theta_i)] \tag{22c}$$

$$= g_{CTDE}^i \tag{22d}$$

where line 22c is due to Lemma 3. Note that the noise term is not included in actors as actors do not communicate.

Based on derivation 22a-22d, we come to the following proof of Theorem 2. Note that $\hat{g}_{CTDE}^i$ is used to denote $Q^\pi(\boldsymbol{h}, \boldsymbol{a}) \nabla_{\theta_i} \log \pi_i(a_i|h_i, \theta_i)$ and $\hat{g}_{DCCDA-noise}^i$ is used to denote $Q_i^\pi(\boldsymbol{h}, \boldsymbol{a}, \epsilon_i) \nabla_{\theta_i} \log \pi_i(a_i|h_i, \theta_i)$. Similar to the variance analysis in Appendix A.1, in order to simplify the notation, we denote $S = \nabla_{\theta_i} \log \pi_i(a_i|h_i, \theta_i)^T \nabla_{\theta_i} \log \pi_i(a_i|h_i, \theta_i)$, which is the inner product of $\nabla_{\theta_i} \log \pi_i(a_i|h_i, \theta_i)$ (due to the square of it):

$$Var(\hat{g}^i_{DCCDA-noise}) - Var(\hat{g}^i_{CTDE}) \tag{23a}$$

$$\stackrel{def.}{=} \left( \mathbb{E}_{\boldsymbol{h},\boldsymbol{a},\epsilon_i}[(\hat{g}^i_{DCCDA-noise})^2] - \left( \mathbb{E}_{\boldsymbol{h},\boldsymbol{a},\epsilon_i}[\hat{g}^i_{DCCDA-noise}] \right)^2 \right) - \left( \mathbb{E}_{\boldsymbol{h},\boldsymbol{a}}[(\hat{g}^i_{CTDE})^2] - \left( \mathbb{E}_{\boldsymbol{h},\boldsymbol{a}}[\hat{g}^i_{CTDE}] \right)^2 \right) \tag{23b}$$

$$= \left( \mathbb{E}_{\boldsymbol{h},\boldsymbol{a},\epsilon_i}[(\hat{g}^i_{DCCDA-noise})^2] - \mathbb{E}_{\boldsymbol{h},\boldsymbol{a}}[(\hat{g}^i_{CTDE})^2] \right) - \left( \left( \mathbb{E}_{\boldsymbol{h},\boldsymbol{a},\epsilon_i}[\hat{g}^i_{DCCDA-noise}] \right)^2 - \left( \mathbb{E}_{\boldsymbol{h},\boldsymbol{a}}[\hat{g}^i_{CTDE}] \right)^2 \right) \tag{23c}$$

$$\stackrel{eq.22}{=} \left( \mathbb{E}_{\boldsymbol{h},\boldsymbol{a},\epsilon_i}[(\hat{g}^i_{DCCDA-noise})^2] - \mathbb{E}_{\boldsymbol{h},\boldsymbol{a}}[(\hat{g}^i_{CTDE})^2] \right) - \underbrace{\left( (g^i_{DCCDA-noise})^2 - (g^i_{CTDE})^2 \right)}_{=0} \tag{23d}$$

$$= \mathbb{E}_{\boldsymbol{h},\boldsymbol{a},\epsilon_i}[(\hat{g}^i_{DCCDA-noise})^2] - \mathbb{E}_{\boldsymbol{h},\boldsymbol{a}}[(\hat{g}^i_{CTDE})^2] \tag{23e}$$

$$\stackrel{def.}{=} \mathbb{E}_{\boldsymbol{h},\boldsymbol{a},\epsilon_i}[(Q^\pi_i(\boldsymbol{h},\boldsymbol{a},\epsilon_i)\nabla_{\theta_i}\log\pi_i(a_i|h_i,\theta_i))^2] - \mathbb{E}_{\boldsymbol{h},\boldsymbol{a}}[(Q^\pi(\boldsymbol{h},\boldsymbol{a})\nabla_{\theta_i}\log\pi_i(a_i|h_i,\theta_i))^2] \tag{23f}$$

$$= \mathbb{E}_{\boldsymbol{h},\boldsymbol{a},\epsilon_i}[Q^\pi_i(\boldsymbol{h},\boldsymbol{a},\epsilon_i)^2 S] - \mathbb{E}_{\boldsymbol{h},\boldsymbol{a}}[Q^\pi(\boldsymbol{h},\boldsymbol{a})^2 S] \tag{23g}$$

$$= \mathbb{E}_{\boldsymbol{h},\boldsymbol{a}}[\mathbb{E}_{\epsilon_i}[Q^\pi_i(\boldsymbol{h},\boldsymbol{a},\epsilon_i)^2 S]] - \mathbb{E}_{\boldsymbol{h},\boldsymbol{a}}[Q^\pi(\boldsymbol{h},\boldsymbol{a})^2 S] \tag{23h}$$

$$= \mathbb{E}_{\boldsymbol{h},\boldsymbol{a}}[\mathbb{E}_{\epsilon_i}[Q^\pi_i(\boldsymbol{h},\boldsymbol{a},\epsilon_i)^2 S] - Q^\pi(\boldsymbol{h},\boldsymbol{a})^2 S] \tag{23i}$$

$$\stackrel{lem.3}{=} \mathbb{E}_{\boldsymbol{h},\boldsymbol{a}}[\underbrace{\left( \mathbb{E}_{\epsilon_i}[Q^\pi_i(\boldsymbol{h},\boldsymbol{a},\epsilon_i)^2] - (\mathbb{E}_{\epsilon_i}[Q^\pi_i(\boldsymbol{h},\boldsymbol{a},\epsilon_i)])^2 \right)}_{u} S] \tag{23j}$$

$$\geq 0 \tag{23k}$$

where line 23b follows the definition of variance. Line 23d is due to $g^i_{DCCDA-noise} = g^i_{CTDE}$ according to derivation 22a-22d. Line 23j is due to Lemma 3. The final inequality in line 23k follows because $u \geq 0$ by Jensen's inequality: $\mathbb{E}[X^2] \geq (\mathbb{E}[X])^2$, and $S$ is the inner product of a vector itself and therefore non-negative. Therefore, the noise version of DCCDA sample gradient has a variance greater or equal than that of the CTDE sample gradient, i.e., $Var(\hat{g}^i_{DCCDA-noise}) \geq Var(\hat{g}^i_{CTDE})$, which completes the proof.

## C  Proofs of the Theoretical Results of the Optimal Baseline

### C.1  Proof of Theorem 3

In this section, we derive the optimal message-dependent baseline. The computation of the message-dependent baseline use each agent's critic as well as encoded messages, where messages can either be noise-free or noisy. Therefore, the message-dependent baseline can be used in both idealistic or non-idealistic communication setting. We have the following theorem:

**Theorem 3.** *The optimal message-dependent baseline for DCCDA-OB gradient estimator is,*

$$b^*_i(h_i, m_{-i}) = \frac{\mathbb{E}_{a_i}[Q_i(h_i, a_i, m_{-i})S]}{\mathbb{E}_{a_i}[S]} \tag{24}$$

*where* $S = \nabla_{\theta_i}\log\pi_i(a_i|h_i,\theta_i)^T \nabla_{\theta_i}\log\pi_i(a_i|h_i,\theta_i)$.

*Proof of Theorem 3.* To prove the theorem, we firstly prove that the message-dependent baseline does not change the policy gradients $g^i_{DCCDA}$ (i.e., $g^i_{DCCDA-OB} = g^i_{DCCDA}$), which will be used to simplify the variance measurement of $g^i_{DCCDA-OB}$. Note that the noisy version of the DCDDA policy gradients, $g^i_{DCCDA-noise}$, also applies to the following derivations, where we can replace $m_{-i}$ with $< h_{-i}, a_{-i}, \epsilon_i >$.

Therefore, we have,

$$\mathbb{E}_{\boldsymbol{h},\boldsymbol{a},\boldsymbol{m}}[b_i(h_i, m_{-i})\nabla_{\theta_i}\log\pi_i(a_i|h_i,\theta_i)] = \mathbb{E}_{\boldsymbol{h},a_{-i},\boldsymbol{m}}[b_i(h_i, m_{-i})\mathbb{E}_{a_i}[\nabla_{\theta_i}\log\pi_i(a_i|h_i,\theta_i)]]$$

$$= \mathbb{E}_{\boldsymbol{h},a_{-i},\boldsymbol{m}}[b_i(h_i, m_{-i})\sum_{a_i}\pi_i(a_i|h_i,\theta_i)\nabla_{\theta_i}\log\pi_i(a_i|h_i,\theta_i)]$$

$$= \mathbb{E}_{\boldsymbol{h},a_{-i},\boldsymbol{m}}[b_i(h_i, m_{-i})\sum_{a_i}\cancel{\pi_i(a_i|h_i,\theta_i)}\frac{\nabla_{\theta_i}\pi_i(a_i|h_i,\theta_i)}{\cancel{\pi_i(a_i|h_i,\theta_i)}}] \tag{25}$$

$$= \mathbb{E}_{\boldsymbol{h},a_{-i},\boldsymbol{m}}[b_i(h_i, m_{-i})\sum_{a_i}\nabla_{\theta_i}\pi_i(a_i|h_i,\theta_i)] = \mathbb{E}_{\boldsymbol{h},a_{-i},\boldsymbol{m}}[b_i(h_i, m_{-i})\nabla_{\theta_i}1] = 0$$

where the second to the last line is due to the sum of all probabilities over agent $i$'s action is 1. By integrating Equation 25 into the policy gradient $g^i_{DCCDA-OB} = \mathbb{E}_{\boldsymbol{h},\boldsymbol{a},\boldsymbol{m}}[(Q_i(h_i, a_i, m_{-i}) - b_i(h_i, m_{-i}))\nabla_{\theta_i} \log \pi_i(a_i|h_i, \theta_i)]$, we obtain the policy gradient $g^i_{DCCDA} = \mathbb{E}_{\boldsymbol{h},\boldsymbol{a},\boldsymbol{m}}[Q^\pi_i(h_i, a_i, m_{-i})\nabla_{\theta_i} \log \pi_i(a_i|h_i, \theta_i)]$. Therefore, we have $g^i_{DCCDA-OB} = g^i_{DCCDA}$. The equality shows that the baseline $b_i(h_i, m_{-i})$ does not introduce bias to $g^i_{DCCDA}$ in expectation. Nevertheless, $g^i_{DCCDA}$ and $g^i_{DCCDA-OB}$ may have different variance properties. We first derive the variance of the DCCDA policy gradient estimate with the message-dependent baseline. We simplify the expression by using $b_i$ to denote the baseline $b_i(h_i, m_{-i})$. Therefore, by the definition of variance, we have:

$$Var(\hat{g}^i_{DCCDA-OB}) = \mathbb{E}_{\boldsymbol{h},\boldsymbol{a},\boldsymbol{m}}[(\hat{g}^i_{DCCDA-OB})^2] - (\mathbb{E}_{\boldsymbol{h},\boldsymbol{a},\boldsymbol{m}}[\hat{g}^i_{DCCDA-OB}])^2$$

$$= \mathbb{E}_{\boldsymbol{h},\boldsymbol{a},\boldsymbol{m}}[((Q_i(h_i, a_i, m_{-i}) - b_i)\nabla_{\theta_i} \log \pi_i(a_i|h_i, \theta_i))^2] - (\mathbb{E}_{\boldsymbol{h},\boldsymbol{a},\boldsymbol{m}}[(Q_i(h_i, a_i, m_{-i}) - b_i)\nabla_{\theta_i} \log \pi_i(a_i|h_i, \theta_i)])^2$$

$$= \mathbb{E}_{\boldsymbol{h},\boldsymbol{a},\boldsymbol{m}}[((Q_i(h_i, a_i, m_{-i}) - b_i))\nabla_{\theta_i} \log \pi_i(a_i|h_i, \theta_i))^2] - (\mathbb{E}_{\boldsymbol{h},\boldsymbol{a},\boldsymbol{m}}[Q_i(h_i, a_i, m_{-i})\nabla_{\theta_i} \log \pi_i(a_i|h_i, \theta_i)])^2$$

(26)

where line 2 follows the definition of $\hat{g}^i_{DCCDA-OB}$ and the last line is due to the fact $\mathbb{E}_{\boldsymbol{h},\boldsymbol{a},\boldsymbol{m}}[b_i(h_i, m_{-i})\nabla_{\theta_i} \log \pi_i(a_i|h_i, \theta_i)]$ is zero according to Equation 25. Note that $S = \nabla_{\theta_i} \log \pi_i(a_i|h_i, \theta_i)^T \nabla_{\theta_i} \log \pi_i(a_i|h_i, \theta_i)$ is used to denote the inner product of the gradient $\nabla_{\theta_i} \log \pi_i(a_i|h_i, \theta_i)$ for notation simplification. We seek the optimal baseline that would minimize this variance by setting the derivatives with respect to the baseline $b_i$ to be zero:

$$\frac{\partial}{\partial b_i}\left[Var(\hat{g}^i_{DCCDA-OB})\right]$$

$$= \frac{\partial}{\partial b_i}\left[\mathbb{E}_{\boldsymbol{h},\boldsymbol{a},\boldsymbol{m}}[(Q_i(h_i, a_i, m_{-i}) - b_i)^2 S]\right] + \frac{\partial}{\partial b_i}\left[(\mathbb{E}_{\boldsymbol{h},\boldsymbol{a},\boldsymbol{m}}[Q_i(h_i, a_i, m_{-i})\nabla_{\theta_i} \log \pi_i(a_i|h_i, \theta_i)])^2\right]$$

$$= \frac{\partial}{\partial b_i}\left[\mathbb{E}_{\boldsymbol{h},\boldsymbol{a},\boldsymbol{m}}[(Q_i(h_i, a_i, m_{-i}) - b_i)^2 S]\right]$$

$$= 0$$

(27)

where the term in the second line does not depend on the baseline $b_i$, and therefore the derivative is 0. By writing out the term in brackets from the second to the last line in Equation 27, we have:

$$\mathbb{E}_{\boldsymbol{h},\boldsymbol{a},\boldsymbol{m}}[(Q_i(h_i, a_i, m_{-i}) - b_i)^2 S]$$

$$= \mathbb{E}_{\boldsymbol{h},\boldsymbol{a},\boldsymbol{m}}[(Q_i(h_i, a_i, m_{-i})^2 - 2b_iQ_i(h_i, a_i, m_{-i}) + b_i^2) S]$$

$$= \mathbb{E}_{\boldsymbol{h},\boldsymbol{a},\boldsymbol{m}}[Q_i(h_i, a_i, m_{-i})^2 S - 2b_iQ_i(h_i, a_i, m_{-i})S + b_i^2 S]$$

$$= \mathbb{E}_{\boldsymbol{h},\boldsymbol{a},\boldsymbol{m}}[Q_i(h_i, a_i, m_{-i})^2 S] + \mathbb{E}_{\boldsymbol{h},\boldsymbol{a}_{-i},\boldsymbol{m}}[-2b_i\mathbb{E}_{a_i}[Q_i(h_i, a_i, m_{-i})S] + b_i^2\mathbb{E}_{a_i}[S]]$$

(28)

where the last line is because $b_i$ does not depend on actions $a_i$. By integrating Equation 28 into Equation 27 we have:

$$\frac{\partial}{\partial b_i}\left[Var(\hat{g}^i_{DCCDA-OB})\right] = \mathbb{E}_{\boldsymbol{h},\boldsymbol{a}_{-i},\boldsymbol{m}}[-2\mathbb{E}_{a_i}[Q_i(h_i, a_i, m_{-i})S] + 2b_i(h_i, m_{-i})\mathbb{E}_{a_i}[S]] = 0 \qquad (29)$$

Therefore, the optimal baseline is,

$$b_i^*(h_i, m_{-i}) = \frac{\mathbb{E}_{a_i}[Q_i(h_i, a_i, m_{-i})S]}{\mathbb{E}_{a_i}[S]} \qquad (30)$$

where the expectation enumerates all possible actions of agent $i$. Then, we complete the proof.

## C.2 Proof of Corollary 1

In this section, we prove that with the optimal message-dependent baseline, the variance of DCCDA policy gradient is reduced. Note that Corollary 1 also holds for non-idealistic communication setting, i.e., $Var(\hat{g}^i_{DCCDA-OB}) \leq Var(\hat{g}^i_{DCCDA-noise})$, where we replace message $m_{-i}$ with $<h_{-i}, a_{-i}, \epsilon_i>$ and follow the same derivations.

We have the following corollary based on Theorem 3:

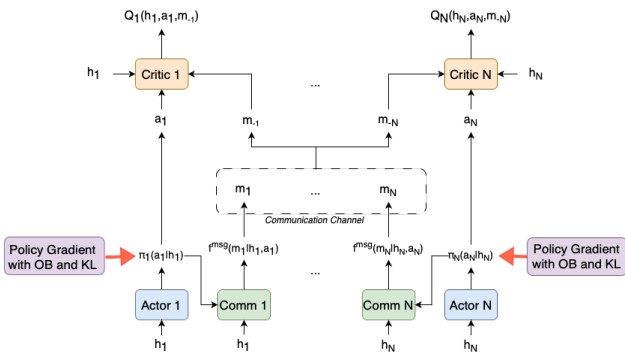

Figure 4: The diagram of DCCDA methods integrated with OB and KL.

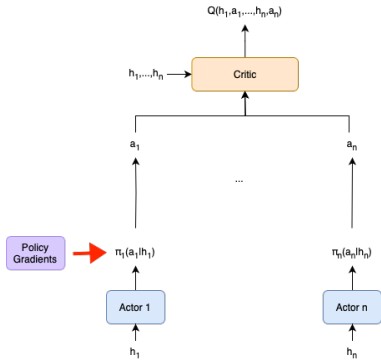

Figure 5: The diagram of CTDE method MAPPO.

**Corollary 1.** *The variance of DCCDA policy gradients is reduced with the optimal message-dependent baseline:* $Var(\hat{g}^i_{DCCDA-OB}) \le Var(\hat{g}^i_{DCCDA})$.

*Proof of Corollary 1.* The proof is achieved by integrating the optimal baseline $b^*_i$ derived from Theorem 3 back to Equation 26, where $Var(\hat{g}^i_{DCCDA}) = \mathbb{E}_{\boldsymbol{h},\boldsymbol{a},\boldsymbol{m}}[(Q_i(h_i, a_i, m_{-i})\nabla_{\theta_i} \log \pi_i(a_i|h_i, \theta_i))^2] - (\mathbb{E}_{\boldsymbol{h},\boldsymbol{a},\boldsymbol{m}}[Q_i(h_i, a_i, m_{-i})\nabla_{\theta_i} \log \pi_i(a_i|h_i, \theta_i)])^2$ by definition. Then we have:

$$Var(\hat{g}^i_{DCCDA-OB}) = Var(\hat{g}^i_{DCCDA}) - \mathbb{E}_{\boldsymbol{h},a_{-i},\boldsymbol{m}}\Big[\frac{(\mathbb{E}_{a_i}[Q_i(h_i, a_i, m_{-i})S])^2}{\mathbb{E}_{a_i}[S]}\Big] \tag{31}$$

where the second term on the right in the Equation is non-negative. Therefore, we have: $Var(\hat{g}^i_{DCCDA-OB}) \le Var(\hat{g}^i_{DCCDA})$, which completes the proof.

# D  Settings, Implementations, Algorithms, Parameters, and Additional Results

## D.1  Comparison between DCCDA and CTDE

DCCDA is fundamentally different from CTDE as well as the method MADDPG. In DCCDA, messages are learned without being predefined, essentially based on the literature on learning communication in multi-agent deep reinforcement learning (Comm-MADRL). In CTDE, the centralized critic needs to fully access all agents' partial information, including their histories and actions, i.e., $Q(h_1, \ldots h_N, a_1, \ldots, a_N)$. Similarly, in MADDPG, each agent holds a joint Q-function as $Q_i(h_1, \ldots h_N, a_1, \ldots, a_N)$, which also needs to access all agents' information. In comparison, in DCCDA, the partial information of agents is encoded as a low-dimensional vector (as messages), i.e., $Q(h_i, a_i, m_{-i})$, where $m_{-i} = \{m_1, \ldots, m_{i-1}, m_{i+1}, m_N\}$ is the received messages, which can significantly reduce the input dimensionality compared to the centralized Q-function. In order to show the differences in architecture between DCCDA and CTDE methods, we compare

the diagram for DCCDA (See Figure 4) and the diagram for CTDE (See Figure 5) to show the differences in their (forward) structure of neural networks. Specifically, in the diagrams, we use IPPO-Comm-OB-KL (the DCCDA method) and MAPPO (the CTDE method) as examples to show how we train the actors, critics, and the extended communication module. Note that IPPO-Comm-OB-KL and MAPPO differ only in their critics (and communication) and the way to train the critics (and communication). It will be easier to identify the differences between IPPO-Comm-OB-KL and MAPPO. As shown in the two diagrams, the red arrows refer to the training phase. In IPPO-Comm-OB-KL, the actor modules, communication (comm) modules, and critic modules are the neural networks. Compared to MAPPO, IPPO-Comm-OB-KL involves a message-generation process using the actor and comm modules and the communication channel. DCCDA agents need to encode, select, and transmit their information.

## D.2 Algorithms and Implementation Details

We present two cases demonstrating the extension of two DCCDA algorithms using our proposed techniques. Firstly, we select the state-of-the-art method under the DCCDA setting focusing on actor-critic methods, GAAC (Liu et al., 2020), and illustrate the extension, GAAC-OB-KL, in Algorithm 1. Due to the absence of learning communication methods under the DCCDA setting, we opt for the state-of-the-art actor-critic method under the DTDE setting (without communication), IPPO (Yu et al., 2022), and introduce a standard communication architecture, resulting in IPPO-Comm. Subsequently, IPPO-Comm is further extended with our proposed techniques, forming IPPO-Comm-OB-KL, which is illustrated in Algorithm 2.

**Algorithm Descriptions**. In Algorithm 1, each agent's actor, critic, and communication model are initialized first. During each training iteration, agents communicate through the communication process introduced by GAAC. We utilize the implementation of GAAC from a publicly accessible repository.[4] In the algorithm, agents store observations, actions, messages, rewards, new observations, Q-values, and action probabilities in their buffer for training including computing the value of the message-dependent baseline and the KL divergence term. When training is enabled, agents train their policies using the gradients defined in $g^i_{DCCDA-OB-KL} = g^i_{DCCDA-OB} + \beta \nabla_{\theta_i} \mathcal{L}_{KL}(\theta_i)$. Critics are trained in a DQN-like manner, and the communication model is trained according to GAAC. Specifically, communication in GAAC is trained via backpropagation, allowing gradients to flow from the critics to both the actors and the communication modules. We use the implementation of IPPO from a publicly accessible repository.[5] In IPPO-Comm-OB-KL (and IPPO-Comm), we integrate a communication architecture that considers the historical information and current actions of sender agents in messages, which is in line with our Assumption 1 to encode all available local information into messages. Specifically, we have explored various methods to generate messages. Ultimately, we chose the most effective approach: concatenating the policy distribution (which probabilistically indicates actions) along with the history/local information (hidden states from the LSTM when using histories as input) of the sender agents. Next, the concatenation is encoded through an MLP, and the output layer uses softmax to produce a distribution of messages. These messages are then selected and sent to receiver agents. The communication module in IPPO-Comm-OB-KL and IPPO-Comm is trained through policy gradients using Q-values rather than backpropagation. The policy gradients of communication module is defined as: $g^i_{msg} = \mathbb{E}_{\boldsymbol{h}, \boldsymbol{a}, \boldsymbol{m}}[\nabla_{\theta^{msg}_i} \log f^{msg}_i(m_i|h_i, a_i, \theta^{msg}_i) Q_j(h_j, a_j, m_{-j})]$, where $Q_j(h_j, a_j, m_{-j})$ is receiver agent's Q-values ($j \neq i$). $g^i_{msg}$ is then used to train each agent's message function $f^{msg}_i$. As a result, IPPO-Comm-OB-KL and GAAC-OB-KL differ only in how agents communicate and train their communication models.

**The use of messages**. In StarCraft and Traffic Junction, messages are a vector of discrete values generated by a probabilistic message function using a softmax output layer. Stochasticity in these environments can arise from multiple sources. For instance, in both scenarios, agents observe the relative positions of other agents within their vision range, which can vary stochastically due to the movement of other agents or enemies. In Traffic Junction specifically, new cars are added to the environment at each timestep with a certain probability. Additionally, both tasks involve learning a stochastic policy. These characteristics introduce randomness in observations, transitions, and policies. When agents communicate, the use of a probabilistic message function allows them to capture and reflect this inherent stochasticity present in the environment.

---

[4]The open-source code is available at https://github.com/starry-sky6688/MARL-Algorithms.
[5]The open-source code is available at https://github.com/marlbenchmark/on-policy.

---

**Algorithm 1** GAAC-OB-KL using regularized policies and message-dependent baselines

---

1: Initialize $\theta_i$ and $\phi_i$, the parameters for each agent's actor and critic. Initialize the communication model for each agent.
2: **for** each training iteration **do**
3:     Initialize data buffer $D_i$ for each agent $i$
4:     Get initial observations $\boldsymbol{o}_0 = \{o_0^1, ..., o_0^i, ..., o_0^N\}$ and set initial history $\boldsymbol{h}_0$
5:     **for** $t = 0$ to *max_steps_per_episode* **do**
6:         **for** each agent $i$ **do**
7:             Decide an action $a_t^i$ and output the corresponding probability distribution: $p_t^i \leftarrow \pi_i(a_t^i|h_t^i, \theta_i)$
8:             Generate messages $m_t^i$ from encoded observations and actions
9:             Send messages to other agents
10:             Aggregate received messages through a two-layer attention mechanism
11:             Generate the corresponding Q-value: $q_t^i \leftarrow Q_i(h_t^i, a_t^i, m_t^{-i})$
12:         **end for**
13:         Get new observations $\boldsymbol{o}_{t+1}$ and rewards $r_t$ by performing actions
14:         Insert experience $(o_t^i, m_t^{-i}, a_t^i, r_t, o_{t+1}^i, q_t^i, p_t^i)$ into $D_i$ and update $h_t^i$ for each agent $i$
15:     **end for**
16:     **for** each agent $i$ **do**
17:         Sample a train batch $b_i$ from buffer $D_i$
18:         Calculate the KL objective $\mathcal{L}_{KL}(\theta_i)$ using sampled experience
19:         Compute the message-dependent baseline $b_i^*(h_i, m_{-i})$ using sampled experience
20:         Update $\theta_i$ with Adam/RMSProp following the gradient defined in $g_{DCCDA-OB-KL}^i$
21:     **end for**
22:     Update the communication model evaluated by $Q_i(h_t^i, a_t^i, m_t^{-i})$ for each agent $i$
23:     Update the critic parameter $\phi_i$ for each agent $i$ with TD-learning
24: **end for**

---

### D.3 Practical usage of the proposed techniques

**The Effect of Value Estimation**. The KL divergence term discourages the actor from selecting actions to which the critic assigns a low value. This may constrain the exploration of the actor, especially when the critic is learning and non-optimal. In practice, we think the negative influence needs to be removed when critics get stuck in a local optimum, which will always push actors back to the local optimum. The baseline values depend on Q-value estimates. When using mini-batch baseline estimation in an iterative training fashion, the baseline values can vary across mini-batches even for the same agent history and action, since the sampled Q-values may originate from different iterations. However, in our case, we use on-policy algorithms (e.g., IPPO), where experience (i.e., mini-batches) is refreshed after each training iteration (at the end of an episode). Consequently, baseline values are computed using the latest Q-value estimates and remain fixed within each iteration. As a result, the policy gradients remain unbiased. Nonetheless, the exact amount of variance reduction still depends on the baseline estimation, which may lead to either greater or lesser variance reduction.

**Implementations of the Proposed Techniques**. Recall that in the proposed baseline technique, we use $S = \nabla_{\theta_i} \log \pi_i(a_i|h_i, \theta_i)^T \nabla_{\theta_i} \log \pi_i(a_i|h_i, \theta_i)$ to denote the inner production of the gradient $\nabla_{\theta_i} \log \pi_i(a_i|h_i, \theta_i)$. This can be computationally challenging due to the extremely high dimension of the parameter space (parametrized by a neural network). Inspired by the work of Kuba et al. (2021), we use the softmax policy, which allows the product to be computed in an analytical form. We further leverage the implementation of the inner product from Kuba et al. (2021) to build our message-dependent baseline, explicitly incorporating messages in the replay buffer and during the computation of the baseline values. Furthermore, in the KL divergence term, we estimate the KL divergence by first computing the Boltzmann policy using local Q-values over mini-batches. The estimated KL divergence is then calculated and averaged across the batch. During backpropagation, gradients from the KL divergence term are propagated to the policies, introducing an additional influence on the policy gradients.

---

**Algorithm 2** IPPO-Comm-OB-KL using regularized policies and message-dependent baselines

---

1: Initialize $\theta_i$ and $\phi_i$, the parameters for each agent's actor and critic. Initialize the communication model for each agent.
2: **for** each training iteration **do**
3:     Initialize data buffer $D_i$ for each agent $i$
4:     Get initial observations $\boldsymbol{o}_0 = \{o_0^1, ..., o_0^i, ..., o_0^N\}$ and set initial history $\boldsymbol{h}_0$
5:     **for** $t = 0$ to *max_steps_per_episode* **do**
6:         **for** each agent $i$ **do**
7:             Decide an action $a_t^i$ and output the corresponding probability distribution: $p_t^i \leftarrow \pi_i(a_t^i | h_t^i, \theta_i)$
8:             Generate messages $m_t^i$ from individual history and policy distribution
9:             Send messages to other agents
10:            Generate the corresponding Q-value: $q_t^i \leftarrow Q_i(h_t^i, a_t^i, m_t^{-i})$
11:         **end for**
12:     Get new observations $\boldsymbol{o}_{t+1}$ and rewards $r_t$ by performing actions
13:     Insert experience $(o_t^i, m_t^{-i}, a_t^i, r_t, o_{t+1}^i, q_t^i, p_t^i)$ into $D_i$ and update $h_t^i$ for each agent $i$
14:     **end for**
15:     **for** each agent $i$ **do**
16:         Sample a train batch $b_i$ from buffer $D_i$
17:         Calculate the KL objective $\mathcal{L}_{KL}(\theta_i)$ using sampled experience
18:         Compute the message-dependent baseline $b_i^*(h_i, m_{-i})$ using sampled experience
19:         Update $\theta_i$ with Adam/RMSProp following the gradient defined in $g_{DCCDA-OB-KL}^i$
20:     **end for**
21:     Update the communication model evaluated by $Q_i(h_t^i, a_t^i, m_t^{-i})$ for each agent $i$
22:     Update the critic parameter $\phi_i$ for each agent $i$ with TD-learning
23: **end for**

---

Table 2: The essential components of all methods.

| methods | settings | critics | policy regularization |
|---------|----------|---------|----------------------|
| COMA | CTDE | $Q(s, \boldsymbol{a}) - \mathbb{E}_{a_i}[Q(s, a_i, a_{-i})]$ | no |
| MAPPO | CTDE | $Q(s, \boldsymbol{a}) - V(s)$ | entropy |
| MAT | CTDE | $Q(s, \boldsymbol{a}) - V(s)$ | entropy |
| IPPO-Comm | DCCDA | $Q_i(h_i, a_i, m_{-i}) - V_i(h_i)$ | entropy |
| IPPO-Comm-OB-KL | DCCDA | $Q_i(h_i, a_i, m_{-i}) - b_i(h_i, m_{-i})$ (Eq. 1) | KL (Eq. 3) |
| GAAC | DCCDA | $Q_i(h_i, a_i, m_{-i})$ | no |
| GAAC-OB-KL | DCCDA | $Q_i(h_i, a_i, m_{-i}) - b_i(h_i, m_{-i})$ (Eq. 1) | KL (Eq. 3) |

## D.4 Comparison in Baseline Methods

We illustrate the essential components of all methods and how they differ from each other in MADRL setting, critics and policy regularization techniques in Table 2. Notably, IPPO-Comm-OB-KL and GAAC-OB-KL inherently differ other methods due to the proposed message-dependent baseline and the regularization concerning communication. COMA, MAPPO, and MAT use baseline techniques based on state-value or action-value functions that do not account for encoded messages. The communication method GAAC does not employ a baseline technique. IPPO-Comm, on the other hand, follows the same training strategies as IPPO, which includes a baseline based on state-value functions.

## D.5 Statistical Tests

In Table 5, we report the median win rate and standard deviation on all evaluated methods in SMAC and Traffic Junction. We also report the mean and 95% confidence interval of all methods in the last 100

Table 3: Important hyperparameters in SMAC and Traffic Junction.

| HYPERPARAMETERS | SMAC | | TRAFFIC JUNCTION | |
| --- | --- | --- | --- | --- |
| | IPPO-COMM-OB-KL | GAAC-OB-KL | IPPO-COMM-OB-KL | GAAC-OB-KL |
| *ACTOR LR* | 5E-4 | 5E-4 | 1E-3 | 1E-3 |
| *CRITIC LR* | 5E-4 | 5E-4 | 1E-2 | 1E-2 |
| *COMM LR* | 5E-4 | 5E-4 | 1E-3 | 1E-3 |
| *GAMMA* | 0.99 | 0.99 | 0.99 | 0.99 |
| *UPDATE EPOCH* | 10 | 10 | 10 | 10 |
| *MINI BATCH* | 1 | 1 | 1 | 1 |
| *OPTIMIZER* | ADAM | ADAM | ADAM | ADAM |
| *OPTIM EPS* | 1E-5 | 1E-5 | 1E-3 | 1E-3 |
| *MAX GRAD NORM* | 10 | 10 | 10 | 10 |
| *HIDDEN DIM* | 64 | 64 | 64 | 64 |
| *COMM DIM* | 64 | 64 | 64 | 64 |
| *ATTENTION DIM* | NONE | 32 | NONE | 32 |
| *EVAL EPISODES* | 32 | 32 | 32 | 32 |

Table 4: Compute time (in hours) across SMAC and Traffic Junction tasks averaged over 8 seeds.

| METHOD | 1O_10B_VS_1R | 3S5Z_VS_3S6Z | 5M_VS_6M | 6H_VS_8Z | MEDIUM | HARD |
| --- | --- | --- | --- | --- | --- | --- |
| COMA | 21 | 26 | 17 | 22 | 8 | 18 |
| MAPPO | 48 | 50 | 20 | 51 | 5 | 11 |
| MAT | 59 | 89 | 31 | 44 | 11 | 19 |
| IPPO-COMM | 50 | 74 | 32 | 70 | 13 | 16 |
| IPPO-COMM-OB-KL | 43 | 78 | 33 | 56 | 15 | 20 |
| G2ANET | 64 | 51 | 29 | 36 | 15 | 36 |
| G2ANET-OB-KL | 56 | 94 | 71 | 55 | 15 | 34 |

evaluation periods in Table 6. GAAC-OB-KL and IPPOComm-OB-KL achieve a higher win rate compared to all the other methods. In the meanwhile, IPPO-Comm-OB-KL and GAAC-OBKL have a lower or similar variance in win rate than other methods except for COMA, which performs much worse in the traffic junction domain.

## D.6 Parameter Choices

Hyper-parameters used for IPPO-Comm-OB-KL and GAAC-OB-KL in the SMAC domain and Traffic Junction are shown in Table 3. Note that we use *hidden dim* to refer to the hidden dimension of the actor and critic model. We use *comm dim* to denote the hidden dimension of the communication model. We further use *attention dim* to denote the hidden dimension of the attention model used by GAAC for aggregating messages from the other agents. Note that hyperparameters for IPPO-Comm-OB-KL and GAAC-OB-KL were optimised using a grid search over learning rate and batch sizes with the grid centred on the hyperparameters used in the original papers (e.g., GAAC and IPPO) and parameter performance tested in all used environments. We further search the optimal parameters introduced by the KL objective (i.e., the temperature parameter and the scaling factor) and report the corresponding performance in Section D.7.

## D.7 Additional Results

**Compute Time**. The experiments reported in the paper were conducted in parallel on a cluster using CPUs (32 cores). Regarding compute time, we set a maximum of 4 days for SMAC tasks and a maximum of 2 days for Traffic Junction tasks. We have summarized the computational cost in terms of wall-clock time for all methods in Table 4 averaged on 8 seeds. The wall clock time can vary across tasks and methods due

Table 5: Median win rate and standard deviation on all evaluated methods in SMAC and Traffic Junction.

| | 1O_10B_VS_1 | 3S5Z_VS_3S6Z | 5M_VS_6M | 6H_VS_8Z | MEDIUM | HARD |
|---|---|---|---|---|---|---|
| COMA | 33.8 (3.3) | 0.0 (0.0) | 0.1 (0.2) | 0.0 (0.0) | 60.9 (6.6) | 42.2 (6.8) |
| MAPPO | 21.5 (2.4) | 65.9 (4.6) | 32.5 (2.4) | 33.9 (3.0) | 71.8 (3.6) | 57.4 (4.9) |
| MAT | 58.0 (5.0) | 4.1 (2.0) | 3.1 (0.9) | 13.5 (2.5) | 72.5 (3.7) | 57.8 (4.9) |
| IPPO-Comm | 2.9 (1.3) | 29.2 (4.2) | 22.2 (2.3) | 35.5 (2.6) | 70.4 (6.4) | 58.2 (4.6) |
| IPPO-Comm-OB | 27.6 (3.4) | 49.3 (4.1) | 70.3 (2.7) | 37.3 (2.2) | 69.4 (3.6) | 61.0 (3.7) |
| IPPO-Comm-KL | 1.5 (0.8) | 25.6 (3.9) | 33.2 (2.8) | 35.8 (2.8) | 71.0 (3.8) | 60.1 (4.6) |
| IPPO-Comm-OB-KL | 42.9 (3.3) | 70.9 (4.0) | 72.8 (2.8) | 56.4 (2.5) | 72.4 (2.7) | 64.2 (3.2) |
| GAAC | 10.9 (2.6) | 0.0 (0.0) | 3.0 (1.1) | 0.0 (0.0) | 70.5 (3.4) | 59.9 (4.6) |
| GAAC-OB | 26.0 (3.6) | 0.7 (0.6) | 32.1 (3.0) | 34.9 (2.4) | 70.3 (3.4) | 58.2 (4.7) |
| GAAC-KL | 2.0 (0.9) | 21.7 (3.5) | 33.2 (2.8) | 36.2 (2.8) | 70.7 (3.2) | 59.7 (4.3) |
| GAAC-OB-KL | 35.0 (2.9) | 23.0 (3.1) | 56.0 (2.6) | 25.3 (2.4) | 73.2 (2.8) | 64.5 (3.0) |

to differences in environment dynamics and implementation details. Additionally, methods that can finish episodes early require fewer time steps, resulting in lower overall wall-clock time to complete a task. As a result, our proposed methods generally do not introduce significant additional computational costs. In particular, the wall-clock time of IPPO-Comm-OB-KL is comparable to that of MAT. While G2Anet-OB-KL incurs higher wall-clock time than G2Anet without OB-KL, it achieves substantially better learning performance. These results suggest that IPPO-Comm-OB-KL maintains a reasonable runtime, and G2Anet-OB-KL demonstrates a trade-off between computational cost and performance gains.

**Variance in Gradient Norm**. We present the variance in gradient norm across training steps for all methods in Figure 6. As shown, GAAC-OB-KL and IPPO-Comm-OB-KL exhibit significantly lower variance in policy gradients throughout training, indicating a more stable learning process compared to CTDE methods and those without the proposed techniques.

**Grid Search**. We conduct a grid search to fine-tune the temperature parameter $\alpha$ and the scaling factor $\beta$ of IPPO-Comm-OB-KL and GAAC-OB-KL. We show the performance of IPPO-Comm-OB-KL and GAAC-OB-KL under different combinations of $\alpha$ and $\beta$ in all 6 tasks (as shown in Figure 7). The label names in plots follow the format of IPPO-Comm-OB-KL_$\alpha$_$\beta$ and GAAC-OB-KL_$\alpha$_$\beta$. Note that we present the performance with the best parameters in the main paper.

Table 6: Bootstrap mean and 95% confidence interval of all evaluated methods in SMAC and Traffic Junction. We mark the maximum mean value in each column in bold and underline.

| | 1O_10B_VS_1 | 3S5Z_VS_3S6Z | 5M_VS_6M | 6H_VS_8Z | MEDIUM | HARD |
|---|---|---|---|---|---|---|
| COMA | 33 (33,34) | 0 (0,0) | 0 (0,0) | 0 (0,0) | 60 (59,62) | 42 (40,43) |
| MAPPO | 21 (21,21) | 65 (64,66) | 32 (32,32) | 33 (33,34) | 71 (71,72) | 57 (56,58) |
| MAT | **58** (57,58) | 4 (3,4) | 3 (2,3) | 13 (13,14) | 72 (71,73) | 57 (56,58) |
| IPPO-Comm | 2 (2,3) | 29 (28,30) | 22 (21,22) | 35 (34,35) | 70 (69,71) | 58 (57,59) |
| IPPO-Comm-OB | 27 (26,28) | 49 (48,50) | 70 (69,70) | 37 (36,37) | 69 (68,70) | 61 (60,61) |
| IPPO-Comm-KL | 1 (1,1) | 25 (24,26) | 33 (32,33) | 35 (35,36) | 71 (70,71) | 60 (59,60) |
| IPPO-Comm-OB-KL | 42 (42,43) | **70** (70,71) | **72** (72,73) | **56** (55,56) | 72 (71,72) | 64 (63,64) |
| GAAC | 10 (10,11) | 0 (0,0) | 3 (2,3) | 0 (0,0) | 70 (69,71) | 59 (58,60) |
| GAAC-OB | 26 (25,26) | 0 (0,0) | 32 (31,32) | 34 (34,35) | 70 (69,70) | 58 (57,59) |
| GAAC-KL | 2 (1,2) | 21 (21,22) | 33 (32,33) | 36 (35,36) | 70 (70,71) | 59 (58,60) |
| GAAC-OB-KL | 35 (34,35) | 23 (22,23) | 56 (55,56) | 25 (24,25) | **73** (72,73) | **64** (63,65) |

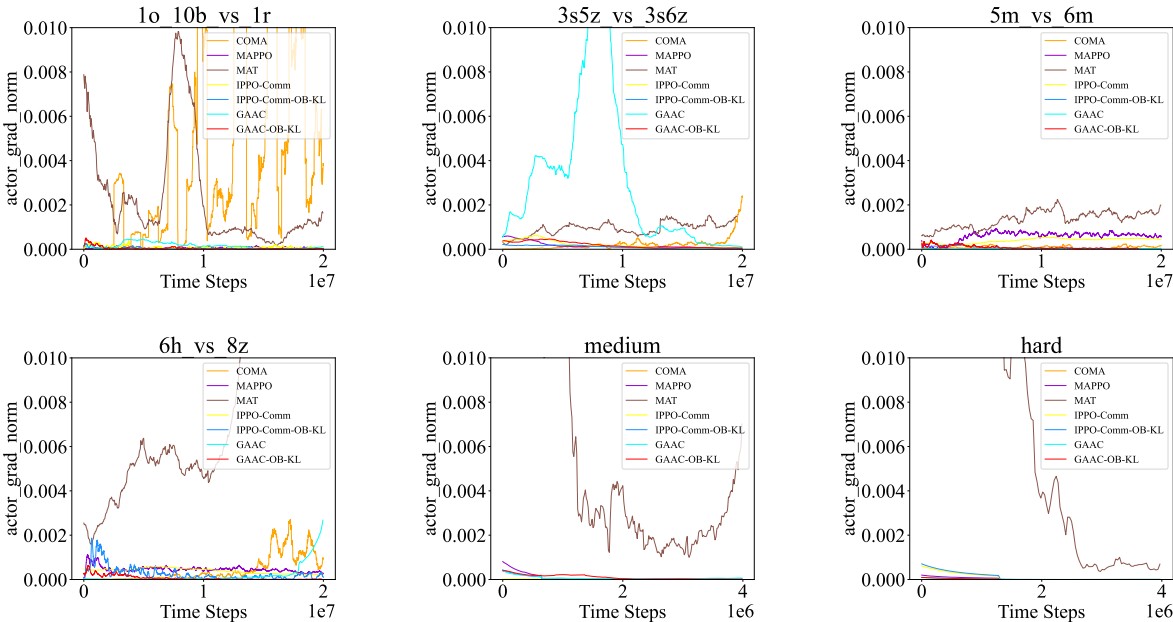

Figure 6: Variance in policy gradient norm of all methods.

## E  Limitations

DCCDA aims to eliminate communication between actors, enabling independent execution without communication. However, during training, value functions (critics) are allowed to communicate, guiding the updates of each agent's policy (actors). Therefore, DCCDA enables agents to benefit from communication to improve learning efficiency during training through communicating critics, without compromising their ability to execute the policies independently at execution time. Regarding the limitations, in scenarios where communication is both feasible and unrestricted during training and execution, DCCDA and the proposed methods may be less suitable.

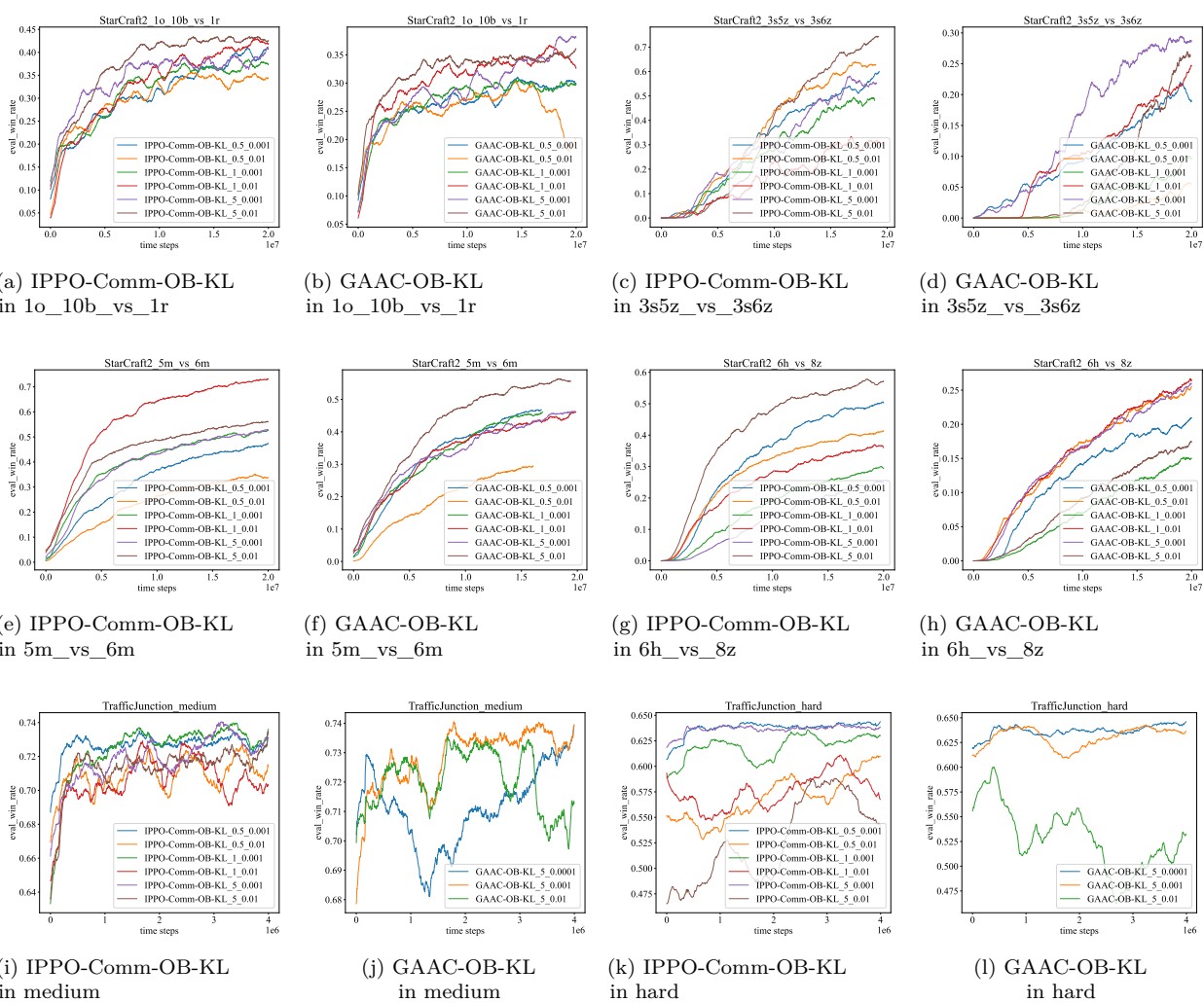

Figure 7: Averaged win rate under different $\alpha$ and $\beta$ in 4 tasks of SMAC and 2 tasks of Traffic Junction.

