# OpenReview forum: "Mitigating Variance Caused by Communication in Decentralized Multi-agent Deep Reinforcement Learning"
_TMLR — Rejected by TMLR_

### Review · Reviewer_Wzkq · 2025-06-13

**Summary Of Contributions:**

The authors of this paper claim to make the first theoretical analysis of variance in policy gradients caused by communication in decentralized multi-agent deep reinforcement learning (MADRL), specifically under the Decentralized Communicating Critics and Decentralized Actors (DCCDA) setting, where communication is allowed only among critics during training. They prove that such communication can increase gradient variance compared to centralized critics, even under ideal communication. To address this, they propose a novel message-dependent baseline that directly targets variance introduced by stochastic messages and derive its optimal form. Additionally, they introduce a KL-divergence-based regularization to align decentralized actors with communicating critics. The authors claim these techniques are modular, model-agnostic, and can be applied to any DCCDA-based method. Empirical results on benchmark environments (SMAC and Traffic Junction) show that integrating these techniques leads to lower gradient variance, improved learning performance, and enhanced training stability, outperforming strong baselines including CTDE methods.

**Audience:**

No

**Claims And Evidence:**

No

**Requested Changes:**

- The authors need to clearly explain why their method qualifies as communication. Can agents complete a pure communication task? For example, the **referential game**? When agent 1 sees a private observation and wants to tell agent 2 through communication to help agent 2 make better use of the information to reach a better outcome, can the proposed method do this?
- Provide a more detailed comparison with MADDPG and explain why DCCDA is not just a variant of CTDE.

**Strengths And Weaknesses:**

This paper discusses a concept of _communication_ that differs from what is generally accepted in the community. By definition, communication involves transmitting some kind of information. For example:

- In fully cooperative tasks, such as referential games, agents use signals to inform uninformed parties about the current state. Most current MARL (Multi-Agent Reinforcement Learning) research focuses on this kind of situation.
- In mixed-motive tasks, communication can take the form of _persuasion_, such as Bayesian persuasion / information design.
- Or it can serve as a _coordination device_, like traffic lights, which help achieve a correlated equilibrium.

All these forms of communication are based on the agents’ models and allow information to be transmitted during execution.

However, the DCCDA in this paper refers to something used in the _training phase_, which is actually a kind of training trick that isn’t used during execution. This is more like _information aggregation_, and resembles a CTDE paradigm, which similar to the extension mentioned in MADDPG. The paper’s discussion of MADDPG is insufficient to clearly distinguish the two approaches. Simply having each agent maintain its own network is not enough to justify introducing a new framework name separate from CTDE.

---

> ### Author Response · Authors · 2025-06-26
> **Response to Reviewer Wzkq**
>
> We sincerely thank the reviewer for the valuable feedback and constructive comments. We appreciate that the reviewer highlights the paper's novel contribution. We will answer the requested changes as follows:
>
> **Request 1**:
>
> We agree that we do not consider communication as in referential games or mixed-motive tasks, or serving as a coordination device. Neither do we allow information to be transmitted during execution. Indeed, the DCCDA proposed in our work uses information aggregation in the training phase. It is similar but still fundamentally different from CTDE. Information exchange in DCCDA qualifies as communication because agents are distributed in the environment with partial observations and individual histories. During training, agents share encoded information (using individual histories and actions) as messages. With DCCDA, agents can neither complete a pure communication task nor communicate during execution as in referential games.
>
> **Request 2**:
>
> We would like to clarify that the setting DCCDA is fundamentally different from CTDE and the method MADDPG. In DCCDA, messages are learned without being predefined, essentially based on the literature on learning communication in multi-agent deep reinforcement learning (Comm-MADRL) [1]. In CTDE, the centralized critic needs to fully access all agents’ partial information, including their histories and actions, i.e., $Q(h_1,…h_N, a_1,…,a_N)$. Similarly, in MADDPG, each agent holds a joint Q-function as $Q_i(h_1,…h_N, a_1,…,a_N)$, which also needs to access all agents’ information. In comparison, in DCCDA, the partial information of agents are encoded as a low-dimensional vector (as messages), i.e., $Q(h_i, a_i,m_{-i})$, where $m_{-i}= \\{ m_1,…,m_{i-1}, m_{i+1},m_N \\}$ is the received messages, which can significantly reduce the input dimensionality compared to the centralized Q-function.
>
> In order to show the differences in architecture between DCCDA and CTDE methods, we compare the diagram for DCCDA (See Figure 4 on page 30) and the diagram for CTDE (See Figure 5 on page 30) to show the differences in their (forward) structure of neural networks. Specifically, in the diagrams, we use IPPO-Comm-OB-KL (the DCCDA method) and MAPPO (the CTDE method) as examples to show how we train the actors, critics, and the extended communication module. Note that IPPO-Comm-OB-KL and MAPPO differ only in their critics (and communication) and the way to train the critics (and communication). It will be easier to identify the differences between IPPO-Comm-OB-KL and MAPPO.
>
> As shown in the two diagrams, the red arrows refer to the training phase. In IPPO-Comm-OB-KL, the actor modules, communication (comm) modules, and critic modules are the neural networks. Compared to MAPPO, IPPO-Comm-OB-KL involves a message-generation process using the actor and comm modules and the communication channel. DCCDA agents need to encode, select, and transmit their information.
>
> We added these discussions to a new subsection in Appendix D (see page 30) and mentioned this in the revised manuscript (see page 3).
>
> [1] C. Zhu, M. Dastani, and S. Wang, "A Survey of Multi-Agent Deep Reinforcement Learning with Communication," Auton. Agents Multi-Agent Syst., vol. 38, no. 4, 2024.

---

> > ### Comment · Reviewer_Wzkq · 2025-06-26
> >
> > Issue 1: “Communication” Terminology
> >
> > I appreciate the additional clarifications, but the core problem remains: the work does **not** study a communication problem. We agree that
> > > the DCCDA proposed in our work uses information aggregation in the training phase.
> >
> >  That is precisely the point. Information aggregation during training is not communication in the multi-agent-learning sense (i.e., no referential game, no explicit coordination device).
> >  Consequently, the current title and framing mis-represent the technical contribution.
> >
> > The classification logic in Figure 1 is muddled. Only 2 and 4 are communication-related; only these two types of algorithms, which introduce communication within execution, have sufficient capacity to handle communication tasks.
> >
> >
> > Issue 2: Relationship to CTDE
> >
> > MARL frameworks exist to produce a good game solver; at deployment we only care about the solver’s quality, while training may relax assumptions—for example, letting agents share information—precisely why centralized training was introduced. Centralized training can use information during training to coordinate agents and help them form **conventions.** During execution, this auxiliary information is no longer needed because the conventions formed are stable and desirable **equilibria.** Removing this information at execution time does not affect the conventions, since they have already reached this stable point.
> >
> > Your motivation and your statements
> > - "the DCCDA proposed in our work uses information aggregation in the training phase."
> > - "DCCDA aims to eliminate communication between actors, enabling independent execution without communication. However, during training, value functions (critics) are allowed to communicate, guiding the updates of each agent’s policy (actors). Therefore, DCCDA enables agents to benefit from communication to improve learning efficiency during training through communicating critics, without compromising their ability to execute the policies independently at execution time."
> > are exactly a textbook justification for centralized training in CTDE.
> >
> > Once they share information, aggregation _requires_ a centralized device. The newly added Figure 4 (DCCDA) on page 30 is merely a concrete instantiation of Figure 5 (CTDE): you’ve cracked open an end-to-end black-box critic and inserted hand-crafted components. At best, this is a _variant_ within CTDE, not a framework parallel to it.
> >
> > ---
> >
> > For these reasons I maintain my original assessment and do not change my score.

---

> > > ### Comment · Reviewer_Wzkq · 2025-06-27
> > >
> > > To clarify, regarding issue 2, I understand that for each agent you maintain a critic that only takes as input the trajectory within its own observation range, so it is “decentralized.”
> > >
> > > However, I would like to point out that you rely on the assumptions required by CTDE. The agents are not fully decentralized — they need "communication" to perform information aggregation, and your justification for this "communication" is exactly the same as that of CTDE, which means you are using CTDE’s assumptions. Once you make this assumption, the advantages of “decentralization” in your framework setting are invalid — *it does not increase the applicability* of the framework.
> > >
> > > In other words, your **motivation** for decentralizing the critic directly **contradicts the justification** for allowing "communication".
> > >
> > >
> > > Note: The word “communication” is put in quotation marks here to refer specifically to what the authors consider information aggregation through exchanging intermediate representations during training.
> > >
> > > If anyone points out that there is a misunderstanding in my reasoning, I am open to changing my assessment.

---

> > > > ### Author Response · Authors · 2025-06-27
> > > >
> > > > We are uncertain if we understand your comments correctly. In the DCCDA setting, training is decentralized which is fundamentally different from centralized training, as used in CTDE. For example, the centralized training in CTDE is known to suffer from the curse of dimensionality [1], while DCCDA is more computationally efficient.
> > > >
> > > > We want to clarify that we use the term DCCDA to denote the existing and well-recognized MARL setting (e.g., [2,3]) where critics do “communicate” with each other during training, while the actors do not communicate during the execution. Such a setting is well-known within MARL with communication literature (see, e.g., [4]). Our work provides the theoretical contributions for this setting by analyzing the variance properties of communication under the DCCDA setting. So, we believe that exchanging information among critics during decentralized training (with non-communicating actors) is a legitimate and well-recognized form of MARL with communication. In this sense, we believe that the title does well-represent the technical contribution of the paper.
> > > >
> > > > DCCDA employs information aggregation in the training phase, but it is still considered a form of communication. Essentially, during communication, the messages are signals/vectors from an embedded space, which is a standard way of modelling communication in the MADRL with communication literature [4]. Moreover, communication is a subject of learning, along with the learning of critics and actors. We would like to note that during decentralized learning of DCCDA, the exchanged signals/vectors become meaningful communication elements since the signals/vectors become correlated with domain elements.
> > > >
> > > > In your latter comment you write:
> > > >
> > > > “The agents are not fully decentralized — they need "communication" to perform information aggregation, and your justification for this "communication" is exactly the same as that of CTDE, which means you are using CTDE’s assumptions.”
> > > > May we ask which CTDE assumption you are referring to? We also do not understand why the agents are not fully decentralized. Technically, in DCCDA, both actors and critics are trained in a decentralized manner using their respective buffers and sampled experience. In comparison, CTDE needs to store and sample the joint experience of all agents during training. The fact that agents communicate does not make DCCDA as CTDE since each agent maintains their own experience and trains in a decentralized fashion.
> > > >
> > > > [1] S. Gronauer, K. Diepold. "Multi-agent deep reinforcement learning: a survey." Artificial Intelligence Review, 55(2), 895-943 (2022).
> > > >
> > > > [2] Yong Liu, Weixun Wang, Yujing Hu, Jianye Hao, Xingguo Chen, Yang Gao. Multi-Agent Game Abstraction via Graph Attention Neural Network. AAAI 2020: 7211-7218.
> > > >
> > > > [3] Shariq Iqbal, Fei Sha. Actor-Attention-Critic for Multi-Agent Reinforcement Learning. ICML 2019: 2961-2970.
> > > >
> > > > [4] C. Zhu, M. Dastani, and S. Wang, "A Survey of Multi-Agent Deep Reinforcement Learning with Communication," Auton. Agents Multi-Agent Syst., vol. 38, no. 4, 2024.

---

> > > > > ### Comment · Reviewer_Wzkq · 2025-06-28
> > > > >
> > > > > To make the point sharp: DCCDA is a form of constrained CTDE, and DCCDA can equivalently be implemented by a CTDE.
> > > > >
> > > > > According to your reply to me on June 25th, the decentralized critic in DCCDA is $Q_i(h_i, a_i, m_{-i})$, while the centralized critic is $Q_i(h_1, \ldots, h_N, a_1, \ldots, a_N)$, as in MADDPG.
> > > > > - According to your communication module in Figure 4, $m_j = f(h_j, a_j)$, where $f$ is a function implemented by a neural network.
> > > > > - Therefore, the decentralized critic is $Q_i(h_i, a_i, m_{-i}) = Q_i(h_i, a_i, \{f(h_j, a_j)\}_{j \ne i}) = Q'_i(h_1, \ldots, h_N, a_1, \ldots, a_N)$.
> > > > > - Since everything is implemented end-to-end with neural networks, there is essentially no difference.
> > > > >
> > > > > Below is the pseudocode for a 2-agent scenario, according to Figure 4:
> > > > >
> > > > > ```
> > > > > Input: (h_1, h_2)
> > > > > Forward:
> > > > > 	h_1 -> decentralized_MLP_1 -> m_1
> > > > > 	h_2 -> decentralized_MLP_2 -> m_2
> > > > > 	(h_1, m_2) -> decentralized_critic_1 -> Q_1
> > > > > 	(h_2, m_1) -> decentralized_critic_2 -> Q_2
> > > > > Output: (Q_1, Q_2)
> > > > > ```
> > > > >
> > > > > Focus on $Q_1$:
> > > > >
> > > > > ```
> > > > > Input: (h_1, h_2)
> > > > > Forward:
> > > > > 	h_2 -> decentralized_MLP_2 -> m_2
> > > > > 	(h_1, m_2) -> decentralized_critic_1 -> Q_1
> > > > > Output: (Q_1)
> > > > > ```
> > > > >
> > > > > In order to obtain $Q_1$, its input still needs to be the joint history. This can equivalently be implemented by a centralized critic:
> > > > >
> > > > > ```
> > > > > Input: (h_1, h_2)
> > > > > Forward:
> > > > > 	(h_1, m_2) -> centralized_critic_1 -> Q_1
> > > > > Output: (Q_1)
> > > > > ```
> > > > >
> > > > > ---
> > > > > ---
> > > > >
> > > > >
> > > > > Assumptions/conditions used in CTDE: We only focus on the execution phase during final deployment, so **during the training phase we can make full use of available information** to guide the agents toward a certain equilibrium.
> > > > >
> > > > > I mentioned this in my previous reply:
> > > > >
> > > > > > Fully cooperative MARL frameworks exist to produce a good game solver; at deployment we only care about the solver’s quality, while training may relax assumptions—for example, letting agents share information—precisely why centralized training was introduced.
> > > > >
> > > > >
> > > > > The motivation of DCCDA mentioned in your reply to Reviewer tp1u:
> > > > >
> > > > > > Our proposed setting, DCCDA, is critical and advantageous in practical applications where communication between various actors may either raise security concerns, require resources that are not available, or communication cannot be reliable during execution.
> > > > >
> > > > > This makes sense, but it only explains why there is no communication during execution. As for why the critic is allowed to communicate during training, your explanation is:
> > > > >
> > > > > > In this and other similar practical applications, DCCDA aims to eliminate communication between actors, enabling independent execution without communication. However, during training, value functions (critics) are allowed to communicate, guiding the updates of each agent’s policy (actors). Therefore, DCCDA enables agents to benefit from communication to improve learning efficiency during training through communicating critics, without compromising their ability to execute the policies independently at execution time.
> > > > >
> > > > > Allowing the critic to communicate during the training phase is essentially about making full use of data during training — this is exactly the insight behind CTDE.
> > > > >
> > > > > However, this still does not explain why a decentralized critic is needed. It’s important to note that the critic is only ever used during the training phase, not during execution. Therefore, if you need a decentralized agent system in an actual task, you would only require decentralized actors, not decentralized critics.
> > > > >
> > > > > In practice, if you need a decentralized critic during training, this implies:
> > > > > - There may be communication constraints (but in that case, the method cannot be based on communication in the first place)
> > > > > - You need to leverage certain desirable properties of DTDE [1] (However, an end-to-end communication framework makes a DTDE framework revert back to CTDE)
> > > > >
> > > > > In any case, introducing DTDE together with communication is contradictory. Regarding why a communication framework makes a DTDE framework revert back to CTDE: according to your definition, communication is understood as connections within the neural network (and according to your design in Figure 4), which is the insight behind CommNet [2] and BiCNet [3].
> > > > >
> > > > > This is also why I said:
> > > > >
> > > > > > The motivation for decentralizing the critic directly contradicts the justification for allowing communication.
> > > > >
> > > > > - [1] Lyu, Xueguang, et al. "On centralized critics in multi-agent reinforcement learning." _Journal of Artificial Intelligence Research_ 77 (2023): 295-354.
> > > > > - [2] Sukhbaatar, Sainbayar, and Rob Fergus. "Learning multiagent communication with backpropagation." _Advances in neural information processing systems_ 29 (2016).
> > > > > - [3] Peng, Peng, et al. "Multiagent bidirectionally-coordinated nets: Emergence of human-level coordination in learning to play starcraft combat games." _arXiv preprint arXiv:1703.10069_ (2017).

---

> > > > > > ### Author Response · Authors · 2025-06-29
> > > > > > **Response to reviewer Wzkq**
> > > > > >
> > > > > > We do not agree that DCCDA can equivalently be implemented by a CTDE. Essentially, DCCDA is not trained in the same way as CTDE, where DCCDA uses decentralized training while CTDE uses centralized training. The pseudocode provided by the reviewer only considers the forward pass of the neural network, while it does not consider the training of the neural network.
> > > > > >
> > > > > > During training, multi-agent deep reinforcement learning uses a replay buffer to store its experiences and sample the experiences from the buffer for training the Q-values. In DCCDA, we only need to store individual history $h_i$, actions $a_i$, and the received messages $m_{-i}$ of each agent $i$ for sampling experience and training the Q-values, i.e., we do not require either storing or sampling the histories $h_{-i}$ and actions $a_{-i}$ of the other agents. In other words, in DCCDA, the training of Q-values $Q_i(h_i,a_i,m_{-i})$ ignores the other agents’ histories $h_{-i}$ and actions $a_{-i}$. This is essentially different from centralized training used by CTDE, where the centralized Q-values require storing all agents’ experience, i.e., $h_i, a_i, h_{-i}, a_{-i}$, and sampling the joint experience in every training loop. The pseudocode of the decentralized training in DCCDA and the centralized training in CTDE is shown below:
> > > > > >
> > > > > > In DCCDA, we have:
> > > > > >
> > > > > > ```
> > > > > > Input: Replay buffer of histories, actions, and received messages
> > > > > > Backward:
> > > > > > 	For each agent i:
> > > > > > 		Sampling mini-batches of (h_i, a_i, m_{-i})
> > > > > > 		(h_i, a_i, m_2) -> decentralized_critic_i -> Q_i
> > > > > > 		Compute TD error and backward gradients
> > > > > > ```
> > > > > >
> > > > > > In CTDE, we have:
> > > > > >
> > > > > > ```
> > > > > > Input: Replay buffer of histories and actions
> > > > > > Backward:
> > > > > > 	Sampling mini-batches of (h_1, a_1, h_2, a_2)
> > > > > > 	(h_1, a_1, h_2, a_2) -> centralized_critic -> Q
> > > > > > 	Compute TD error and backward gradients
> > > > > >
> > > > > > ```
> > > > > >
> > > > > > As we have stated in the previous reply, the centralized critic suffers from the curse of dimensionality, i.e., the sample complexity grows exponentially with the input dimensionality. However, DCCDA utilizes low-dimensional messages instead of the full history and actions of agents, which enjoys computational efficiency.
> > > > > >
> > > > > > We do not agree that an end-to-end communication framework makes a DTDE framework revert back to CTDE. DCCDA and CTDE have different input dimensionality of Q-functions, sampling, and training process of Q-functions. DCCDA is not gaining insights from the end-to-end centralized training frameworks CommNet or BiCNet. CommNet uses a jointly shared policy rather than Q-functions for all agents (which is obviously identified by Figure 1 in the paper of CommNet). Moreover, CommNet performs mean operations on the hidden layers of all agents, which essentially centralizes training, i.e., requiring the joint experience of agents for sampling and training as a big neural network, which can not be implemented as decentralized training and covered by DCCDA. Also, instead of the mean operations, BiCNet uses a bidirectional RNN to connect all agents’ hidden layers, which also can not be implemented as decentralized training and covered. Moreover, the hidden layers involved in the mean operation (used in CommNet) and the RNN structure (used in BiCNet) at every time step are usually forbidden to be stored in the experience buffer and perform sampling due to the high dimensionality. Therefore, neither CommNet or BiCNet can be incorporated into DCCDA.

---

> > > > > > > ### Comment · Reviewer_Wzkq · 2025-06-29
> > > > > > >
> > > > > > > I have tried my best to explain, but it seems that the authors still cannot understand what I’m trying to convey. There’s also the NeurIPS review deadline in a couple of days, so I’m unable to continue any further discussion. I believe I’ve fulfilled my duty so this will be my last reply:
> > > > > > >
> > > > > > > - The definition of CTDE is exactly what you stated in your reply to me on June 26th, and I think that’s correct — this is supported by some literature [1]. I thought we had already reached a consensus on this. What you’re doing now is: after I pointed out the contradiction in your story, you’re trying to blur it again by dragging in “learning patterns” instead of arguing based on the definition directly — so this is invalid.
> > > > > > >     - If you remove parameter sharing in a CTDE setting and maintain a separate buffer for each agent, but the input is still the joint history, can you call this decentralized?
> > > > > > >     - All other CTDE algorithms have some modifications and specific learning designs, but none of them claim they’re not CTDE but an entirely new framework. Why is it that just because you store intermediate variables ($m_j$ can be obtained from other agents’ history via an MLP), this suddenly becomes a new decentralized framework?
> > > > > > >     - Before my previous reply, I checked: the feedback in your method doesn’t even involve intermediate values of the communication module but goes from the two ends of the module instead, and the tasks are fully cooperative — which makes it even less “decentralized.”
> > > > > > >     - Whether the algorithm has benefits is a separate issue — I do agree that storing intermediate variables might be beneficial. However, the motivation behind your story does not make sense. If you admit that it’s CTDE with improvements on top of it, then these advantages would actually be worth further discussion.
> > > > > > > - I was not referring to the specific implementations of CommNet and BiCNet, but to the insight that they treat communication as neural network connections.
> > > > > > > - “CommNet and BiCNet can be incorporated into DCCDA.” — I have never made such a statement; this is _not_ my claim. Refuting a point that I never made clearly shows that you didn’t understand my entire argument.
> > > > > > >
> > > > > > > Lastly, I want to emphasize that the authors have also never directly responded to my main concerns: The motivation for decentralizing the critic directly contradicts the justification for allowing communication.
> > > > > > >
> > > > > > > [1] Lyu, Xueguang, et al. "On centralized critics in multi-agent reinforcement learning." _Journal of Artificial Intelligence Research_ 77 (2023): 295-354.

---

> > > > > > > > ### Author Response · Authors · 2025-06-30
> > > > > > > > **Response to the final comments of reviewer Wzkq**
> > > > > > > >
> > > > > > > > In our view, our disagreements can be summarized as follows:
> > > > > > > > - The reviewer seems to believe that an end-to-end communication in a DTDE framework reverts to CTDE, and that our work is therefore CTDE with improvements on top of it. We disagree with this view because the communicating critics and (non-communicating) actors in our setting are still trained in a decentralized fashion, which is inherently different from centralized training used in CTDE  (see the **pseudocode** posted on 29 Jun 2025 at 11:39, and the example with **QMIX** below).
> > > > > > > >
> > > > > > > > - The reviewer seems to believe that introducing DTDE together with communication is contradictory. The decentralization of critics and actors in our setting helps reduce the computational complexity involved in learning (i.e., mitigating the curse of dimensionality in centralized training), and communication among critics improves the learning efficiency of actors. Moreover, our focused setting is an existing and well-recognized paradigm in the literature [1,2].
> > > > > > > >
> > > > > > > > Now we respond to the last comment of the reviewer.
> > > > > > > >
> > > > > > > > We can imagine that you did your best to explain your point, but hopefully, you can also see that we do our best to understand your argument and do our best to reply directly to your comments. If we do not converge to a conclusion, it may be that we cannot understand each other or do not agree with each other’s views. We also have the impression that you do not understand our arguments and replies.
> > > > > > > >
> > > > > > > > ```
> > > > > > > > The definition of CTDE is exactly what you stated in your reply to me on June 26th, and I think that’s correct — this is supported by some literature. I thought we had already reached a consensus.
> > > > > > > > ```
> > > > > > > >
> > > > > > > > In our reply on June 26, we first tried to indicate that we understand your comment and agreed that our work does not consider communication in the way it is done in referential games and that agents do not communicate during execution (we still agree with this). It is great that you now indicate that we understood your position. However, you seem to ignore what we wrote afterwards, i.e., “information exchange in DCCDA qualifies as communication because agents are distributed in the environment with partial observations and individual histories. During training, agents share encoded information (using individual histories and actions) as messages. With DCCDA, agents can neither complete a pure communication task nor communicate during execution as in referential games.”
> > > > > > > >
> > > > > > > > We seem to disagree not at the technical level, but at a conceptual level. We do our best, but do not agree that decentralizing critics contradicts communication. We miss a logical formulation of this claimed contradiction. Communicating critics during the training phase is an existing approach in the MADRL literature with communication. You do not reply to this point, which we raised several times in our interaction. We believe it is quite a restricted view that you try to impose on MADRL frameworks with communication.
> > > > > > > >
> > > > > > > > In general, you seem not to recognize our technical contribution that shows rigorously that communication between critics increases variance that can be mitigated by concrete techniques, because you have a specific conceptual view on MADRL frameworks with communication. We do not agree that one can talk about communication in MADRL frameworks only if communication is done in an identical way as done in referential games. We believe, like other authors in MADRL with communication literature (as we stated in our previous responses), that MADRL frameworks where critics communicate during training can also be considered as a MADRL framework with communication.
> > > > > > > >
> > > > > > > > You seem to refer to ‘arguing based on definition directly’ without indicating which definition you have in mind. As far as we are concerned, we did not propose a definition that is in contradiction with the existing standard definition.
> > > > > > > >
> > > > > > > > ```
> > > > > > > > after I pointed out the contradiction in your story, you’re trying to blur it again by dragging in “learning patterns” instead of arguing based on the definition directly — so this is invalid.
> > > > > > > >
> > > > > > > > If you remove parameter sharing in a CTDE setting and maintain a separate buffer for each agent, but the input is still the joint history, can you call this decentralized?
> > > > > > > > ```
> > > > > > > >
> > > > > > > > When we do depart from CTDE to a decentralized setting, in the extreme case, where each agent can indeed see the joint history of all agents, each agent will maintain a joint critic, necessarily accessing the full information of all agents. Despite this, as long as agents are training in a decentralized fashion, i.e., they have independent training processes, and separate replay buffers (as the reviewer pointed out), we view this as DTDE. However, this does require lossless communication channels, which may not be applicable in realistic scenarios. In both our theoretical and technical settings, we do consider imperfect communication channels where messages can be noisy due to the imperfection of encoders.

---

> > > > > > > > > ### Author Response · Authors · 2025-06-30
> > > > > > > > > **Response to the final comments of reviewer Wzkq**
> > > > > > > > >
> > > > > > > > > ```
> > > > > > > > > All other CTDE algorithms have some modifications and specific learning designs, but none of them claim they’re not CTDE but an entirely new framework. Why is it that just because you store intermediate variables ($m_j$ can be obtained from other agents’ history via an MLP), this suddenly becomes a new decentralized framework?
> > > > > > > > > ```
> > > > > > > > >
> > > > > > > > > We would like to point out that all other CTDE algorithms use centralized training rather than decentralized training. For example, for the well-known value-based CTDE algorithms, QMIX [1], where the joint Q-function is decomposed into several individual Q-functions, the training of all agents requires the joint experience of all agents in every sample, i.e., the Q-values for all agents are evaluated under the joint experience of agents. Take a 2-agent case as an example, QMIX decomposes the joint Q-value Q(\tau_1, \tau_2, a_1, a_2) into Q(\tau_1, a_1) and Q(\tau_2, a_2). So during execution, only Q(\tau_1, a_1) and Q(\tau_2, a_2) are needed to perform decentralized execution. During training, the training of Q(\tau_1, a_1) and Q(\tau_2, a_2) requires (\tau_1, a_1) and (\tau_2, a_2) coming from the same sample (so the same time step). Therefore, (\tau_1, \tau_2, a_1, a_2) must be stored together for every time step, to ensure the tuples (\tau_1, a_1) and (\tau_2, a_2) come from the same sample when doing random sampling. However, imagine that QMIX is allowed to perform decentralized training, (\tau_1, a_1) and (\tau_2, a_2) are stored separately. During the decentralized training, (\tau_1, a_1) and (\tau_2, a_2) are not necessarily coming from the same time step (since we perform random sampling separately for each replay buffer), and the joint Q-value Q(\tau_1, \tau_2, a_1, a_2) becomes invalid.
> > > > > > > > >
> > > > > > > > > Therefore, it is not only because we store messages separately in the replay buffer, but also because agents in decentralized training have independent training processes (we specifically mean the sampling process).
> > > > > > > > >
> > > > > > > > > ```
> > > > > > > > > I was not referring to the specific implementations of CommNet and BiCNet, but to the insight that they treat communication as neural network connections.
> > > > > > > > >
> > > > > > > > > “CommNet and BiCNet can be incorporated into DCCDA.” — I have never made such a statement; this is not my claim. Refuting a point that I never made clearly shows that you didn’t understand my entire argument.
> > > > > > > > > ```
> > > > > > > > >
> > > > > > > > > We are sorry that we do not mean that you have stated CommNet and BiCNet can be incorporated into DCCDA. We do not agree that “a communication framework makes a DTDE framework revert back to CTDE”. We were clarifying the differences between DCCDA and the communication methods CommNet and BiCNet. We agree that CommNet and BiCNet treat communication as neural network connections. However, we would like to further clarify that CommNet and BiCNet can never be implemented in a decentralized fashion, since the neural network of agents in CommNet and BiCNet can not be broken into independent components due to the mean operations (used by CommNet) and the RNN structure (used by BiCNet).
> > > > > > > > >
> > > > > > > > > ```
> > > > > > > > > Lastly, I want to emphasize that the authors have also never directly responded to my main concerns: The motivation for decentralizing the critic directly contradicts the justification for allowing communication.
> > > > > > > > > ```
> > > > > > > > >
> > > > > > > > > We do not agree with this argument (see above).
> > > > > > > > >
> > > > > > > > > [1] Y. Liu, W. Wang, Y. Hu, J. Hao, X. Chen, and Y. Gao, “Multi-Agent Game Abstraction via Graph Attention Neural Network,” in AAAI, 2020, pp. 7211–7218.
> > > > > > > > >
> > > > > > > > > [2] C. Zhu, M. Dastani, and S. Wang, “A Survey of Multi-Agent Deep Reinforcement Learning with Communication,” Autonomous Agents and Multi-Agent Systems, vol. 38, no. 4, 2024.

---

> > > > > > > > > > ### Comment · Reviewer_Wzkq · 2025-06-30
> > > > > > > > > >
> > > > > > > > > > I’m not sure when the response deadline for TMLR is, but, after reading this reply, I realized I had to respond, because the author was misrepresenting my stance. I once said:
> > > > > > > > > >
> > > > > > > > > > > The definition of CTDE is exactly what you stated in your reply to me on June 26th, and I think that’s correct — this is supported by some literature [1]. I thought we had already reached a consensus on this.
> > > > > > > > > >
> > > > > > > > > > I made it very clear here that by “consensus” I was referring to the definition of CTDE. What I meant by saying you were right was the CTDE description in MADDPG—that is, your own reply:
> > > > > > > > > >
> > > > > > > > > > > In CTDE, the centralized critic needs to fully access all agents' partial information, including their histories and actions, i.e., $Q\left(h_1, \ldots h_N, a_1, \ldots, a_N\right)$. Similarly, in MADDPG, each agent holds a joint Q-function as $Q_i\left(h_1, \ldots h_N, a_1, \ldots, a_N\right)$, which also needs to access all agents' information. In comparison, in DCCDA, the partial information of agents are encoded as a low-dimensional vector (as messages), i.e., $Q\left(h_i, a_i, m_{-i}\right)$, where $m_{-i}=\{m_1, \ldots, m_{i-1}, m_{i+1}, m_N\}$ is the received messages, which can significantly reduce the input dimensionality compared to the centralized Q-function.
> > > > > > > > > >
> > > > > > > > > > The definition of CTDE here and the definition of your model are correct — I have no objections. What I mean by the definition is that the input to the centralized critic needs to include the joint history, which is supported by many papers; I will list some of them later. My point here is that under your definition, I pointed out the equivalence between the two, because this definition only requires the critic’s input. Therefore, DCCDA is just CTDE.
> > > > > > > > > >
> > > > > > > > > > It must be pointed out that CTDE is a very broad concept, and it’s hard to get outside this framework. As long as this condition is met, it’s CTDE: during training, information that is unavailable during execution is used — such information includes the environment state and opponents’ private information. In the counterexample you gave with the buffer, the buffer stores $m_{-i}$, so it still makes use of extra information. I will give the relevant statements from the literature below.
> > > > > > > > > >
> > > > > > > > > > As far as I know, the only situation that is recognized as decentralized in the community is when the agents are independent — for example, IPPO. These algorithms are usually used as baselines. In other words, only when the critic is $Q(h_i, a_i)$ can it be considered decentralized. As long as there is any data exchange that is not allowed during execution — including storing intermediate message values — it cannot be considered decentralized. That’s the definition. This is exactly why I said "The motivation for decentralizing the critic directly contradicts the justification for allowing communication." Because communication inevitably involves exchanging private data.
> > > > > > > > > >
> > > > > > > > > > The authors have a point regarding their definition of DTDE:
> > > > > > > > > > > As long as agents are training in a decentralized fashion, i.e., they have independent training processes, and separate replay buffers (as the reviewer pointed out), we view this as DTDE.
> > > > > > > > > >
> > > > > > > > > > This is not correct. Independent processes do not guarantee that their training data is also independent. There is relevant literature that shows this:
> > > > > > > > > >
> > > > > > > > > > "Learning to Communicate with Deep Multi-Agent Reinforcement Learning"
> > > > > > > > > > > We focus on settings with centralised learning but decentralised execution. In other words, communication between agents is not restricted during learning, which is performed by a centralised algorithm; however, during execution of the learned policies, the agents can communicate only via the limited-bandwidth channel.
> > > > > > > > > > > ...
> > > > > > > > > > > The second approach, which we call differentiable inter-agent learning (DIAL), is based on the insight that centralised learning affords more opportunities to improve learning than just parameter sharing.
> > > > > > > > > >
> > > > > > > > > > This indicates that CTDE does not necessarily require parameter sharing. Therefore, agents do not have to share a single network. They can each maintain their own critic. And thus, of course, the networks are trained by themselves.
> > > > > > > > > >
> > > > > > > > > > Also, your insight — that communication is unrestricted during training but restricted during execution — aligns perfectly with the CTDE formulation here.

---

> > > > > > > > > > > ### Comment · Reviewer_Wzkq · 2025-06-30
> > > > > > > > > > >
> > > > > > > > > > > Here is more:
> > > > > > > > > > >
> > > > > > > > > > > "Counterfactual Multi-Agent Policy Gradients"
> > > > > > > > > > > > Since learning is centralised, we can therefore use a centralised critic that conditions on the joint action and all available state information, while each agent’s policy conditions only on its own action-observation history.
> > > > > > > > > > >
> > > > > > > > > > >
> > > > > > > > > > > "Multi-Agent Actor-Critic for Mixed Cooperative-Competitive Environments"
> > > > > > > > > > > > We adopt the framework of centralized training with decentralized execution, allowing the policies to use extra information to ease training, so long as this information is not used at test time.
> > > > > > > > > > > > ...
> > > > > > > > > > > > Since the centralized critic function explicitly uses the decision-making policies of other agents, we additionally show that agents can learn approximate models of other agents online and effectively use them in their own policy learning procedure.
> > > > > > > > > > >
> > > > > > > > > > > This indicates that CTDE uses others’ decision-making policies. In your example, that’s the other agents’ communication MLP. The buffer storage is simply the difference between online and offline.
> > > > > > > > > > >
> > > > > > > > > > > What I mean is that during training you also need to use the messages sent by others earlier. First, this is extra information in the execution phase, so by definition it’s no longer decentralized. Second, these messages actually implicitly contain others’ private histories and their decision-making processes, because `opponent_message = opponent_MLP(opponent_private_information)`. So you’re actually making use of others’ decision policies in the training phase. Otherwise, you cannot get the needed training data.
> > > > > > > > > > >
> > > > > > > > > > > The papers mentioned above are canonical works in the MARL field and are among the earliest papers that define the concept of CTDE. Each of them has at least 2,000 citations. At this point, this issue should be clearly explained. All the other insights were my attempts to help the authors understand, but it doesn’t matter if they don’t get it, because I have now laid out the specific statements. The reason I take this distinction so seriously is that if this work gets accepted, it could lead to the abuse of the concept of "decentralized." So my stance is firm: unless the authors acknowledge that they are using CTDE and restate the motivation of the story, I will not change my assessment.
> > > > > > > > > > >
> > > > > > > > > > >
> > > > > > > > > > > Regarding the communication concept: I admit that the authors are right on this point. My definition of communication was too restricted, as I was previously focused on it from a game theory perspective. But clarifying this distinction in the paper would be helpful.

---

> > > > > > > > > > > > ### Author Response · Authors · 2025-07-01
> > > > > > > > > > > > **Response to Reviewer Wzkq**
> > > > > > > > > > > >
> > > > > > > > > > > > Dear reviewer,
> > > > > > > > > > > >
> > > > > > > > > > > > Thank you for your further explanation. We would like to first clarify that our implementations (the two DCCDA methods) are indeed based on IPPO and follow exactly the same training mechanism. We understand your concern about the potential misuse of the term “decentralization”. However, our theoretical analysis has considered the relation between communicating critics and centralized critics, as presented in Lemma 1 of the Appendix: under lossless communication, communicating critics become equivalent to centralized critics in expectation. In other words, DCCDA becomes equivalent to CTDE under lossless communication. On the other hand, with noisy communication, communicating critics are no longer equivalent to centralized critics unless an unbiased surrogate reward function is introduced, as shown in Lemma 3 of the Appendix.
> > > > > > > > > > > >
> > > > > > > > > > > > We acknowledge your perspective on CTDE, which can be very broad; however, we followed the existing approaches that were considered as a decentralized setting [1,2]. Moreover, our implementations are fundamentally based on the DTDE method IPPO, it could be confusing to refer to the implemented algorithms as CTDE. Despite this, if you think that we should not follow this view, we can add a discussion about this issue by summarizing some part of our discussion (including the issue of whether actors can be considered as independent, and thus the setting can be considered as DTDE, if the actors use aggregated information obtained by communicating critics that are not available during execution), and change the title of the paper to “Mitigating Variance Caused by Communicating Critics in Multi-agent Deep Reinforcement Learning”. Moreover, we will explicitly discuss the relation between communicating critics and centralized critics under both lossless and noisy communication, and narrow down our definition of “decentralized communicating critics” by explicitly stating that the implemented algorithms are built on IPPO. Would this address your comment?
> > > > > > > > > > > >
> > > > > > > > > > > > We are not sure whether you will accept these changes, but we sincerely hope you will consider raising the score. The main contribution of our paper is the variance analysis, which aims to address the stability of learning. We will do our best to reduce any confusion for the general audience, so that readers can focus more clearly on our core contributions.
> > > > > > > > > > > >
> > > > > > > > > > > > [1] Y. Liu, W. Wang, Y. Hu, J. Hao, X. Chen, and Y. Gao, “Multi-Agent Game Abstraction via Graph Attention Neural Network,” in AAAI, 2020, pp. 7211–7218.
> > > > > > > > > > > >
> > > > > > > > > > > > [2] C. Zhu, M. Dastani, and S. Wang, “A Survey of Multi-Agent Deep Reinforcement Learning with Communication,” Autonomous Agents and Multi-Agent Systems, vol. 38, no. 4, 2024.

---

### Review · Reviewer_TTmP · 2025-06-14

**Summary Of Contributions:**

The authors present theoretical analyses and algorithms for multi-agent deep reinforcement learning. In particular, they study the actor-critic scenarios where the critics (but not the agents) can communicate with each other during training and then the agents act independently, without communication, at execution time. They call this situation DCCDA, for Decentralized Communicating Critics and Decentralized Agents. The communication between critics can introduce variance in the policy gradients. The authors present a technique for mitigating such variance using a message-dependent baseline technique in conjunction with a Kullback-Leibler divergence regularization term. Positive results are presented on 2 benchmark domains: a Starcraft battle video-game-like domain and a traffic simulation in which many cars drive on a 2-way road.

**Audience:**

Yes

**Broader Impact Concerns:**

No specific broader impact concerns. Generally speaking, the potential applications to autonomous vehicles and military systems need to be contemplated by anyone working in this area but I don't think the paper needs a specific additional statement here.

**Claims And Evidence:**

No

**Requested Changes:**

Can you explain more specifics about how the experiments illustrate the theoretical concepts of DCCDA and of the methods? E.g. what are the messages passed back and forth by the critics in Starcraft and Traffic Junction and why are the messages stochastic?

Can you clarify whether the variance of a policy gradient is a vector or scalar quantity and whether the directional variance (like cosine similarity of gradients) is of interest as well as the norm?

Do you have results on a third domain?

I'm gonna say 'No' for now on claims and evidence but could well switch to "Yes' with some revisions. In any case, my review should not get much weight by the editors, since I don't know the field well at all.

**Strengths And Weaknesses:**

I do not work in reinforcement learning or in multi-agent systems and I alerted the journal editors to my lack of experience in these areas when given this review assignment. While I enjoyed reading the paper and am glad for the chance to review it, I apologize ahead of time if some of my comments are not well-informed. I think the best I can do here is to offer some observations which may help the paper be accessible to a broader audience without requiring too much prior knowledge of the specific field.

As best I can tell, the paper should be of interest to at least some readers of TMLR. It certainly looks likely to me (although I can't say for sure) that the study of policy gradient variance induced by communication among critics in multi-agent RL is a novel contribution with practical significance.

My comments mostly have to do with clarity. My biggest concern here is that the challenges having to do with communication-induced policy gradient variance are not well explained, particularly when the experiments are introduced. The authors say that messages are generated in a stochastic manner (page 2) and that this induces policy gradient variance. This seems to be a  core issue addressed by the paper, so I think it needs more explanation. I wish this stochasticity had been explained in more detail. Is the source of randomness simply the randomness of the transition function to the new states? Or are the authors saying that given 2 situations with the exact same state and history, a critic may generate a different message at one time than at another time because they generate messages with some randomness? If so, why do  they do that? What is the purpose or reason for this randomness? Is it to try different random initializations in the learning process? When interacting with opponents in a game-theoretic situation, I can see why you might need randomness, but it is not clear to me where the randomness comes from here.  It would have been helpful to spend more time when introducing the Starcraft and Traffic Junction experiments explaining how the various aspects of the DCCDA situation map onto the concrete domains. E.g. what are the nature of the messages passed back and forth by the Traffic Junction critics and why do they need to be stochastic?

I also found the whole discussion of policy gradient variances to be a bit confusing insofar as a gradient is a vector quantity (and hence the variance thereof would also be, I would think) yet much of the math seems to treat it as a scalar. This is probably my fault- I think there may be some assumed shorthand here which readers more familiar with the field may know- but I thought I'd let the authors know that it's confusing to someone fairly new to the field. On page 11, the authors talk about the variance of gradient *norms*, which makes more sense since that's a scalar quantitiy. So I wasn't sure if this "norm" was implicit in the other dicussions of gradient variance or what the deal is here. I would think that there would be interest in both the angular/directional variance of the gradients as well as their norm. So maybe some more explanation here would be helpful.

I don't think 'idealistic' is the best term for the noise-free scenario. I would go with maybe 'noise-free' or 'lossless'.

I also think 2 experimental domains is maybe just adequate, but a little light as far as experiments go. A third would be nice.

---

> ### Author Response · Authors · 2025-06-26
> **Response to Reviewer TTmP**
>
> We sincerely thank the reviewer for the valuable feedback and constructive comments. We appreciate that the reviewer highlights the paper's novel contribution with practical significance. We will answer the requested changes as follows:
>
> **Request 1**:
>
> Given the history determined by probabilistic observation function and actions determined by stochastic policy in the Dec-POMDP setting (Section 3.1), the messages reflect this stochasticity, i.e., the uncertainty involved in the history and actions are lifted to the stochasticity of messages. Because of this, we assume messages are sampled from a probabilistic distribution, which is standard in the implementation of MADRL with communication (see, e.g., [1]).
>
> In both StarCraft and Traffic Junction, messages are a vector of values generated by a probabilistic message function using a softmax output layer. Stochasticity in these environments can arise from multiple sources. For instance, in both scenarios, agents observe the relative positions of other agents within their vision range, which can vary stochastically due to the movement of other agents or enemies. In Traffic Junction specifically, new cars are added to the environment at each timestep with a certain probability. Additionally, both tasks involve learning a stochastic policy. These characteristics introduce randomness in observations, transitions, and policies. When agents communicate, the use of a probabilistic message function allows them to capture and reflect this inherent stochasticity present in the environment.
>
> We added a paragraph under Appendix D.2 to explain the use of messages under practical environments (see page 31).
>
> **Request 2**:
>
> We would like to clarify that we focus on the total variance of the policy gradient vector, which measures the overall variability across all dimensions of the vector, following the variance analysis and notations in the literature [2]. Note that the total variance is a scalar rather than a vector. The variance analysis of the policy gradient vector necessarily involves a covariance matrix, where the diagonal elements denote the variance of each gradient component. The total variance is given by the trace of the covariance matrix, that is, the sum of its diagonal elements, incorporating the squared norm of the gradient vector. In order to show the magnitude of the gradients under different methods, we compare the norm of the policy gradient vector, following the work [3]. We clarified these in the revised Appendices A.1 (on page 21) and B.2 (on page 27), and the revised manuscript on page 11.
>
> **Request 3**:
>
> In our experiments, the two environments Traffic Junction and SMAC are used to evaluate the performance of two DCCDA methods. We used multiple tasks in every environment, with a diverse number of agents, roles, and terrain, which cover different communication requirements and dynamics. For example, the tested ‘1o10b_vs_1r’ task in SMAC contains an overseer that detects the enemy while 10 other agents act to kill the enemy. Compared with tasks that only contain agents with the same role (e.g. 5m_vs_6m), this task involves agents with multiple roles and therefore requires different communication. Therefore, we think our rich and diverse tested tasks and environments are sufficient to evaluate various aspects of MADRL.
>
> **For Broader Impact Concerns**
>
> We deleted the broader impact concerns as suggested.
>
> [1] T. Wang, J. Wang, C. Zheng, and C. Zhang, "Learning Nearly Decomposable Value Functions via Communication Minimization," ICLR, 2020.
>
> [2] X. Lyu, Y. Xiao, B. Daley, and C. Amato, "Contrasting Centralized and Decentralized Critics in Multi-Agent Reinforcement Learning," AAMAS, 2021, pp. 844–852.
>
> [3] J. G. Kuba, M. Wen, L. Meng, S. Gu, H. Zhang, D. Mguni, J. Wang, and Y. Yang, "Settling the Variance of Multi-Agent Policy Gradients," NeurIPS, 2021.

---

### Review · Reviewer_868L · 2025-06-17

**Summary Of Contributions:**

This paper analyzes variance introduced by communication in decentralized multi-agent actor-critic reinforcement learning, proving it never falls below centralized baselines under both idealistic and noisy channels. It proposes an optimal message-dependent baseline and a KL-based policy regularizer that reduces variance, incorporates them into GAAC and an IPPO-Comm variant, and demonstrates stability and performance gains on SMAC and Traffic Junction benchmarks.

**Audience:**

Yes

**Broader Impact Concerns:**

No ethical concerns in this paper.

**Claims And Evidence:**

Yes

**Requested Changes:**

1. How would the variance bounds change if messages were quantized or bandwidth-limited, introducing non-Gaussian discretization noise?
2. Given mini-batch baseline estimation, what bias-variance trade-offs arise, and could adaptive baselines (e.g., exponential moving averages) further stabilize updates?
3. Does the KL regularizer risk constraining exploration or biasing the policy away from optimality; under what conditions might its influence need to be removed?
4. I suggest adding some discussion on centralized multi-agent RL literature such as [1][2][3].

Reference:

[1] Dubey, Abhimanyu, and Alex Pentland. "Provably efficient cooperative multi-agent reinforcement learning with function approximation." arXiv preprint arXiv:2103.04972 (2021).

[2] Hsu, Hao-Lun, et al. "Randomized exploration in cooperative multi-agent reinforcement learning." arXiv preprint arXiv:2404.10728 (2024).

[3] Min, Yifei, et al. "Cooperative multi-agent reinforcement learning: asynchronous communication and linear function approximation." International Conference on Machine Learning. PMLR, 2023.

**Strengths And Weaknesses:**

Strengths:
1. Provides the first formal variance analysis of policy gradients with communicating critics, covering both perfect and noisy messaging and showing variance is larger than CTDE baselines.
2. Derives a closed-form message-dependent baseline that provably minimizes gradient variance, which extends classic baseline theory to the communication dimension.
3. Introduces a KL-based actor–critic alignment regularizer and demonstrates its plug-and-play use in two distinct algorithms (GAAC, IPPO-Comm), confirming model-agnostic applicability.

Weaknesses:
1. Variance proofs hinge on the assumption that decentralized critics converge to the exact centralized value in expectation; the impact of function-approximation error is not analyzed.
2. Baseline and KL terms require accurate Q-value estimates, yet the paper gives no complexity or sample-efficiency bounds on their estimation, leaving practical variance control unquantified.
3. Hyper-parameters $\alpha$ (temperature) and $\beta$ (KL weight) are selected by grid search without theoretical guidance or extensive sensitivity analysis, limiting reproducibility and insight.

---

> ### Author Response · Authors · 2025-06-26
> **Response to Reviewer 868L**
>
> We sincerely thank the reviewer for the valuable feedback and constructive comments. We also appreciate the reviewer for highlighting our theoretical contributions. We answer the requested changes as follows:
>
> **Request 1**:
>
> We would like to first clarify that in our variance analysis, we consider discrete messages and therefore do not consider discretization noise to be Gaussian (used for continuous messages) or non-Gaussian. With discrete messages, we use summations over message values, while continuous-valued messages require integrals.
>
> On the other hand, if we consider messages as continuous values and explicitly quantize messages, regardless of whether the noise is Gaussian or non-Gaussian, if a surrogate reward function is well-defined, i.e., an unbiased surrogate reward function can be achieved (See Lemma 3 in Appendix B on page 25), the variance properties established in our work still hold. We consider that messages are quantized or bandwidth-limited, introducing non-Gaussian discretization noise as a future work. We incorporated this in the revised manuscript on page 12.
>
> **Request  2**:
>
> We agree with the reviewer that baseline values depend on Q-value estimates. When using mini-batch baseline estimation in an iterative training fashion, the baseline values can vary across mini-batches even for the same agent history and action, since the sampled Q-values may originate from different iterations. However, in our case, we use on-policy algorithms (e.g., IPPO), where experience (i.e., mini-batches) is refreshed after each training iteration (at the end of an episode). Consequently, baseline values are computed using the latest Q-value estimates and remain fixed within each iteration. As a result, the policy gradients remain unbiased. Nonetheless, the exact amount of variance reduction still depends on the baseline estimation, which may lead to either greater or lesser variance reduction. We added these discussions to a new subsection under Appendix D.3 on page 32, and mentioned this on page 8.
>
> Regarding adaptive baselines (e.g., exponential moving averages), one well-known example is Generalized Advantage Estimation (GAE) [1]. While GAE is based on state-value functions rather than Q-functions, it cannot be directly applied to our setting. A potential adaptation would be to replace the Q-function in our proposed baseline with $r_i + V(h_i)$ using the state-value function from the GAE. We consider integrating GAE as a direction for future work. We incorporate this into the revised manuscript on page 12.
>
> **Request 3**:
>
> The KL divergence term discourages the actor from selecting actions to which the critic assigns a low value (See page 8). This may constrain the exploration of the actor, especially when the critic is learning and non-optimal. In practice, we think the negative influence needs to be removed when critics get stuck in a local optimum, which will always push actors back to the local optimum. We added these discussions to a new subsection under Appendix D.3 on page 32, and mentioned this on page 8.
>
> **Request 4**:
>
> We have incorporated the suggested papers into the revised manuscript (See page 4). Essentially, these works consider communication between decentralized agents and a central server, while our work does not assume a central server.
>
> [1] Schulman et al., "High-Dimensional Continuous Control Using Generalized Advantage Estimation," ICLR (Poster), 2016.

---

### Review · Reviewer_tp1u · 2025-06-19

**Summary Of Contributions:**

This paper focuses on the Decentralized Communicating Critics and Decentralized Actors (DCCDA) setting in multi-agent deep reinforcement learning, where agents can communicate only during training through their critics, but execute policies independently without communication in both training and inference. The paper provides the first theoretical analysis of the variance in policy gradients under the DCCDA setting. Based on it, the paper proposes message-dependent baseline and KL regularization for boosting the existing MARL method. The proposed techniques are integrated into two existing Comm-MADRL algorithms and evaluated on benchmark environments.

**Audience:**

Yes

**Claims And Evidence:**

No

**Requested Changes:**

* Could the authors provide more analysis of the proposed method in real-world scenarios, with their strength and limitations?

* Could the authors provide more evidence for the assumptions of the noise and how the method will work under different types of noise?

* Could the authors analyze how the computational cost grows with the number of agents or environment complexity, and compare it with existing baselines?

**Strengths And Weaknesses:**

### Strengths:
1. The writing and illustration are clear and easy to follow.
1. The theoretical analysis of variance in policy gradients under both idealistic and non-idealistic communication settings is appealing.

### Weakness

1. While the studied DCCDA setting seems interesting, the paper could better highlight real-world applications where these setups are critical. The assumption that agents can communicate freely during training but the communications are completely absent at execution lacks strong real-world justification.
2. While the non-idealistic setting is analyzed, the noise model (binary reward flipping) seems simple.
3. The KL regularization relies on Boltzmann policies derived from Q-values, which might not always align well with the true optimal policy, especially in sparse-reward settings.
4. The computational cost of the proposed techniques is not thoroughly analyzed. Table 4 mentions runtime but lacks discussion on scalability with larger agent populations or more complex environments.

---

> ### Author Response · Authors · 2025-06-26
> **Response to Reviewer tp1u**
>
> We sincerely thank the reviewer for the valuable feedback and constructive comments. We also appreciate the reviewer for highlighting our theoretical contributions. We answer the requested changes as below:
>
> **Request 1**:
>
> Our proposed setting, DCCDA, is critical and advantageous in practical applications where communication between various actors may either raise security concerns, require resources that are not available, or communication cannot be reliable during execution. A typical practical application is low-energy UAVs [1]. When a UAV operates in an open environment with other moving UAVs, maintaining communication channels during execution over time can slow down data collection, drain battery life, and increase the risk of data leakage.
>
> In this and other similar practical applications, DCCDA aims to eliminate communication between actors, enabling independent execution without communication. However, during training, value functions (critics) are allowed to communicate, guiding the updates of each agent’s policy (actors). Therefore, DCCDA enables agents to benefit from communication to improve learning efficiency during training through communicating critics, without compromising their ability to execute the policies independently at execution time.
>
> Regarding the limitations, in scenarios where communication is both feasible and unrestricted during training and execution, DCCDA and the proposed methods may be less suitable. We added a section (Appendix E) to the Appendix and discussed strengths and limitations on page 35.
>
> **Request 2**:
>
> We would like to first clarify that the noisy reward model, specifically binary reward flipping, was adopted to simplify the variance analysis and to demonstrate how value estimation is affected by noisy messages, which in turn impacts the variance of policy gradients. As stated on page 23, this noisy reward model can be extended beyond binary rewards by following the work of [2], which employs a confusion matrix where each entry represents the probability of a true reward being observed as a different value. In this non-binary reward setting, conducting a variance analysis requires assuming the domain knowledge of how noise alters each reward.
>
> Regarding the noise itself, we assume it can be categorized into two types: those that alter a given reward and those that do not, as described on page 23. For simplicity in the theoretical analysis, we also assume the noise is discrete, allowing the use of summation over all possible noisy values, as shown in Equations 19–21. Furthermore, theoretically speaking, under different types of noise, such as Uniform or Poisson distribution, our theoretical results hold as long as a surrogate reward function is defined. Lemma 3 (see page 25) generalizes to different types of noise distributions.
>
> On the other hand, different types of noise can impact Q-value estimation, and consequently, our proposed method can be affected when Q-value estimates become degraded, leading to fluctuations in the baseline values. As we noted in our response to Reviewer _868L_, when using mini-batch baseline estimation in an iterative training fashion, baseline values can vary across mini-batches, even for the same agent history and action, because the sampled Q-values may come from different training iterations. However, in our work, we employ on-policy algorithms (e.g., IPPO), where experience (i.e., mini-batches) is refreshed after each training iteration (at the end of an episode). As a result, baseline values are computed using up-to-date Q-value estimates and remain fixed within each iteration. This ensures that the policy gradients remain unbiased. Nonetheless, the effectiveness of variance reduction still depends on the accuracy of the baseline estimates, which can vary in mini-batches depending on the Q-value estimation. We added these discussions to a new subsection under Appendix D.3 on page 32, and mentioned this on page 8.
>
> **Request 3**:
>
> We have summarized the computational cost in terms of wall-clock time for all methods in the revised Table 4 (see page 34). The wall clock time can vary across tasks and methods due to differences in environment dynamics and implementation details. As shown in the revised Table 4, our proposed methods generally do not introduce significant additional computational costs. In particular, the wall clock time of IPPO-Comm-OB-KL is comparable to that of MAT. While G2Anet-OB-KL incurs higher wall-clock time than G2Anet without OB-KL, it achieves substantially better learning performance. These results suggest that IPPO-Comm-OB-KL maintains a reasonable runtime, and G2Anet-OB-KL demonstrates a trade-off between computational cost and performance gains. We incorporated these discussions in Appendix D.7 on page 35.
>
> [1] Oubbati et al., Synchronizing UAV Teams by Deep Reinforcement Learning, IEEE Trans. Veh. Technol., 2022.
>
> [2] Wang et al., Reinforcement Learning with Perturbed Rewards, AAAI, 2020.

---

### Author Response · Authors · 2025-06-26
**Response to all reviewers**

We sincerely thank all reviewers for their valuable feedback and constructive comments, which have greatly improved the quality of our paper. We have responded to each reviewer and uploaded a revised version of the manuscript. The revised content is highlighted in BLUE.

---

### Decision · Action_Editor_Nsg9 · 2025-08-13

**Recommendation:** Reject

**Additional Comments:**

See above.

**Audience:**

Yes

**Audience Explanation:**

The concrete results—how stochastic critic messaging affects policy-gradient variance, the empirical gains from the proposed message-dependent baseline, and the KL-regularised policy updates—are of direct interest to researchers who design CTDE algorithms. These readers will value the variance analysis and the plug-and-play techniques once the paper is recast as a CTDE study.

**Claims And Evidence:**

No

**Claims Explanation:**

The review process has crystallized a single, overriding conceptual objection. Reviewers Wzkq, tp1u and 868L contend that the moment critics exchange any information during training—even through low-dimensional learned messages—the proposed DCCDA construction collapses into a specific instance of CTDE. All claims about “decentralised critics with communication” or “variance reduction in a decentralised framework” must therefore be restated as claims about a CTDE variant.

This mis-framing undermines every subsequent motivation for the noise model, the KL regulariser and the variance analysis. Second, the paper offers no systematic study of how computation or communication scale with the number of agents or environment complexity—Table 4 lists runtimes but omits scalability curves and baseline comparisons. Third, the empirical evaluation relies exclusively on a simplistic binary reward-flipping noise model, without evidence that the proposed techniques survive under other noise regimes. Finally, the Boltzmann-based KL regulariser, built on Q-value logits, may misalign with the true optimal policy in sparse-reward settings, yet this limitation is neither analysed nor mitigated.

While addressing the terminology issue is straightforward, the deeper narrative must be re-engineered and the missing empirical evidence added. These deficiencies currently prevent the manuscript from meeting the technical-bar threshold for acceptance, and the paper is therefore recommended for rejection.

**Resubmission Of Major Revision:**

The authors may consider submitting a major revision at a later time.